# Dcp2 C-terminal *cis*-binding elements control selective targeting of the decapping enzyme by forming distinct decapping complexes

**Feng He\*, Chan Wu, Allan Jacobson\***

Department of Microbiology and Physiological Systems, University of Massachusetts Chan Medical School, Worcester, United States

**Abstract** A single Dcp1–Dcp2 decapping enzyme targets diverse classes of yeast mRNAs for decapping-dependent 5′ to 3′ decay, but the molecular mechanisms controlling mRNA selectivity by the enzyme remain elusive. Through extensive genetic analyses we reveal that Dcp2 C-terminal domain *cis*-regulatory elements control decapping enzyme target specificity by orchestrating formation of distinct decapping complexes. Two Upf1-binding motifs direct the decapping enzyme to nonsense-mediated mRNA decay substrates, a single Edc3-binding motif targets both Edc3 and Dhh1 substrates, and Pat1-binding leucine-rich motifs target Edc3 and Dhh1 substrates under selective conditions. Although it functions as a unique targeting component of specific complexes, Edc3 is a common component of multiple complexes. Scd6 and Xrn1 also have specific binding sites on Dcp2, allowing them to be directly recruited to decapping complexes. Collectively, our results demonstrate that Upf1, Edc3, Scd6, and Pat1 function as regulatory subunits of the holo-decapping enzyme, controlling both its substrate specificity and enzymatic activation.

**\*For correspondence:**
feng.he@umassmed.edu (FH);
allan.jacobson@umassmed.edu
(AJ)

**Competing interest:** See page
30

**Reviewing Editor:** Ruben L
Gonzalez, Columbia University,
United States

## Editor's evaluation

This paper is an important contribution to the fields of mRNA decapping and mRNA decay. Using a series of elegant genetic approaches and assays, this study tackles the difficult challenge of mapping binding interactions and subsequently assigning functions to binding partners involved in mRNA decapping. As a result, the findings reported in this work represent an important milestone towards deepening our mechanistic understanding of mRNA decapping and mRNA decay.

## Introduction

Decapping, the removal of mRNA 5′ cap structures, is a fundamental step in eukaryotic mRNA turnover that commits a transcript to complete 5′ to 3′ exoribonucleolytic digestion by Xrn1 (*Grudzien-Nogalska and Kiledjian, 2017*; *Parker, 2012*). Decapping plays a key role in general 5′ to 3′ mRNA decay (*Decker and Parker, 1993*), nonsense-mediated mRNA decay (NMD) (*He and Jacobson, 2001*), AU-rich element-mediated mRNA decay (*Fenger-Grøn et al., 2005*; *Pedro-Segura et al., 2008*), microRNA-mediated gene silencing (*Behm-Ansmant et al., 2006*; *Rehwinkel et al., 2005*), and transcript-specific degradation (*Badis et al., 2004*; *Dong et al., 2007*). In *Saccharomyces cerevisiae*, mRNA decapping is carried out by a single enzyme comprised of the Dcp1 regulatory subunit and the Dcp2 catalytic subunit. Dcp1 is a small EVH domain protein (*Beelman et al., 1996*; *She et al., 2004*) and Dcp2 is a 970-amino acid protein containing a highly conserved catalytic domain at its

N-terminus and a largely disordered C-terminal domain embedded with multiple regulatory elements (*Charenton et al., 2017*; *Dunckley and Parker, 1999*; *Gaudon et al., 1999*; *He and Jacobson, 2015*).

In addition to the Dcp1–Dcp2 decapping enzyme, yeast mRNA decapping is also regulated by several proteins collectively known as decapping activators, including Upf1, Pat1, Lsm1, Dhh1, Scd6, Edc1, Edc2, and Edc3 (*Parker, 2012*). Upf1 is required for degradation of nonsense-containing mRNAs (*He and Jacobson, 2001*; *He et al., 2003*). Pat1, Lsm1, and Dhh1 were originally thought to be required for general mRNA decay (*Bouveret et al., 2000*; *Coller et al., 2001*; *Fischer and Weis, 2002*; *Hatfield et al., 1996*; *Tharun et al., 2000*), but recent evidence indicates that they each target a specific subset of yeast mRNAs (*He et al., 2018*). Scd6 targets a limited number of specific transcripts for decapping (*Zeidan et al., 2018*). Edc3 was also originally thought to be required for general mRNA decapping (*Kshirsagar and Parker, 2004*), but this factor exhibits exquisite substrate specificity and appears to target just two transcripts, *YRA1* pre-mRNA and *RPS28B* mRNA (*Badis et al., 2004*; *Dong et al., 2007*). Edc1 and Edc2 were isolated as high-copy suppressors of *dcp1* and *dcp2* mutations (*Dunckley et al., 2001*), and these two factors can stimulate mRNA decapping in vitro (*Borja et al., 2011*; *Steiger et al., 2003*), but they do not appear to be required for mRNA decapping in vivo at least under normal growth conditions. Whether Edc1 and Edc2 also target-specific mRNAs have not been investigated. Several specific functions, including translation repression, decapping enzyme activation, and codon optimality sensing have been proposed for decapping activators (*Coller and Parker, 2005*; *Nissan et al., 2010*; *Radhakrishnan et al., 2016*), but each of the proposed functions is subject to sufficient controversy or uncertainty to render their actual roles largely unknown (*Arribere et al., 2011*; *He and Jacobson, 2015*; *Sweet et al., 2012*; *Webster et al., 2018*).

Mechanistic investigations of mRNA decapping over the last two decades have mostly used biochemical and structural approaches and focused on the catalytic mechanisms of the decapping enzyme (*Charenton and Graille, 2018*; *Valkov et al., 2017*). These studies provided significant insights into the structures of the Dcp1–Dcp2 decapping enzyme (*She et al., 2006*; *She et al., 2004*; *She et al., 2008*), its conformational dynamics (*Floor et al., 2012*; *Wurm et al., 2017*), cap- and RNA-binding properties (*Deshmukh et al., 2008*; *Floor et al., 2010*), and catalytic mechanisms (*Aglietti et al., 2013*). These experiments also revealed binding patterns of Edc1, Edc3, and Pat1 to the decapping enzyme and suggested potential functions or mechanisms of action of these factors in enzymatic activation (*Charenton et al., 2017*; *Charenton et al., 2016*; *Fromm et al., 2012*; *Mugridge et al., 2018*; *Paquette et al., 2018*; *Valkov et al., 2016*; *Paquette et al., 2018*; *Valkov et al., 2016*). However, these biochemical and structural studies all used C-terminally truncated Dcp2, isolated peptides or domains from the decapping activators, and generic mRNA substrates. Thus, some of the proposed mechanisms for decapping activation that emerged from these studies may need additional validation. Further, these experiments did not address how the decapping enzyme is targeted to different substrate mRNAs.

Our recent genetic experiments demonstrated that the Dcp2 C-terminal domain plays a crucial role in the control of mRNA decapping (*He et al., 2018*; *He and Jacobson, 2015*). This domain contains an inhibitory element (IE) and a set of specific linear binding elements for several decapping activators, including Edc3 (E3), Upf1 (U1$_1$, U1$_2$), and Pat1 (L$_1$–L$_9$) (*Figure 1A*). Here, we have generated specific *DCP2* deletions that eliminated either a single element or combinations of different elements and analyzed the consequences of these deletions on Dcp2 interactions with specific decapping activators and decay of different decapping substrates. Our experiments uncover the molecular mechanisms that control the selective targeting of the yeast decapping enzyme and reveal the functional contributions of Upf1, Edc3, Scd6, Dhh1, and Pat1 in decapping of their respective mRNA targets.

## Results
### Loss of both Upf1-binding motifs eliminates Upf1's binding to Dcp2 and causes selective partial stabilization of NMD substrates

Wild-type (WT) Dcp2 exhibited a strong two-hybrid interaction with Upf1. Deletion of either the first (U1D1) or the second (U1D2) Upf1-binding motif had no discernible effect on Upf1's binding to Dcp2, but loss of both Upf1-binding motifs (U1D1–U1D2) eliminated Upf1 binding to Dcp2 (*Figure 1B*). In contrast, loss of both Upf1-binding motifs affected neither Edc3 nor Pat1 binding to Dcp2 (*Figure 1B*).

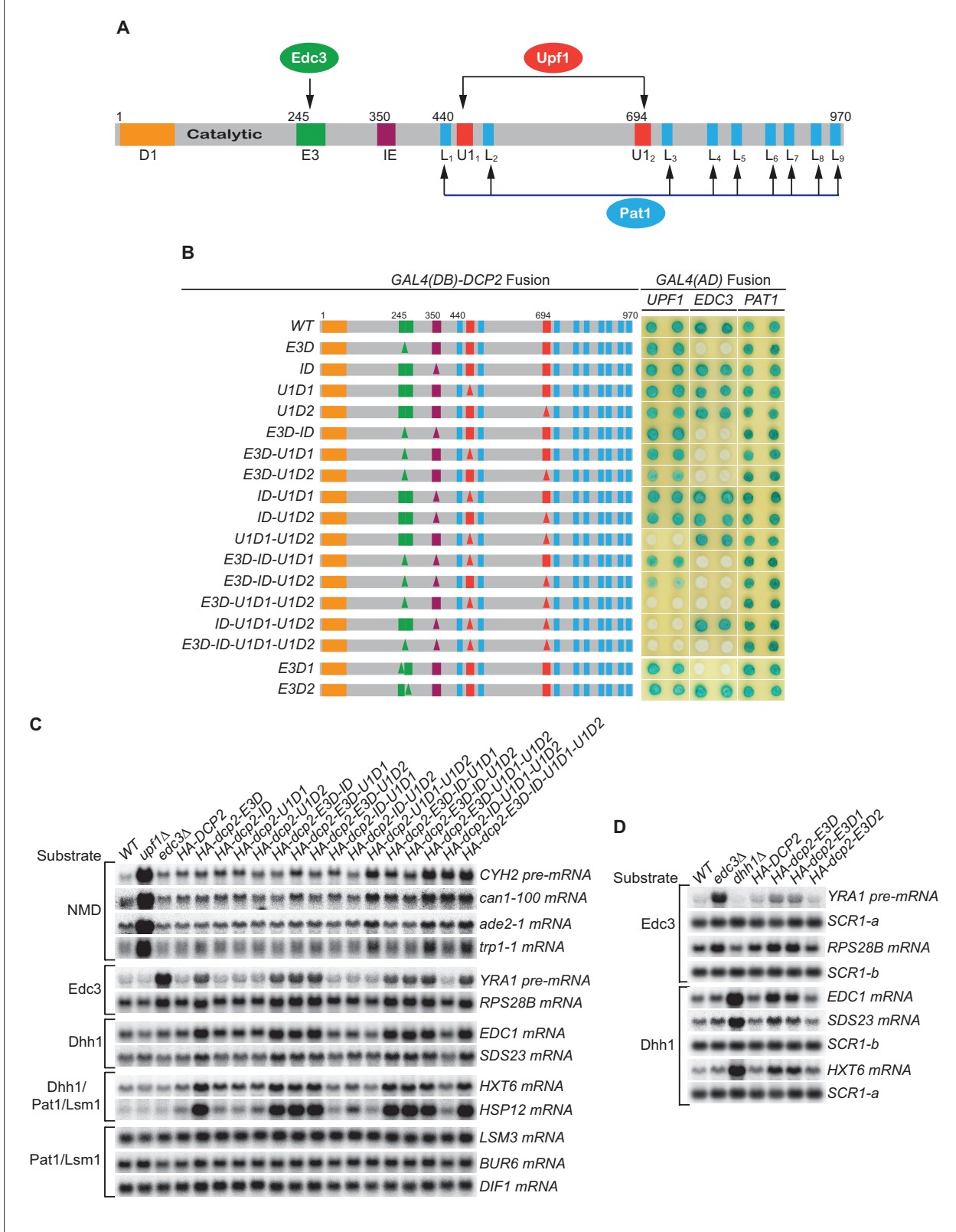

**Figure 1.** Consequences of deleting Dcp2's Edc3- and Upf1-binding motifs, and its inhibitory element. (**A**) Dcp2 schematic depicting its distinct regulatory elements. D1, Dcp1-binding site; E3, Edc3-binding motif; U1₁ and U1₂, Upf1-binding motifs; L₁–L₉, leucine-rich Pat1-binding motifs; IE, the inhibitory element. (**B**) Two-hybrid assays evaluating the consequences of specific *dcp2* deletions on Dcp2 interactions with Upf1, Edc3, and Pat1. Left, schematics of deletion alleles, with specific element deletions marked by triangles. The letter D in the allele names is used to indicate that a specific

*Figure 1 continued*

element has been deleted, such that *E3D* denotes deletion of the E3 element, *U1D1* denotes deletion of the U1$_1$ element, etc. Right, two-hybrid colony color assays, duplicated horizontally, with blue color indicating interaction and white indicating no interaction. (**C**) Northern analyses of individual transcript levels in cells expressing specific *dcp2* deletion alleles or harboring deletions of *UPF1* or *EDC3*. Groupings depict transcripts subject to common regulation. (**D**) Northern analyses of Edc3 and Dhh1 substrates in cells expressing *dcp2* alleles deleted for E3-1 or E3-2, or harboring deletions of *EDC3* or *DHH1*. Further analyses of the transcripts considered in (**C**) and (**D**) are presented in *Figure 1—figure supplement 1* and *Figure 1—figure supplement 2*. In all blots, lower case letters denote *SCR1* blots duplicated for clarity of presentation.

The online version of this article includes the following source data and figure supplement(s) for figure 1:

**Source data 1.** Northern analyses of Edc3 and Dhh1 substrates in cells harboring the E3-1 or E3-2 deletions of *DCP2* or deletions of *EDC3* or *DHH1* (*Figure 1D*).

**Figure supplement 1.** Loss of the Upf1- and Edc3-binding motifs, respectively, causes selective stabilization of nonsense-mediated mRNA decay (NMD) or Edc3/Dhh1 substrates.

**Figure supplement 1—source data 1.** Northern analyses of different decapping substrates in cells harboring specific *dcp2* deletions of the inhibitory element and the Edc3- or Upf1-binding motifs (*Figure 1B*).

**Figure supplement 2.** Loss of the Upf1- and Edc3-binding motifs, respectively, causes selective stabilization of nonsense-mediated mRNA decay (NMD) or Edc3/Dhh1 substrates.

These results indicate that the two Upf1-binding motifs have redundant functions and either can promote independent Upf1 binding to Dcp2.

WT yeast expressed low levels of NMD substrates, including the *CYH2* pre-mRNA and the *ade2-1*, *can1-100*, and *trp1-1* mRNAs. Loss of either the first or the second Upf1-binding motif had no significant effect on the levels of accumulation for each of these NMD substrates. However, loss of both Upf1-binding motifs caused approximately two- to threefold increases in the levels of these transcripts (*Figure 1C*, *Figure 1—figure supplement 1A, B*, and *Figure 1—figure supplement 2*, *HA-dcp2* alleles containing U1D1–U1D2). These increases were much smaller in magnitude than those caused by deletion of *UPF1*, which usually led to >10-fold increases for these transcripts. Loss of both Upf1-binding motifs did not affect the levels of the Edc3, Dhh1, Pat1/Lsm1, and Pat1/Lsm1/Dhh1 substrates (*Figure 1C*, *Figure 1—figure supplement 1A, B*, and *Figure 1—figure supplement 2*). This selective stabilization of nonsense-containing mRNAs indicates that the two Upf1-binding motifs control targeting of the decapping enzyme to NMD substrates. In addition, the two motifs have independent activities and are functionally redundant in promoting NMD. The partial stabilization of NMD substrates caused by loss of both Upf1-binding motifs suggests that decapping is not a major rate-limiting step in the overall NMD pathway. It also appears that NMD substrates can be degraded by an alternative route in the absence of active recruitment of the decapping enzyme by Upf1.

## Loss of the Edc3-binding motif eliminates Edc3 binding to Dcp2 and causes selective stabilization of both Edc3 and Dhh1 substrates

WT Dcp2 also exhibited strong two-hybrid interaction with Edc3. Loss of E3 (E3D) did not affect the binding of Upf1 or Pat1 to Dcp2 but eliminated Edc3 binding to Dcp2 (*Figure 1B*), indicating that the E3 motif promotes selective binding of Edc3 to the decapping enzyme. Loss of the E3 motif did not affect the levels of NMD and Pat1/Lsm1 substrates, but did lead to stabilization of the Edc3 substrates, *RPS28B* mRNA and *YRA1* pre-mRNA, to different extents (*Figure 1C*, *Figure 1—figure supplement 1A, B*, and *Figure 1—figure supplement 2*, *HA-dcp2-E3D*). Compared to their fold increases in *edc3Δ* cells, loss of the Edc3-binding motif caused complete stabilization of *RPS28B* mRNA (~twofold increases in both E3D and *edc3Δ* cells), but only partial stabilization of *YRA1* pre-mRNA (threefold increase in E3D vs. ninefold increase in *edc3Δ* cells). The different effects of loss of the E3 motif may be indicative of different roles of Edc3 in decapping of these transcripts. Decapping of *RPS28B* mRNA is likely rate limiting and totally dependent on Edc3-mediated recruitment of the decapping enzyme. The principal function of Edc3 in *RPS28B* mRNA decay would thus be to recruit the decapping enzyme and, when this recruitment is blocked by E3 deletion, *RPS28B* mRNA is not degraded efficiently by an alternative pathway. In contrast, Edc3-mediated recruitment of the decapping enzyme may contribute only partially to the overall decapping process of *YRA1* pre-mRNA and Edc3 may play an additional role in the decay of this transcript. Also, when Edc3-mediated recruitment of the decapping enzyme is

blocked by E3 deletion, *YRA1* pre-mRNA appears to be degraded by an alternative route, albeit less efficiently. Both possibilities were validated by experiments described below.

Although deletion of *EDC3* did not affect the levels of Dhh1 substrates, loss of the E3 motif caused partial stabilization of these mRNAs (*Figure 1C*, *Figure 1—figure supplement 2*, compare *edc3Δ* vs. *HA-dcp2-E3D*; *He et al., 2018*). This result indicates that, in addition to Edc3 substrates, E3 also controls targeting of the decapping enzyme to Dhh1-regulated mRNAs. Given the disparate mRNA decay phenotypes caused by *edc3Δ* and E3D for Dhh1 substrates, the E3-mediated targeting the decapping enzyme to Dhh1 substrates is unlikely to be carried out solely via interaction with Edc3. At least in the absence of Edc3, one additional factor may bind E3 (see below for Scd6 data) and target the decapping enzyme to Dhh1-regulated mRNAs. A role for E3 in the decapping of Dhh1 substrates may appear surprising, but physical interactions between Edc3 and Dhh1 have been reported (*He and Jacobson, 2015*; *Sharif et al., 2013*). The partial stabilization of Dhh1 substrates caused by loss of E3 can be explained similarly to that described above for *YRA1* pre-mRNA, that is, recruitment of the decapping enzyme makes a small kinetic contribution to overall decay and degradation of these substrates can occur by an alternative route.

The *dcp2 E3D* allele contains a deletion of a conserved 37-codon segment that we identified previously as encoding the Edc3-binding element (*He and Jacobson, 2015*). Phylogenetic sequence comparisons suggested that this segment may encode composite-binding elements (*Figure 1—figure supplement 1A*). To map this region further, we generated two smaller deletions in the originally defined element. E3D1 eliminates the first part of the element (17 amino acids, E3-1) and E3D2 eliminates the second part (20 amino acids, E3-2). Loss of E3-2 did not affect Edc3 binding to Dcp2, but loss of E3-1 eliminated Edc3's binding to Dcp2 (*Figure 1B*), indicating that Edc3 binds to the conserved 17-amino acid segment. Loss of E3-1 also caused complete stabilization of *RPS28B* mRNA and partial stabilization of *YRA1* pre-mRNA and Dhh1-regulated mRNAs (*Figure 1D*, *Figure 1—figure supplement 1C*), suggesting that the E3-1 element controls selective targeting of the decapping enzyme to both Edc3 and Dhh1 substrates.

## Loss of the Pat1-binding motifs eliminates Pat1 binding to Dcp2, but has no effect on levels of Pat1/Lsm1-regulated mRNAs

We next focused on the nine helical leucine-rich motifs $L_1$–$L_9$, each of which, except $L_8$, was shown to bind Pat1 in yeast two-hybrid and GST-pulldown assays (*Charenton et al., 2017*; *He and Jacobson, 2015*). However, whether these eight motifs are all engaged in Pat1 binding and whether they promote independent or collaborative binding of Pat1 to Dcp2 in the context of full-length protein or decapping complex is unknown. WT Dcp2 exhibited strong two-hybrid interaction with Pat1 and loss of the first five leucine-rich motifs ($L_1$–$L_5$) eliminated this interaction (*Figure 2A*, alleles LD1-5 to LD1-9). In contrast, loss of the last four leucine-rich motifs ($L_9$–$L_6$) did not affect Pat1 binding to Dcp2 (*Figure 2A*, allele L9-6). Consecutive deletions between $L_1$ and $L_5$ from either the N-terminus or the C-terminus yielded graded responses for Pat1 binding. From the N-terminus, loss of $L_4$ weakened and further loss of $L_5$ eliminated Pat1 binding (*Figure 2A*, alleles L1–4 and L1–5). From the C-terminus, loss of $L_4$ and $L_3$ greatly weakened and further loss of $L_2$ eliminated Pat1 binding (*Figure 2A*, alleles LD9-4, LD9-3, and LD9-2). None of the leucine-rich element deletions affected the binding of Edc3 or Upf1 to Dcp2 (*Figure 2A*). These results indicate that in the context of full-length Dcp2, leucine-rich motifs $L_1$–$L_5$ control the selective binding of Pat1 to Dcp2, most likely with a contribution from each motif. It is also possible that leucine-rich motifs $L_6$ to $L_9$ collaborate with $L_4$ and $L_5$ to promote the selective binding of Pat1 to Dcp2. Further, if Pat1 only uses its C-terminal binding domain to engage Dcp2, as demonstrated in a recent structural study (*Charenton et al., 2017*), then the multiple-motif requirement for Pat1's binding to Dcp2 may indicate that Pat1 binds to full-length Dcp2 using allovalency (*Klein et al., 2003*).

Consistent with the selective targeting of the decapping enzyme by both the Edc3- and Upf1-binding motifs, none of the leucine-rich element deletions affected the levels of the Edc3, Dhh1, or NMD substrates (*Figure 2B*, *Figure 2—figure supplement 1C*). Surprisingly, none of the leucine-rich element deletions altered the levels of Pat1-regulated mRNAs, including both Pat1/Lsm1 and Pat1/Lsm1/Dhh1 substrates (*Figure 2B, C*). This indicates that Pat1-mediated targeting of the decapping enzyme does not make a significant contribution to the overall decay of Pat1-regulated transcripts, raising a question about the role of Pat1 in decapping. Because loss of Pat1 causes significant

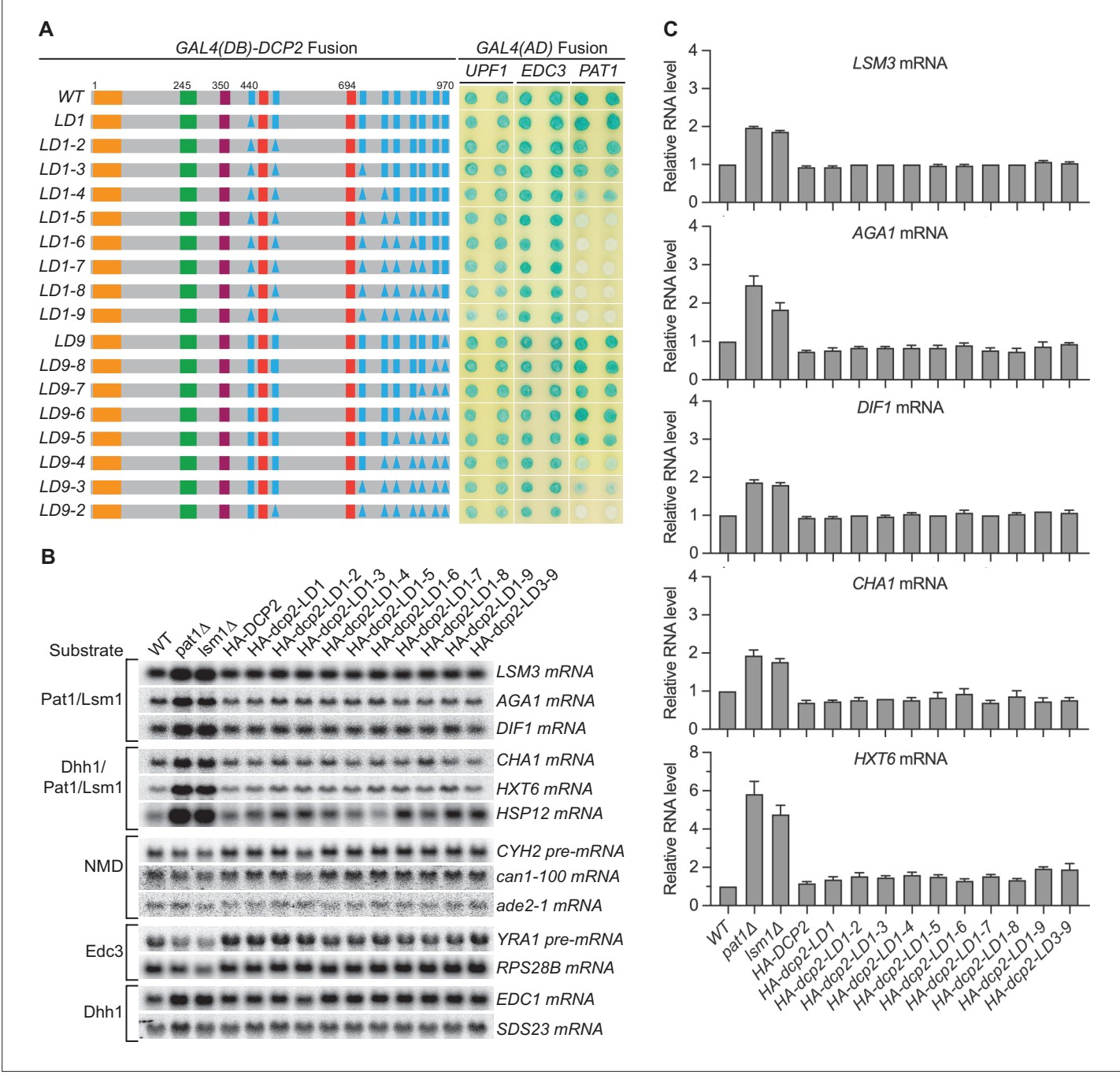

**Figure 2.** Consequences of deleting Dcp2 leucine-rich Pat1-binding motifs. (**A**) Two-hybrid assays evaluating the consequences of deleting the leucine-rich motifs ($L_1$–$L_9$) from the Dcp2 C-terminal domain. As in **Figure 1B**, schematics of the individual *dcp2* alleles are shown on the left (with specific element deletions denoted by triangles) and duplicate two-hybrid assays are on the right. (**B**) Northern analyses of individual Pat1/Lsm1 and Pat1/Lsm1/ Dhh1 substrate levels in cells harboring individual or combined deletions of Dcp2 leucine-rich motifs. (**C**) Bar graphs of average ± SEM for a subset of the northern analyses depicted in B. The relative levels of each decapping substrate in different strains were determined from three independent experiments, with one representative blot for each transcript shown in panel B. See also **Figure 2—figure supplement 1**.

The online version of this article includes the following source data and figure supplement(s) for figure 2:

**Figure supplement 1.** Loss of single, multiple, or even all nine leucine-rich motifs has no effect on decay of Pat1/Lsm1 and Pat1/Lsm1/Dhh1 substrates.

**Figure supplement 1—source data 1.** Northern analyses of different decapping substrate levels in cells harboring individual or combined deletions of Dcp2 leucine-rich motifs.

stabilization of both the Pat1/Lsm1 and Pat1/Lsm1/Dhh1 substrates (*Figure 2B*; *He et al., 2018*), one possible explanation for this surprising observation is that decapping is not rate limiting for Pat1-regulated mRNAs, and Pat1 performs an unidentified major function upstream, independent of its role in the recruitment and activation of the decapping enzyme (*Lobel et al., 2019*; *Nissan et al., 2010*).

## In the absence of active recruitment of the decapping enzyme, different decapping substrates are still degraded by decapping-dependent 5′ to 3′ decay

As described above, the Upf1-, Edc3-, and Pat1-binding motifs in Dcp2 promote selective binding of these factors to the decapping enzyme. Yet, loss of both Upf1-binding motifs in Dcp2 led to only partial stabilization of NMD substrates and loss of the Edc3-binding motif resulted in partial stabilization of Edc3 and Dhh1 substrates except for the *RPS28* mRNA. Further, loss of the Pat1-binding motifs did not cause significant stabilization of Pat1/Lsm1 or Pat1/Lsm1/Dhh1 substrates. The little or no effect of these deletions on different decapping substrates raised the question of decay mechanism for these mRNAs in the absence of active decapping enzyme recruitment. Two possibilities for their decay in the absence of active recruitment of the decapping enzyme seemed likely: degradation by a 3′ to 5′ exosome-dependent decay mechanism or decapping by an alternative route but overall degradation by decapping-dependent 5′ to 3′ decay. To distinguish between these possibilities, we generated double mutant cells that combine the *dcp2 cis* element deletions U1D1–U1D2, E3D or E3D1, and LD1-9 (*Figure 3A*) with deletions of key genes required for 5′ to 3′ decay (*XRN1*) or 3′ to 5′ decay (*SKI2* or *SKI7*) and analyzed the mRNA decay phenotypes of different decapping substrates in the resulting 'double' mutant cells (*Figure 3B, C*).

In *dcp2-U1D1-U1D2* cells, deletion of *SKI2* or *SKI7* did not cause additional stabilization of the NMD substrates (*CYH2* pre-mRNA and *can1-100* mRNA) whereas deletion of *XRN1* caused substantial stabilization of these transcripts (*Figure 3B*). Similarly, deletion of *SKI2* or *SKI7* did not cause additional stabilization of both Edc3 and Dhh1 substrates, for example, *YRA1* pre-mRNA, *EDC1*, and *SDS23* mRNAs, in *dcp2-E3D* or *E3D1* cells, but deletion of *XRN1* caused substantial stabilization of each of these transcripts (*Figure 3B*, *Figure 3—figure supplement 1*). Deletions of *SKI2* or *SKI7* also did not cause discernible stabilization of the Pat1/Lsm1 substrates, *BUR6*, *DIF1*, and *LSM3* mRNAs, in *dcp2-LD1-9* cells, but deletion of *XRN1* stabilized these mRNAs substantially (*Figure 3C*). In each case, deletion of *XRN1* in the respective *dcp2 cis* element mutant cells yielded similar fold increases in transcript levels as those caused by *XRN1* deletion in *DCP2* WT cells (*Figure 3B, C*). These results indicate that, in the absence of active recruitment of the decapping enzyme, the decapping substrates examined here, including the NMD, Edc3, Dhh1, and Pat1/Lsm1 substrates, are not degraded by exosome-dependent 3′ to 5′ decay, but are all still degraded by decapping-dependent 5′ to 3′ decay. Hence, these decapping substrates can all be decapped by an alternative route when the normal Dcp2 *cis*-element-mediated active recruitment of the decapping enzyme is blocked.

## In the absence of active Edc3 recruitment of the decapping enzyme, Edc3 and Dhh1 substrates are degraded by Pat1-mediated decay

The experiments of *Figure 3* demonstrated that different decapping substrates can be decapped by an alternative route when recruitment of the decapping enzyme to these mRNAs is blocked by specific *dcp2 cis*-element deletions. To determine whether the Edc3-, Upf1-, and Pat1-mediated decapping pathways have redundant activities or may function as backup systems for each other, we constructed a third set of *dcp2* alleles harboring different combinations of element deletions that eliminate the binding of Edc3 (E3D or E3D1), Upf1 (U1D1–U1D2), and Pat1 (LD1-8, LD1-9, or LD9-2) to Dcp2 (*Figure 4A* and *Figure 4—figure supplements 1A; 3A*). In addition, we also included two inconsequential Upf1-binding element deletions (U1D1 or U1D2) and one partially functional Pat1-binding element deletion (LD9-3) in this set.

Each of these alleles yielded the expected binding patterns of Edc3, Upf1, and Pat1 to Dcp2 except the four alleles that contain single Upf1-binding motif deletions (*Figure 4A*, alleles E3D-U1D1-LD1-9, E3D-U1D2-LD1-9, E3D1-U1D1-LD1-9, E3D1-U1D2-LD1-9). The mutant Dcp2 proteins generated from these four alleles showed no two-hybrid interaction with Upf1 (*Figure 4A*), but appeared fully functional in NMD (see below). As both Upf1-binding motifs border the leucine-rich motifs, we

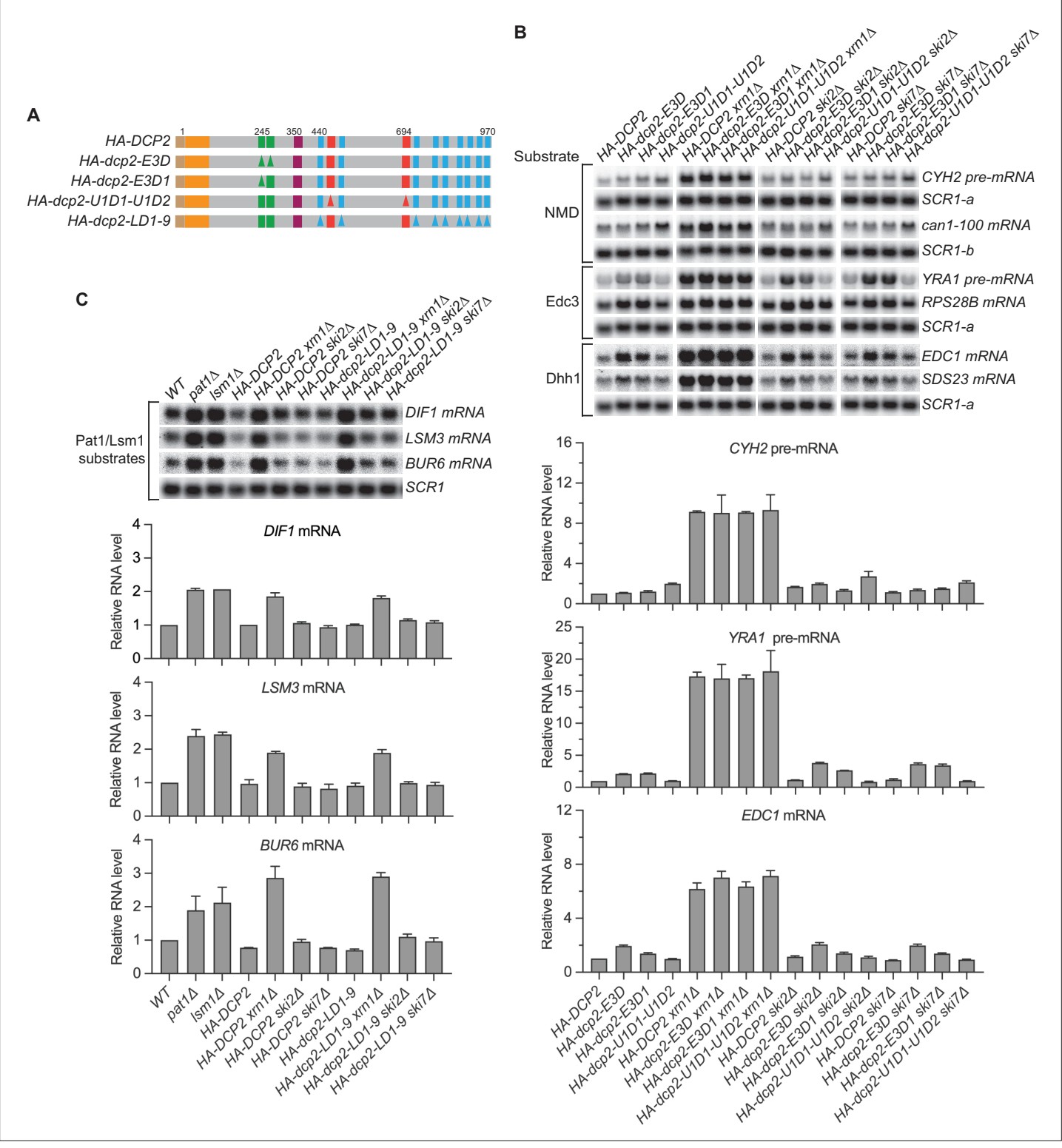

**Figure 3.** Nonsense-mediated mRNA decay (NMD), Edc3, Dhh1, and Pat1 substrates are still degraded by decapping-dependent 5′ to 3′ decay in the absence of active decapping enzyme recruitment. (**A**) Schematics of *dcp2* alleles that eliminate Edc3, Upf1, or Pat1 binding to Dcp2. (**B**) Deletion of *XRN1*, but not deletions of *SKI2* or *SKI7*, causes significant stabilization of NMD substrates in *HA-dcp2-U1D1-U1D2* cells, and Edc3 and Dhh1 substrates in *HA-dcp2-E3D* or *E3D1* cells. (**C**) Deletion of *XRN1*, but not deletions of *SKI2* or *SKI7*, causes significant stabilization of Pat1 substrates in *HA-dcp2-LD1-9* cells. Northern analyses in B and C as in *Figures 1 and 2*. Bar graphs in lower panels of B and C depict relative levels of decapping substrates in different strains determined from average ± SEM of three independent experiments. One representative northern blot for each transcript is shown

*Figure 3 continued on next page*

*Figure 3 continued*

in the upper panels. In the upper panel of B, lower case letters denote *SCR1* blots duplicated for clarity of presentation. See also *Figure 3—figure supplement 1*.

The online version of this article includes the following source data and figure supplement(s) for figure 3:

**Source data 1.** Northern analyses of different nonsense-mediated mRNA decay (NMD), Edc3, and Dhh1 substrates in cells harboring the *HA-dcp2-U1D1-U1D2*, *E3D*, or *E3D1* alleles and deletions of *XRN1*, *SKI2*, or *SKI7* (*Figure 3B*).

**Source data 2.** Northern analyses of different Pat1 substrates in cells harboring the *HA-dcp2-LD1-9* allele and deletions of *XRN1*, *SKI2*, or *SKI7* (*Figure 3C*).

**Figure supplement 1.** Nonsense-mediated mRNA decay (NMD), Edc3, and Dhh1 substrates are still degraded by decapping-dependent 5′ to 3′ decay in the absence of active recruitment of the decapping enzyme.

suspect that the LD1-9 deletion may indirectly affect the conformation of these two binding motifs and diminish Upf1 binding to Dcp2 in the two-hybrid assay.

Although none of the leucine-rich element deletions affected the decay of Edc3 and Dhh1 substrates (*Figure 2B*), combining the leucine-rich element deletions LD1-8, LD1-9, LD9-3, and LD9-2 with the Edc3-binding element deletions E3D or E3D1 caused additional substantial stabilization of the Edc3 substrate *YRA1* pre-mRNA and the Dhh1 substrates *EDC1*, *SDS23*, and *HXT6* mRNAs (*Figure 4B*, *Figure 4—figure supplement 2*). None of the combinations of these element deletions affected the Pat1/Lsm1 substrates or the NMD substrates (*Figure 4B*, *Figure 4—figure supplement 2*). These results indicate that in the absence of active recruitment of the decapping enzyme by Edc3, both Edc3 and Dhh1 substrates are degraded by the Pat1-mediated decay pathway, suggesting that Pat1-mediated decapping serves as a backup or fail-safe system in the decay of Edc3 and Dhh1-regulated mRNAs. We also noticed a subtle functional difference for these *dcp2* alleles. In otherwise the same deletion context, cells harboring E3D1 consistently had lower transcript levels than those harboring E3D for Edc3 and Dhh1 substrates (*Figure 4B*). These differences could be indicative of E3-2 element function in decay of these mRNAs, as E3D eliminates both E3-1 and E3-2 elements and E3D1 eliminates only the E3-1 element. Similarly, cells harboring LD9-3 also had consistently lower transcript levels than those harboring LD9-2 for both Edc3 and Dhh1 substrates, suggesting that LD9-3 deletion maintains more function of Dcp2 than that of LD9-2 in decay of these mRNAs. Consistent with this interpretation, LD9-3 weakens but LD9-2 eliminates Pat1 binding to Dcp2 (*Figure 4A*, compare alleles E3D-LD9-2 to E3D-LD9-3, and E3D1-LD9-2 to E3D1-LD9-3).

Combining the leucine-rich element deletion LD1-9 with the Upf1-binding element deletion U1D1–U1D2 did not cause additional stabilization of the NMD substrates *CYH2* pre-mRNA, *can1-100*, and *ade2-1* mRNAs (*Figure 4C*, *Figure 4—figure supplement 4*, allele U1D1-U1D2-LD1-9 compared to allele U1D1-U1D2 in *Figure 1B*). Further combining the Edc3-binding motif deletions E3D or E3D1 also did not cause additional stabilization of the *CYH2* pre-mRNA and *can1-100* mRNA, but did appear to cause small but significant increases in *ade2-1* mRNA levels (*Figure 4C*, *Figure 4—figure supplement 4*, compare allele U1D1-U1D2-LD1-9 to alleles E3D-U1D1-U1D2-LD1-9 and E3D1-U1D1-U1D2-LD1-9). These results indicate that Pat1-mediated decapping either does not function as a backup system or does not make a significant contribution to the decay of NMD substrates in the absence of Upf1-mediated recruitment of the decapping enzyme. However, Edc3-mediated decapping may function as a backup system for some nonsense-containing mRNAs. Further, the combination of element deletions E3D-U1D1-U1D2-LD1-9 eliminates all the binding motifs of known decapping activators in Dcp2, yet this combination of deletions still did not have any significant effect on the levels of accumulation of the Pat1/Lsm1 substrates (*Figure 4C*, *Figure 4—figure supplement 4*). This result indicates that neither Edc3-mediated decapping nor Upf1-mediated decapping functions as a backup system in decay of Pat1/Lsm1 substrates, raising the possibility that Pat1/Lsm1 substrates can be decapped without the enhancement function of any decapping activators.

The combination of element deletions U1D1-U1D2-LD1-9 created a *dcp2* allele harboring a lonely Edc3-binding motif. This *dcp2* allele had the activity of WT *DCP2* in promoting the decay of both Edc3 and Dhh1 substrates (*Figure 4C*, *Figure 4—figure supplement 4*, compare U1D1-U1D2-LD1-9 to WT). Similarly, the combination of deletions E3D-U1D2-LD1-9 and E3D1-U1D2-LD1-9 created two *dcp2* alleles harboring a lonely Upf1-binding motif $U1_1$, and the combination of deletions E3D-U1D1-LD1-9 and E3D1-U1D1-LD1-9 created two *dcp2* alleles harboring a lonely Upf1-binding motif $U1_2$.

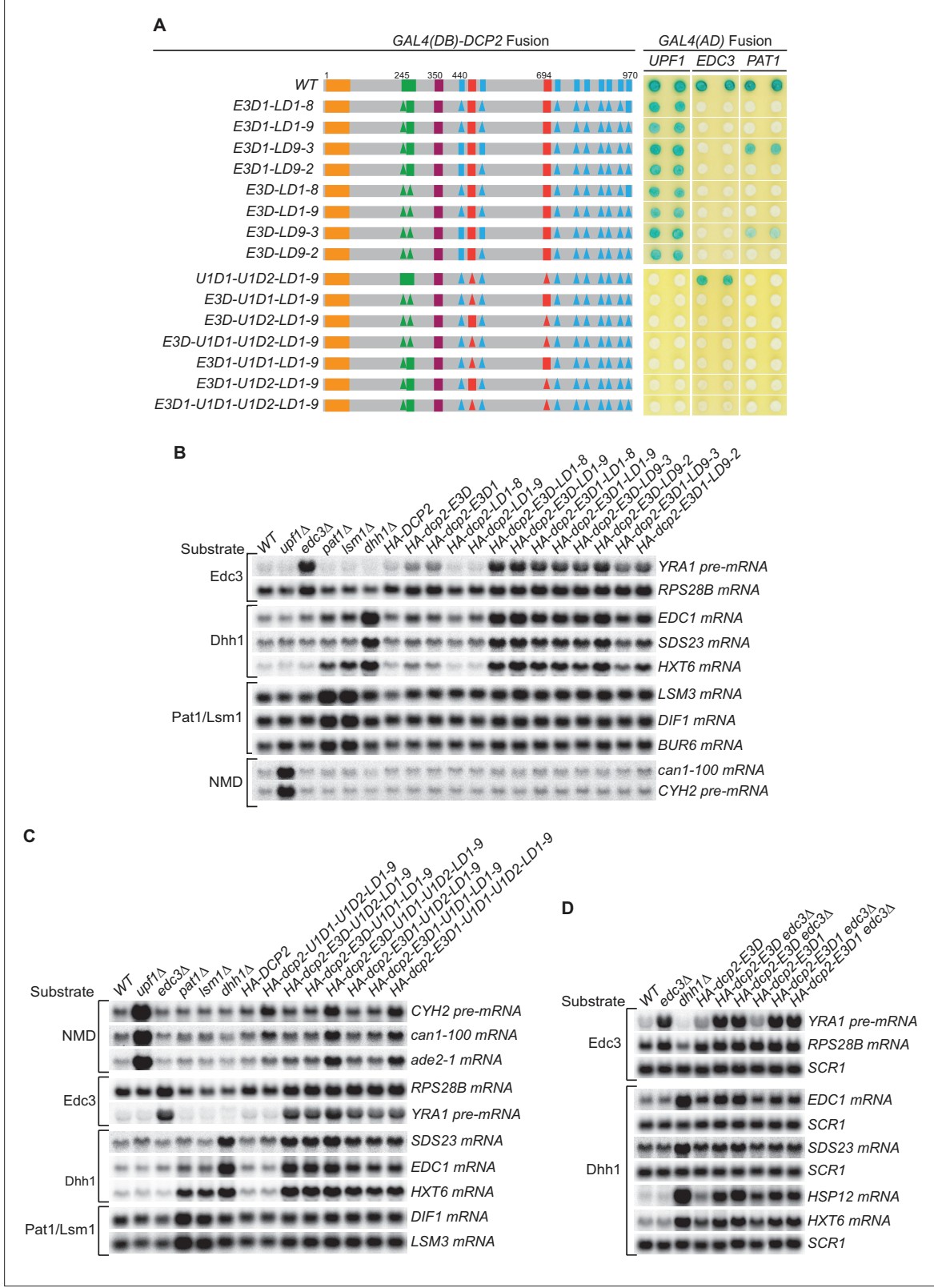

**Figure 4.** Genetic interactions between the Dcp2 Edc3-, Upf1-, and Pat1-binding motifs, or between the Edc3-binding motif and Edc3, that affect mRNA decapping. (**A**) Two-hybrid assays examining the effects of different combinations of element deletions on Edc3, Upf1, and Pat1 binding to Dcp2. Allele schematics and two hybrid analyses are as in *Figure 1B*. (**B**) Northern analyses of the consequences of simultaneous loss of the Dcp2 Edc3-binding motif and leucine-rich motifs. See also *Figure 4—figure supplement 1* and *Figure 4—figure supplement 2*. (**C**) Northern analyses of

*Figure 4 continued on next page*

*Figure 4 continued*

the consequences of simultaneous loss of the Dcp2 Upf1-binding motifs and leucine-rich motifs, and the roles of single Dcp2 Edc3 or Upf1-binding motifs. See also *Figure 4—figure supplement 3* and *Figure 4—figure supplement 4*. (**D**) Northern analyses of the consequences for Edc3 and Dhh1 substrates caused by loss of the Dcp2 Edc3-binding motif and *trans* deletion of *EDC3*. See also *Figure 4—figure supplement 5*. Northern analyses as in *Figures 1 and 2*. Two-hybrid analyses as in *Figure 1*.

The online version of this article includes the following source data and figure supplement(s) for figure 4:

**Source data 1.** Northern analyses of the consequences for Edc3 and Dhh1 substrates caused by loss of the Dcp2 Edc3-binding motif and *trans* deletion of *EDC3* (*Figure 4D*).

**Figure supplement 1.** Simultaneous loss of the Edc3-binding motif and the leucine-rich motifs has synergistic effects and causes substantial stabilization of both Edc3 and Dhh1 substrates.

**Figure supplement 1—source data 1.** Northern analyses of the consequences of simultaneous loss of the Dcp2 Edc3-binding motif and leucine-rich motifs (*Figure 4B*).

**Figure supplement 2.** Simultaneous loss of the Edc3-binding motif and the leucine-rich motifs has synergistic effects and causes substantial stabilization of both Edc3 and Dhh1 substrates.

**Figure supplement 3.** Simultaneous loss of the Upf1-binding motifs and the leucine-rich motifs has no synergistic effects on mRNA decapping, and a single Edc3- or Upf1-binding motif alone can promote efficient decapping of Edc3 or nonsense-mediated mRNA decay (NMD) substrates.

**Figure supplement 3—source data 1.** Northern analyses of the consequences of simultaneous loss of the Dcp2 Upf1-binding motifs and leucine-rich motifs, and the roles of single Dcp2 Edc3- or Upf1-binding motifs (*Figure 4C*).

**Figure supplement 4.** Simultaneous loss of the Upf1-binding motifs and the leucine-rich motifs has no synergistic effects on mRNA decapping, and a single Edc3- or Upf1-binding motif alone can promote efficient decapping of Edc3 or nonsense-mediated mRNA decay (NMD) substrates.

**Figure supplement 5.** Loss of the Edc3-binding motif and *trans* deletion of *EDC3* have additive effects on decapping of both Edc3 and Dhh1 substrates.

Each of these four *dcp2* alleles had the activity of WT *DCP2* in promoting NMD (*Figure 4C*, compare alleles E3D-U1D2-LD1-9, E3D1-U1D2-LD1-9, E3D-U1D1-LD1-9, and E3D1-U1D1-LD1-9 to WT). These results indicate that the Edc3-binding motif and each of the Upf1-binding motifs can function independently of other elements and promote specific mRNA decay activities.

## Edc3 carries out one additional function upstream of recruitment of the decapping enzyme in decay of both Edc3 and Dhh1 substrates

Except for *RPS28B* mRNA, loss of the Dcp2 Edc3-binding motif, both Upf1-binding motifs, or the Pat1-binding motifs caused only partial stabilization, or no stabilization, of Edc3, NMD, and Pat1/Lsm1 substrates. In contrast, loss of the corresponding specific binding factors Edc3, Upf1, or Pat1 all resulted in substantial stabilization of their targeted mRNAs (see above). These results strongly suggest that Edc3-, Upf1-, or Pat1-mediated recruitment of the decapping enzyme is not rate limiting to the overall decay process and each of these decapping activators most likely carries out an additional major function upstream of the recruitment of the decapping enzyme in decay of the respective targeted mRNAs. To test this idea further, we constructed double mutant strains containing *dcp2 cis* deletions E3D or E3D1 and deletion of *EDC3* and analyzed the decay phenotypes of Edc3 and Dhh1 substrates in the resulting strains. Because E3D and E3D1 both eliminate Edc3 binding to Dcp2, we reasoned that any additional stabilization caused by *EDC3* deletion of specific mRNAs in E3D or E3D1 cells is likely due to loss of an extra function of Edc3.

Deletion of *EDC3* caused additional seven- to eightfold stabilization of the Edc3 substrate *YRA1* pre-mRNA in E3D or E3D1 cells (*Figure 4D*, *Figure 4—figure supplement 5*). Deletion of *EDC3* also caused additional stabilization of the Dhh1 substrates *EDC1* and *SDS23* mRNAs and the Pat1/Lsm1/Dhh1 substrates *HSP12* and *HXT6* mRNAs (*Figure 4D*, *Figure 4—figure supplement 5*). These results provide evidence that Edc3 is directly involved in the decay of Dhh1-regulated mRNAs, and demonstrate that besides recruiting the decapping enzyme, Edc3 indeed carries out an additional function in decay of both Edc3 and Dhh1 substrates.

## Dcp2 *cis*-binding elements promote the assembly of distinct decapping complexes in yeast cells

The experiments described above indicate that the Dcp2 C-terminal *cis*-binding elements promote independent binding of Edc3, Upf1, and Pat1 to Dcp2 and control the selective targeting of the

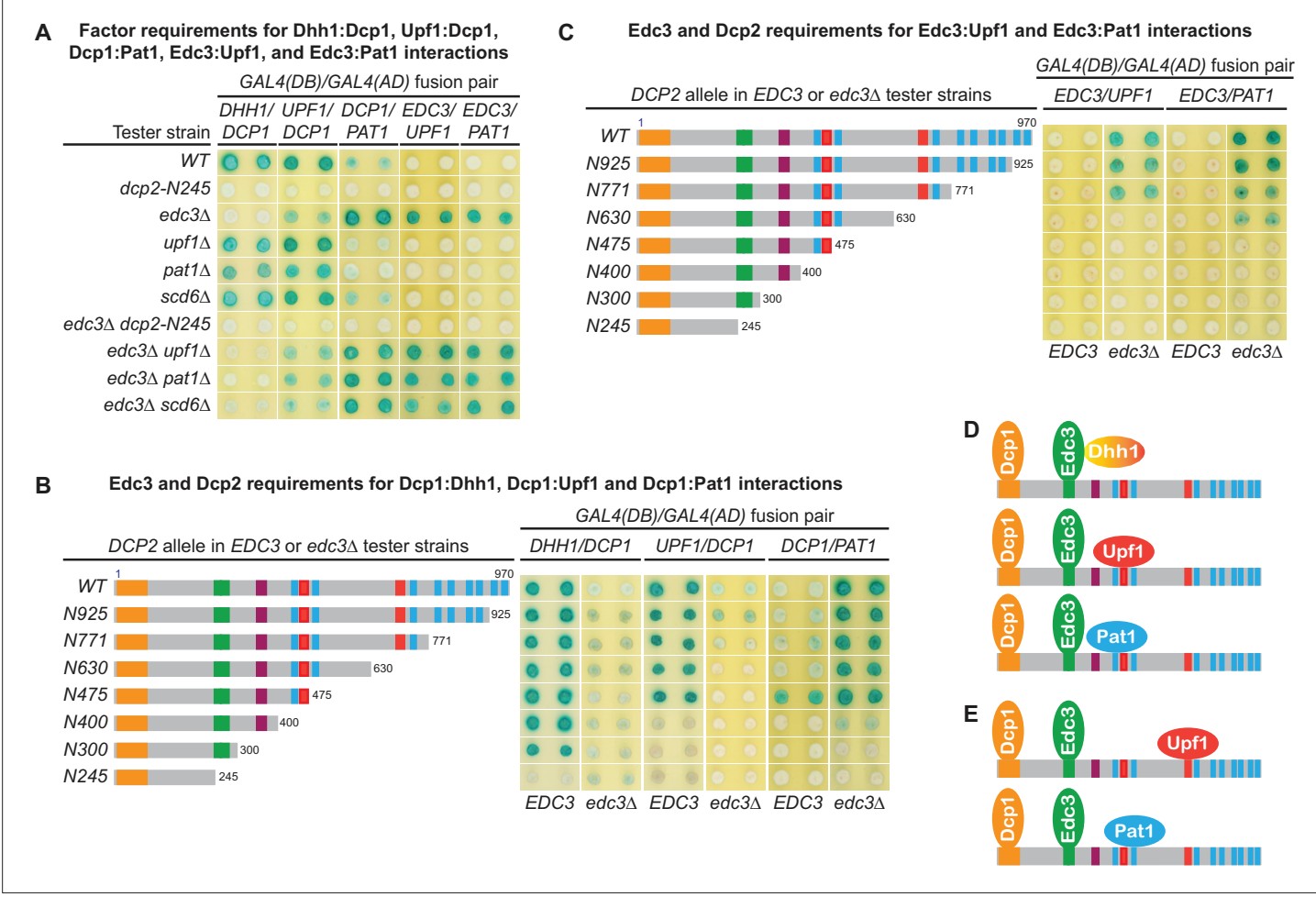

**Figure 5.** The Dcp1–Dcp2 decapping enzyme forms distinct decapping complexes in vivo with the decapping activators Edc3, Dhh1, Upf1, and Pat1. Two-hybrid assays, as in *Figure 1*, were used to dissect the molecular basis of five Dcp2-bridged interactions between Dcp1 and Dhh1, Upf1, or Pat1, and between Edc3 and Upf1 or Pat1. (**A**) Dcp1:Dhh1, Dcp1:Upf1, Dcp1:Pat1, Edc3:Upf1, and Edc3:Pat1 interactions in tester strains harboring different gene deletions or truncations of decapping factors. (**B**) Dcp1:Dhh1, Dcp1:Upf1, and Dcp1:Pat1 interactions in tester strains harboring Dcp2 C-terminal truncations in *EDC3* and *edc3Δ* cells. (**C**) Edc3:Upf1 and Edc3:Pat1 interactions in tester strains harboring Dcp2 C-terminal truncations in *EDC3* and *edc3Δ* cells. (**D**) Three different decapping complexes inferred from two-hybrid analyses in A and B. (**E**) Two different decapping complexes inferred from two-hybrid analyses in A and C.

decapping enzyme to different decapping substrates. These results and our earlier observation that different decapping activators each target a specific subset of yeast mRNAs (*He et al., 2018*) strongly suggest that different Dcp2 *cis*-binding elements promote the assembly of distinct decapping complexes. To test this idea, we explored two-hybrid interactions that are bridged through Dcp2 between Dcp1 and Edc3, Upf1, Pat1, or Dhh1 (*Figure 5A*; *He and Jacobson, 2015*) and dissected the molecular basis for each of these bridged interactions. We constructed two sets of yeast two-hybrid tester strains and analyzed each of the bridged interactions in these strains. One set of strains contains single deletions of *EDC3*, *UPF1*, *PAT1*, and *SCD6* or a previously described *dcp2-N245* truncation that eliminates the entire Dcp2 C-terminal domain (*He and Jacobson, 2015*). This set of strains also contains double mutants that combine *EDC3* deletion either with deletions of *UPF1*, *PAT1*, and *SCD6* or with the *dcp2-N245* truncation. The second set of strains contains different C-terminal truncations of Dcp2 in both *EDC3* and *edc3Δ* backgrounds. Each of these Dcp2 truncations eliminates a distinct set of its *cis*-binding elements. The first set of tester strains was designed to assess the factor requirements and the second set of tester strains was designed to evaluate the Dcp2 *cis*-binding element dependence for each of the observed interactions.

As shown in *Figure 5A*, Dhh1 exhibited strong two-hybrid interaction with Dcp1 in WT cells. Deletions of *UPF1*, *PAT1*, or *SCD6* did not affect the Dhh1:Dcp1 interaction. In contrast, loss of the entire C-terminal domain of Dcp2 or deletion of *EDC3* eliminated Dhh1:Dcp1 interaction. These results indicate that the Dhh1:Dcp1 interaction observed in WT cells requires both the Dcp2 C-terminal domain and Edc3, suggesting that this interaction occurs in a Dcp1–Dcp2–Edc3–Dhh1 complex. Further, in a WT *EDC3* background, Dcp2 C-terminal truncations maintaining the E3 motif left the Dhh1:Dcp1 interaction intact. In contrast, the Dcp2 C-terminal truncation eliminating the E3 motif abolished the Dhh1:Dcp1 interaction (*Figure 5B*). These results show that the observed Dhh1:Dcp1 interaction also requires an intact Dcp2 E3 motif, indicating that the Dcp2 Edc3-binding motif promotes assembly of the Dcp1–Dcp2–Edc3–Dhh1 decapping complex in vivo (*Figure 5D*). In an *edc3Δ* background, Dhh1 exhibited no interaction with Dcp1 (the slight blue color of the transformants was due to self-activation of Gal4(DB)-Dhh1 fusion protein) (*Figure 5B*; *He and Jacobson, 2015*), indicating that Dhh1 assembles into the Dcp1–Dcp2–Edc3–Dhh1 via association with Edc3, consistent with the physical interaction between these two factors (*He and Jacobson, 2015*; *Sharif et al., 2013*).

In WT cells, Upf1 also exhibited strong two-hybrid interaction with Dcp1 (*Figure 5A*). Deletions of *UPF1*, *PAT1*, or *SCD6* did not affect Upf1:Dcp1 interaction. In contrast, deletion of *EDC3* diminished and loss of the entire C-terminal domain of Dcp2 eliminated Upf1:Dcp1 interaction. These results indicate that Upf1:Dcp1 interaction in WT cells requires the C-terminal domain of Dcp2 and is enhanced by the presence of Edc3, suggesting that this interaction occurs in a Dcp1–Dcp2–Edc3–Upf1 complex. In a WT *EDC3* background, Dcp2 C-terminal truncations eliminating the second Upf1-binding motif $U1_2$ did not affect Upf1:Dcp1 interaction (*Figure 5B*, alleles N630 and N475); however, further eliminating the first Upf1-binding motif $U1_1$ abolished Upf1:Dcp1 interaction (*Figure 5B*, alleles N400, N300, and N245). In an *edc3Δ* background, Upf1 interacted weakly with Dcp1 and Dcp2 C-terminal truncations eliminating the second Upf1-binding motif $U1_2$ abolished Upf1:Dcp1 interaction (*Figure 5B*, compare alleles N771 to N630). These results indicate that the Upf1-binding motifs $U1_1$ and $U1_2$ can both promote the assembly of Upf1-containing decapping complexes and that Edc3 enhances the binding of Upf1 to the $U1_1$ motif to promote assembly of the Dcp1–Dcp2–Edc3–Upf1 decapping complex in vivo (*Figure 5D*).

In contrast to Dhh1 and Upf1, Pat1 exhibited only weak two-hybrid interaction with Dcp1 in WT cells (*Figure 5A*). Deletions of *UPF1*, *PAT1*, or *SCD6* did not alter Pat1:Dcp1 interaction, but loss of the entire Dcp2 C-terminal domain eliminated Pat1:Dcp1 interaction (*Figure 5A*). Interestingly, deletion of *EDC3* enhanced Pat1:Dcp1 interaction (*Figure 5A*). These results indicate that Pat1 can associate with the Dcp1–Dcp2 decapping enzyme both in the presence and absence of Edc3. Eliminating different leucine-rich motifs between $L_2$ and $L_9$ by progressive Dcp2 C-terminal truncations did not significantly alter the respective Pat1:Dcp1 interaction strength in *EDC3* and *edc3Δ* backgrounds. However, further eliminating the leucine-rich motif $L_1$ abolished Pat1:Dcp1 interaction in an *EDC3* background and greatly diminished Pat1:Dcp1 interaction in an *edc3Δ* background (*Figure 5B*). These results show that Pat1:Dcp1 interaction requires the $L_1$ motif in Dcp2 regardless of the cellular status of Edc3, suggesting that this motif may promote assembly of several different decapping complexes including a Dcp1–Dcp2–Edc3–Pat1 complex (*Figure 5D*).

## Edc3 is a common component of multiple decapping complexes

To further validate our conclusion that Dcp2 *cis*-binding elements promote assembly of distinct decapping complexes and to explore the assembly dynamics of these complexes, we examined whether Dcp2 bridges interactions between Edc3 and Upf1 or Pat1. As shown in *Figure 5A*, in contrast to the Upf1:Dcp1 interaction, Upf1 did not show two-hybrid interaction with Edc3 in WT cells. Upf1 also did not show interaction with Edc3 in *upf1Δ*, *pat1Δ*, or *scd6Δ* cells. Surprisingly, Upf1 exhibited strong two-hybrid interaction with Edc3 in *edc3Δ* cells. Deletions of *UPF1*, *PAT1*, or *SCD6* in *edc3Δ* cells did not alter this strong Upf1:Edc3 interaction; however, deletion of the entire Dcp2 C-terminal domain in *edc3Δ* cells eliminated Upf1:Edc3 interaction. These results indicate that Dcp2 bridges an interaction between Upf1 and Edc3 and that this bridged Upf1:Edc3 interaction is inhibited by endogenous Edc3. In an *edc3Δ* background, eliminating the second Upf1-binding motif $U1_2$ in Dcp2 by C-terminal truncation abolished the Upf1:Edc3 interaction (*Figure 5C*, compare alleles N771 and N630). This result shows that the Upf1:Edc3 interaction requires the $U1_2$ motif in Dcp2 and that this motif can promote

assembly of a Dcp1–Dcp2–Edc3–Upf1 decapping complex in vivo, with Upf1 binding to the Dcp2 U1$_2$ motif (*Figure 5E*).

Similar to the Upf1:Edc3 interaction, Pat1 also did not show two-hybrid interaction with Edc3 in WT cells, deletion of *EDC3* promoted Pat1:Edc3 interaction, and deletion of the entire C-terminal domain of Dcp2 eliminated Pat1:Edc3 interaction in *edc3Δ* cells (*Figure 5A*). These results indicate that Dcp2 also bridges an interaction between Pat1 and Edc3 and that this bridged interaction is inhibited by endogenous Edc3. In an *edc3Δ* background, eliminating different leucine-rich motifs between L$_3$ and L$_9$ did not affect Pat1:Edc3 interaction; however, further eliminating the L$_2$ motif abolished Pat1:Edc3 interaction (*Figure 5C*). This result shows that Pat1:Edc3 interaction requires the L$_2$ motif in Dcp2 and that this motif can promote assembly of a Dcp1–Dcp2–Edc3–Pat1 decapping complex in vivo, with Pat1 binding to the Dcp2 L$_2$ motif (*Figure 5E*).

Our two-hybrid analyses of the bridged molecular interactions between Upf1 and Dcp1 or Edc3 revealed two distinct Upf1-containing decapping complexes with the same composition but different configurations. Similarly, analyses of the bridged molecular interactions between Pat1 and Dcp1 or Edc3 also revealed two distinct Pat1-containing decapping complexes of the same composition but different configurations (*Figure 5D, E*). These complexes share Dcp1, Dcp2, and Edc3 with Upf1 binding to either U1$_1$ or U1$_2$, or with Pat1 binding to either L$_1$ or L$_2$ in Dcp2. We suspect that the binding site preference for Upf1 and Pat1 in these decapping complexes is likely caused by the constraint of configurations of the Gal4 DNA-binding and activation domains fused to the respective binding partners. Nevertheless, these results indicate that Upf1 and Pat1 can both bind to two different binding motifs in Dcp2, raising the possibility that different Upf1- and Pat1-binding motifs in Dcp2 may control structural transitions of specific decapping complexes. It is also possible that the binding site preference for Upf1 and Pat1 is controlled by the number of Edc3 molecules (i.e., monomer or dimer) in the decapping complexes.

Our observation that endogenous Edc3 can completely inhibit the bridged molecular interactions between Edc3 and Upf1 or Pat1 is intriguing (*Figure 5A*). It indicates that free Edc3-binding sites from Dcp2 molecules are limiting and suggests that the entire pool of Dcp2 molecules are likely to be stably bound by endogenous Edc3. Consistent with this interpretation, as demonstrated above, Edc3 exists as a common component of multiple decapping complexes.

## Xrn1 binds to an internal fragment of Dcp2 and is recruited to the decapping complex by Dcp2

To assess potential coupling between decapping and 5′ to 3′ exoribonucleolytic decay, we tested two hybrid interactions between Xrn1 and all known decapping factors. Xrn1 interacted with Dcp1, Edc3, Pat1, and Upf1 (*Figure 6A*). Xrn1 did not interact with full-length Dcp2, but interacted with a 726-amino acid C-terminal Dcp2 fragment Dcp2-ND244 (*Figure 6A*), suggesting that Xrn1 binding to Dcp2 may be dependent on a specific Dcp2 conformation. Deletions of *EDC3*, *UPF1*, or *PAT1* and elimination of the entire C-terminal domain of Dcp2 did not affect binding of Xrn1 to Dcp2-ND244 or Pat1 (*Figure 6C*), suggesting that Xrn1 may bind to Dcp2 and Pat1 directly. To identify the Xrn1-binding region of Dcp2, we analyzed Xrn1 interactions with a panel of Dcp2 fragments (*Figure 6—figure supplement 1A*). In contrast to the binding of Edc3, Upf1, and Pat1 to Dcp2, Xrn1 binding to Dcp2 required a large Dcp2 fragment with the minimal Xrn1-binding region encompassing an internal fragment from aa 316–575 (*Figure 6D*, *Figure 6—figure supplement 1B*).

In contrast to the Xrn1:Dcp2-ND244 and Xrn1:Pat1 interactions, the Xrn1:Dcp1, Xrn1:Edc3, and Xrn1:Upf1 interactions are all bridged by Dcp2. As shown in *Figure 6B*, deletions of *UPF1*, *PAT1*, or *SCD6* did not significantly affect Xrn1:Dcp1 interaction. However, elimination of the Dcp2 C-terminal domain or deletion of *EDC3* abolished Xrn1:Dcp1 interaction, suggesting that Xrn1:Dcp1 interaction is bridged by Dcp2 and enhanced by Edc3, and occurs in a Dcp1–Dcp2–Edc3–Xrn1 complex (*Figure 6F*). In support of the latter conclusion, Xrn1:Dcp1 interaction is also dependent on an intact Xrn1-binding site in Dcp2. In an *EDC3* background, loss of partial Xrn1-binding region by Dcp2 C-terminal truncation abolished Xrn1:Dcp1 interaction (*Figure 6E*, compare alleles N630 and N475). Deletions of *EDC3*, *UPF1*, *PAT1*, or *SCD6* did not affect Xrn1:Edc3 interaction, but elimination of the entire Dcp2 C-terminal domain abolished Xrn1:Edc3 interaction (*Figure 6B*), indicating that this interaction is bridged by Dcp2. As additional support for this conclusion, in both *EDC3* and *edc3Δ* backgrounds, Xrn1:Edc3 interaction requires an intact Xrn1-binding site in Dcp2, as loss of partial Xrn1-binding

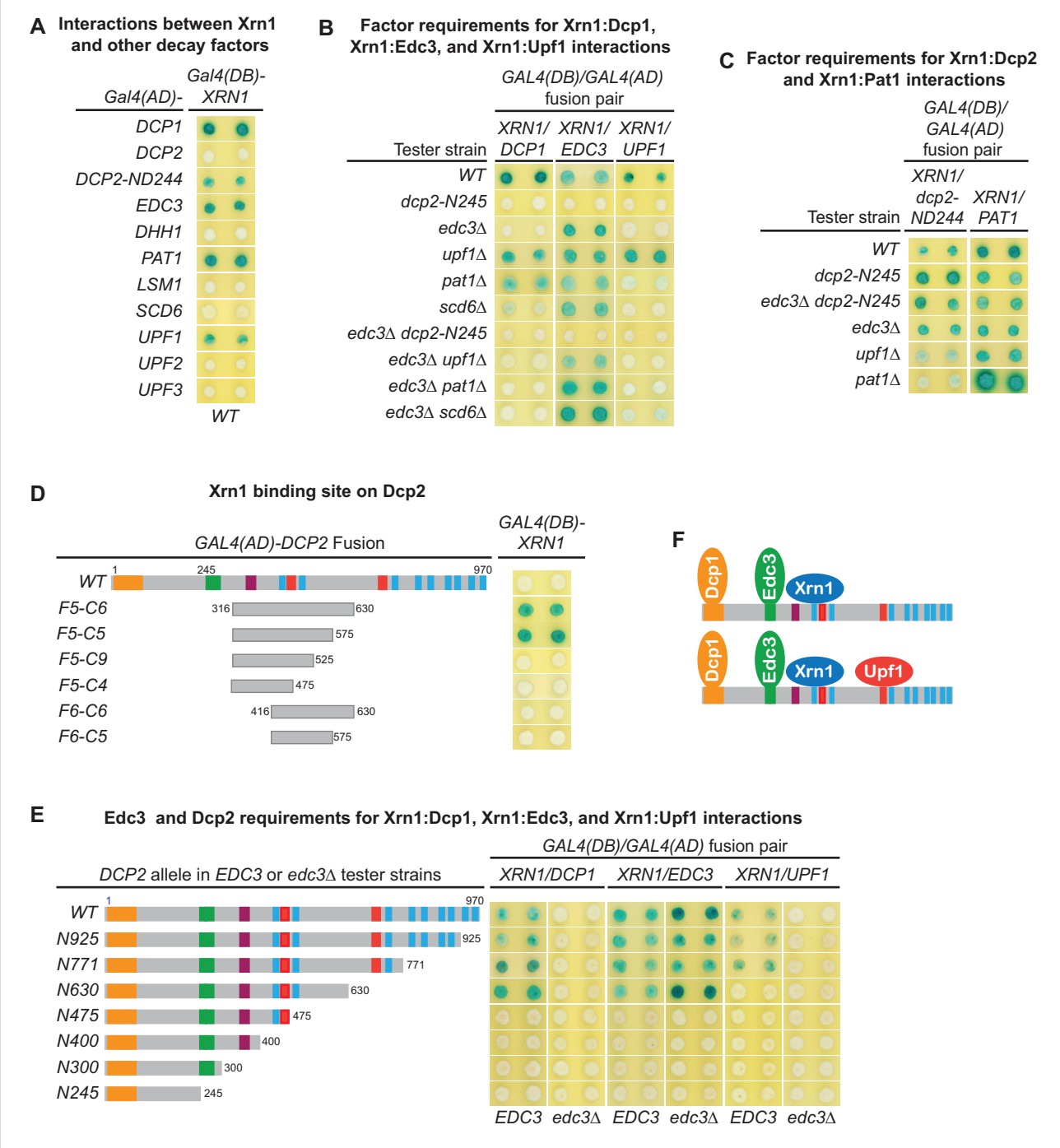

**Figure 6.** Xrn1 binds to Dcp2 and is directly recruited to different decapping complexes by Dcp2. (**A**) Two-hybrid analyses of Xrn1 interactions with Dcp1, Edc3, Pat1, Upf1, and an N-terminally truncated Dcp2 fragment in WT cells. (**B**) Two-hybrid analyses of Dcp2-bridged interactions between Xrn1 and Dcp1, Edc3, or Upf1. (**C**) Two-hybrid analyses of interactions between Xrn1 and Pat1 or N-terminally truncated Dcp2. (**D**) Two-hybrid analyses of Xrn1 binding to an internal Dcp2 fragment. See also *Figure 6—figure supplement 1*. (**E**) Dcp1:Xrn1, Edc3:Xrn1, and Upf1:Xrn1 interactions in tester strains harboring Dcp2 C-terminal truncations and *EDC3* or *edc3Δ* alleles. (**F**) Two different Xrn1-containing decapping complexes inferred from two-hybrid analyses in B and E. Allele schematics and two-hybrid analyses as in *Figure 1*.

The online version of this article includes the following figure supplement(s) for figure 6:

**Figure supplement 1.** Xrn1 binds to an internal region of Dcp2.

region eliminates Xrn1:Edc3 interaction (*Figure 6E*, compare alleles N630 and N475). In contrast to Upf1:Edc3 and Pat1:Edc3 interactions, Xrn1:Edc3 interaction was not inhibited by endogenous Edc3. We suspect that the observed Xrn1:Edc3 interaction may involve dimerization of exogenous Edc3 with endogenous Edc3 bound to Dcp2 in a decapping complex. Elimination of the Dcp2 C-terminal domain and deletion of *EDC3* both abolished Xrn1:Upf1 interaction (*Figure 6B*), indicating that this interaction is bridged by Dcp2 and occurs in a Dcp1–Dcp2–Edc3–Xrn1–Upf1 complex (*Figure 6F*). Consistent with being an interaction bridged by Dcp2, Xrn1:Upf1 interaction required the second Upf1-binding motif U1$_2$ in Dcp2, as loss of the U1$_2$ motif by Dcp2 C-terminal truncation eliminated Xrn1:Upf1 interaction (*Figure 6E*, compare alleles N771 and N630).

Collectively, our two-hybrid analyses show that Xrn1 binds to an internal fragment of Dcp2 and is recruited to the decapping complex by Dcp2. These results indicate that two important events in 5′ to 3′ mRNA decay, Dcp2-mediated decapping and Xrn1-mediated 5′ to 3′ exoribonucleolytic digestion, are physically coupled in vivo, suggesting that in addition to controlling the selective targeting of the decapping enzyme to different decapping substrates, the C-terminal domain of Dcp2 also controls efficient 5′ to 3′ exonucleolytic decay, ensuring that decapped mRNAs are degraded in a timely manner.

## Scd6 interacts with multiple decapping factors and exists in both Edc3- and Pat1-containing decapping complexes

Our genetic analyses presented in *Figure 1C, D* indicated that the E3-1 motif binds not only Edc3 but an additional factor as well. As Scd6 and Edc3 share similar domain structures and appear to have redundant functions in mRNA decay (*Albrecht and Lengauer, 2004*; *Decourty et al., 2008*), we reasoned that Scd6 might bind to the Dcp2 E3-1 motif. To test this idea, we first examined two-hybrid interactions between Scd6 and each of the known yeast decapping factors. We found that Scd6 interacted with Dcp1, Dcp2, Edc3, and Pat1, but did not interact with all other decay factors tested, including Dhh1 (*Figure 7A*).

To assess whether the observed interactions between Scd6 and Dcp1, Dcp2, Edc3, and Pat1 in WT cells are direct or mediated by interactions with other factors, we repeated the approach of *Figure 5A*, assaying each of the Scd6:Dcp1, Scd6:Dcp2, Scd6:Edc3, and Scd6:Pat1 interactions in a panel of tester strains harboring specific decay factor gene deletions or a large Dcp2 C-terminal truncation. As shown in *Figure 7B*, Scd6 interacted with Dcp1 strongly in WT cells and this interaction was not affected by deletions of *UPF1*, *PAT1*, or *SCD6*. However, Scd6:Dcp1 interaction was greatly diminished by deletion of *EDC3*, and was eliminated by the *dcp2-N245* truncation. In an *edc3Δ* background, Scd6 interacted weakly with Dcp1. This weak Scd6:Dcp1 interaction was not affected by deletions of *UPF1* or *SCD6*, but was eliminated by deletion of *PAT1* or the *dcp2-N245* truncation. Together, these results indicate that the observed Scd6:Dcp1 interaction in WT cells is bridged by the C-terminal domain of Dcp2 and that both Edc3 and Pat1 can promote the joining of Scd6 into decapping complexes. In the presence of Edc3, Scd6 exists in a Dcp1–Dcp2–Edc3–Scd6 complex and, in the absence of Edc3, Scd6 can exist in a Dcp1–Dcp2–Scd6–Pat1 complex.

Scd6 interacted weakly with Edc3 in WT cells (*Figure 7B*). Much like the Upf1:Edc3 and Pat1:Edc3 interactions shown in *Figure 5A*, the Scd6:Edc3 interaction was enhanced by deletion of *EDC3* and was eliminated by the *dcp2-N245* truncation in both *EDC3* and *edc3Δ* backgrounds (*Figure 7B*). These results indicate that the observed Scd6:Edc3 interaction in WT cells is bridged by the C-terminal domain of Dcp2 and that endogenous Edc3 inhibits the Scd6:Edc3 two-hybrid interaction. Interestingly, in an *EDC3* background, deletion of *PAT1* also promoted the Scd6:Edc3 interaction (*Figure 7B*), suggesting that Pat1 may regulate the interaction of Scd6 with an Edc3-containing decapping complex.

The Scd6:Dcp2 and Scd6:Pat1 interactions observed in WT cells (*Figure 7A*) appear to be direct, as these two interactions were not affected by the *dcp2-N245* truncation or the deletions of the *EDC3* or *PAT1* (*Figure 7C*). A direct Scd6:Dcp2 interaction is further supported by the fact that the observed Scd6:Dcp1 and Scd6:Edc3 interactions are both bridged through Dcp2 (*Figure 7B*). Collectively, these results indicate that Scd6 interacts with multiple decapping factors including Dcp1, Dcp2, Edc3, and Pat1 and that, depending on the Edc3 status in the cell, Scd6 can exist in Dcp1–Dcp2–Edc3–Scd6 or Dcp1–Dcp2–Scd6–Pat1 complexes.

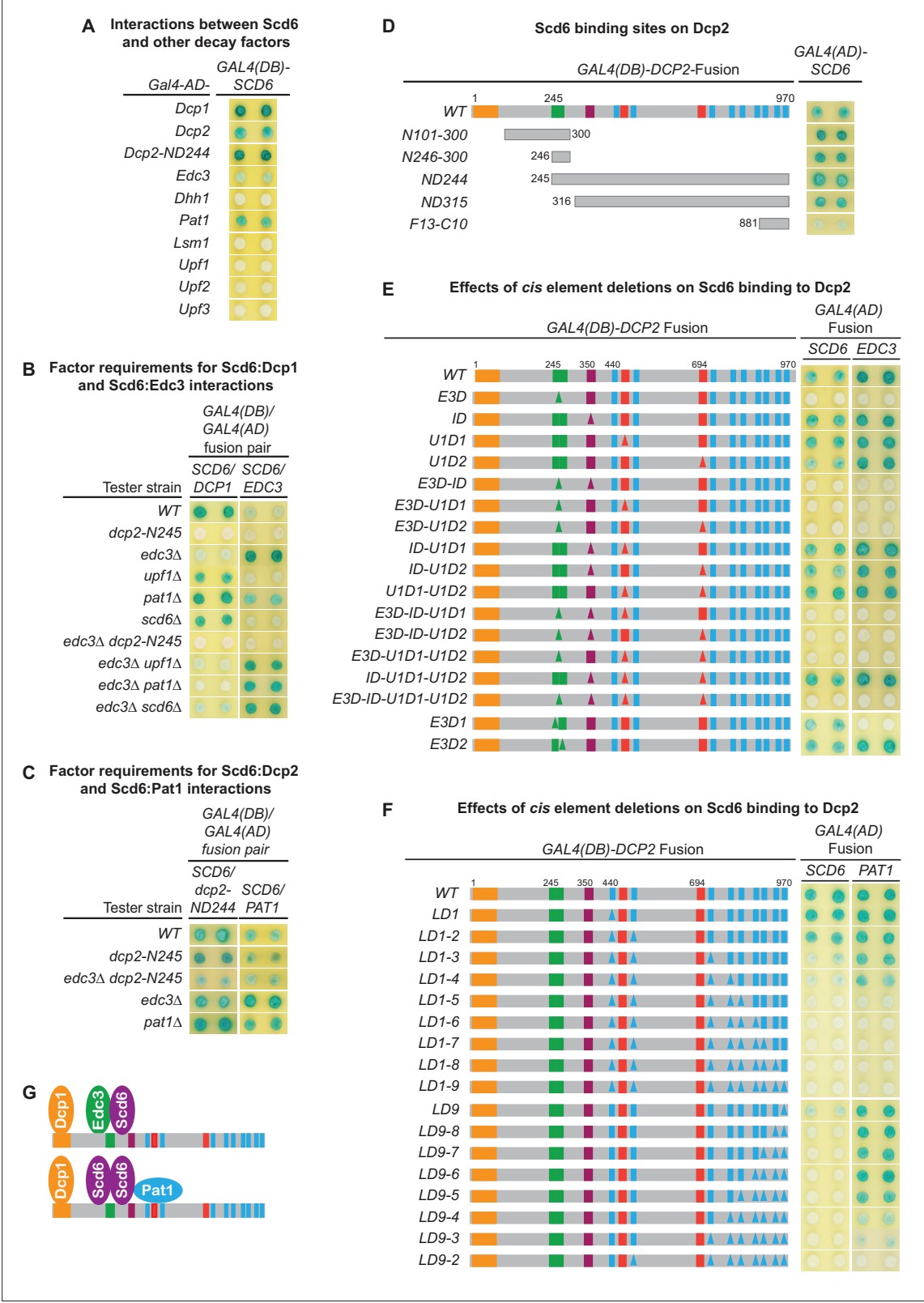

**Figure 7.** Scd6 binds to Dcp2 via multiple elements including the Edc3-binding and leucine-rich motifs and exists in both Edc3- and Pat1-containing decapping complexes. (**A**) Two-hybrid analyses of Scd6 interactions with known yeast decapping factors in a WT tester strain. (**B**) Two-hybrid analyses of Dcp2-bridged interactions between Scd6 and Dcp1 or Edc3. (**C**) Two-hybrid analyses of interactions between Scd6 and Pat1 or N-terminally truncated Dcp2. (**D**) Two-hybrid analyses of Scd6 binding to different Dcp2 fragments. See also *Figure 7—figure supplement 1*. (**E**) Two-hybrid assays evaluating

*Figure 7 continued on next page*

*Figure 7 continued*

the consequences of deleting the Dcp2 inhibitory element and the Edc3- or Upf1-binding motifs on Dcp2 interactions with Scd6 and Edc3. (**F**) Two-hybrid assays evaluating the consequences of deleting the leucine-rich motifs ($L_1$–$L_9$) from the Dcp2 C-terminal domain on Dcp2 interactions with Scd6 and Pat1. (**G**) Two different decapping complexes inferred from two-hybrid analyses in B, E, and F. Allele schematics and two-hybrid analyses as in *Figure 1*.

The online version of this article includes the following figure supplement(s) for figure 7:

**Figure supplement 1.** Scd6 binds to multiple Dcp2 fragments with or without specific overlaps.

## Scd6 binds to Dcp2 through multiple elements including the Edc3-binding and leucine-rich motifs

To map the Scd6-binding site(s) on Dcp2, we tested the two-hybrid interaction of Scd6 with a panel of Dcp2 fragments (*Figure 7—figure supplement 1*). Scd6 interacted strongly with the ND244 fragment that contains the entire Dcp2 C-terminal domain (*Figure 7D*, *Figure 7—figure supplement 1*). Scd6 interacted moderately with two N-terminal Dcp2 fragments (N101-300 and N246-300) and moderately or weakly with six additional C-terminal fragments (ND315, ND415, F9-C10, F10-C10, F12-C10, and F13-C10) (*Figure 7D*, *Figure 7—figure supplement 1*). The N101-300 and N246-300 fragments and the ND244 fragment share a 55-amino acid overlap of the Edc3-binding region. The six C-terminal fragments share different overlaps with the N244 fragment, but have no overlap with the two N-terminal fragments. Thus, the pattern of Scd6 binding to different Dcp2 fragments indicates that Scd6 most likely binds multiple elements in Dcp2 and that one of the Scd6-binding elements coincides with the Edc3-binding region.

To further elucidate the mode of Scd6's binding to Dcp2 and to identify the potential Scd6-binding elements in Dcp2, we analyzed the two-hybrid interactions of Scd6 with two panels of *dcp2* element deletion alleles (*Figure 7E, F*). Scd6 interacted strongly with WT full-length Dcp2, and loss of the inhibitory element (*ID*), the first Upf1-binding element (*U1D1*), or the second Upf1-binding element (*U1D2*) either individually or in different combinations did not affect Scd6:Dcp2 binding (*Figure 7E*). Loss of the Edc3-binding motif E3-1 (*E3D1*) or the E3-2 motif (*D3D2*) partially weakened the Scd6:Dcp2 interaction; however, loss of both the E3-1 and E3-2 motifs (*E3D*) eliminated Scd6:Dcp2 interaction (*Figure 7E*, compare the *E3D1* and *E3D2* alleles to all the *E3D*-containing alleles). These results indicate that the E3-1 and E3-2 motifs can both promote Scd6 binding and function redundantly to recruit Sdc6 to Dcp2.

Loss of the leucine-rich motifs $L_1$ and $L_2$ (*LD1-2*) did not affect the binding of Scd6 to Dcp2; further loss of the $L_3$ motif (*LD1-3*) or both $L_3$ and $L_4$ motifs (*LD1-4*) partially weakened Scd6:Dcp2 interaction to similar extents, and further loss of the $L_5$ motif (*LD1-5*) eliminated Scd6:Dcp2 interaction (*Figure 7F*). These results indicate that leucine-rich motifs $L_3$ and $L_5$ are required for Scd6 binding to Dcp2. These two motifs each make a partial contribution to Scd6 binding and function redundantly to recruit Scd6 to Dcp2. Interestingly, loss of the last leucine-rich motif $L_9$ (*LD9*) also weakened Scd6:Dcp2 interaction, and loss of the last two leucine-rich motifs $L_8$ and $L_9$ (*LD9-8*) eliminated Scd6:Dcp2 interaction (*Figure 7F*), indicating that leucine-rich motifs $L_8$ and $L_9$ are also required for Scd6 binding to Dcp2. The $L_8$ and $L_9$ motifs each make a partial contribution for Scd6 binding and appear to function redundantly in recruiting Scd6 to Dcp2.

In sum, we identified three pairs of functionally redundant binding motifs in Dcp2 (E3-1 and E3-2, $L_3$ and $L_5$, and $L_8$ and $L_9$) that promote Scd6 binding to Dcp2. As loss of any of these binding motif pairs eliminates Scd6 binding to Dcp2, Scd6 may use three different regions to engage these three pairs of Dcp2-binding motifs cooperatively. In this case, Scd6 may bind to full-length Dcp2 either as a monomer or as a binding-induced dimer with the dimer having a higher affinity, and each monomer engaging three different binding motifs, one from each of the three pairs. Alternatively, Scd6 may use its Lsm domain singlehandedly to engage Dcp2. In this case, Scd6 may bind to full-length Dcp2 using allovalency as described above for Pat1, requiring a minimum of five Dcp2-binding motifs.

## Discussion

### C-terminal *cis*-binding motifs promote independent binding of Edc3, Scd6, Upf1, and Pat1 to Dcp2 and control selective targeting of the decapping enzyme to distinct substrate mRNAs

The yeast decapping enzyme targets thousands of mRNAs for decapping-dependent decay (*Celik et al., 2017*; *He et al., 2018*) and the genetic experiments described here elucidate the mechanistic basis for the enzyme's selective targeting of specific subgroups of those mRNAs. Dcp2's large C-terminal domain encompasses multiple regulatory elements that serve as binding sites for proteins heretofore thought of as 'decapping activators' and our systematic deletion of those sites has defined their roles. Loss of both Upf1-binding motifs eliminated Upf1 binding to Dcp2 and caused selective partial stabilization of NMD substrates; loss of the Edc3-binding motif eliminated Edc3 binding to Dcp2 and caused selective partial or complete stabilization of Edc3 substrates, as well as selective partial stabilization of Dhh1 substrates (*Figure 1C, D*). Further, deletions of the leucine-rich Pat1-binding motifs eliminated Pat1 binding to Dcp2 and, when these deletions were combined with deletions of the Edc3-binding motif, caused selective additional stabilization of Dhh1 substrates (*Figure 4B*).

Our observation that loss of the Edc3-binding motif caused selective stabilization of Dhh1 substrates uncovers a new direct role of Edc3 in selective targeting of the decapping enzyme to Dhh1-regulated mRNAs. This role is not totally unexpected since Edc3 interacts directly with Dhh1 (*He and Jacobson, 2015*; *Sharif et al., 2013*), Dhh1 association with the decapping enzyme is dependent on Edc3 (*Figure 5A*), and deletion of *EDC3* causes additional stabilization of Dhh1-regulated mRNAs in *dcp2-E3D* and *dcp2-E3D1* cells (*Figure 4D*). In contrast to Dcp2 *cis*-element E3D or E3D1 deletions, *trans* deletion of *EDC3* had no discernible effect on Dhh1 substrates (*Figures 1B and 4B*), a result suggesting that at least one additional factor binds to the Edc3-binding motif and can target the decapping enzyme to Dhh1-regulated mRNAs. Since Scd6 also binds to the Edc3-binding motif E3-1 and exists in both Edc3 and Pat1-containing decapping complexes (*Figure 7B, E*), we propose that Scd6 functions redundantly with Edc3 in targeting the decapping enzyme to Dhh1 substrates. Surprisingly, given the multiple proposed roles for Pat1 in mRNA decapping (*Charenton et al., 2017*; *Lobel et al., 2019*; *Nissan et al., 2010*), we found that loss of the Pat1-binding motifs had no effect on Pat1/Lsm1 substrates or any other decapping substrates (*Figure 2B*). We propose that Pat1 binding to the leucine-rich motifs in the Dcp2 C-terminal domain still controls the selective targeting of the decapping enzyme to Pat1/Lsm1 substrates and may even enhance the decapping rates of the targeted mRNAs as demonstrated in vitro (*Lobel et al., 2019*), but the Pat1-mediated decapping step is likely not rate limiting for decay of Pat1/Lsm1 substrates.

### Dcp2 C-terminal *cis*-binding motifs promote assembly of distinct target-specific decapping complexes in vivo

Consistent with our observation that Dcp2 *cis*-binding elements control the selective targeting of the decapping enzyme to different decapping substrates, our two-hybrid experiments reveal that the same elements can promote in vivo assembly of distinct target-specific complexes. The Edc3-binding motif E3-1 can promote the assembly of a Dhh1-containing Dcp1–Dcp2–Edc3–Dhh1 complex, the two Upf1-binding motifs $U1_1$ and $U1_2$ can promote assembly of two distinct Upf1-containing Dcp1–Dcp2–Edc3–Upf1 complexes of the same composition but different configurations with Upf1 binding either to the $U1_1$ or $U1_2$ motifs, and the Pat1-binding motifs $L_1$ and $L_2$ can promote assembly of two distinct Pat1-containing Dcp1–Dcp2–Edc3–Pat1 complexes of the same composition but different configurations with Pat1 binding either to the $L_1$ or $L_2$ motifs (*Figure 5D, E*). Further, the motifs E3-1 and E3-2 in combination with motifs $L_3$ and $L_5$, and $L_8$ and $L_9$ can promote the assembly of a Dcp1–Dcp2–Edc3–Scd6 complex and a Dcp1–Dcp2–Scd6–Scd6–Pat1 complex (*Figure 7G*). Based on several observations, including: (1) Edc3 associates with itself (*Decker et al., 2007*; *He and Jacobson, 2015*), (2) Edc3 targets two specific transcripts *RPS28B* mRNA and *YRA1* pre-mRNA for decapping-dependent decay (*Badis et al., 2004*; *Dong et al., 2007*), (3) the *YRA1* intron contains two functionally redundant Edc3-binding elements (*Dong et al., 2010*), and (4) Edc3 binds to the 3'-UTR decay-inducing element of *RPS28B* mRNA as a dimer (*He et al., 2014*), we postulate that the Edc3-binding motif can promote assembly of at least one additional Edc3 substrate-specific decapping complex, a Dcp1–Dcp2–Edc3–Edc3 complex containing Edc3 as a homodimer (*Figure 8*). Because Edc3 is directly involved in

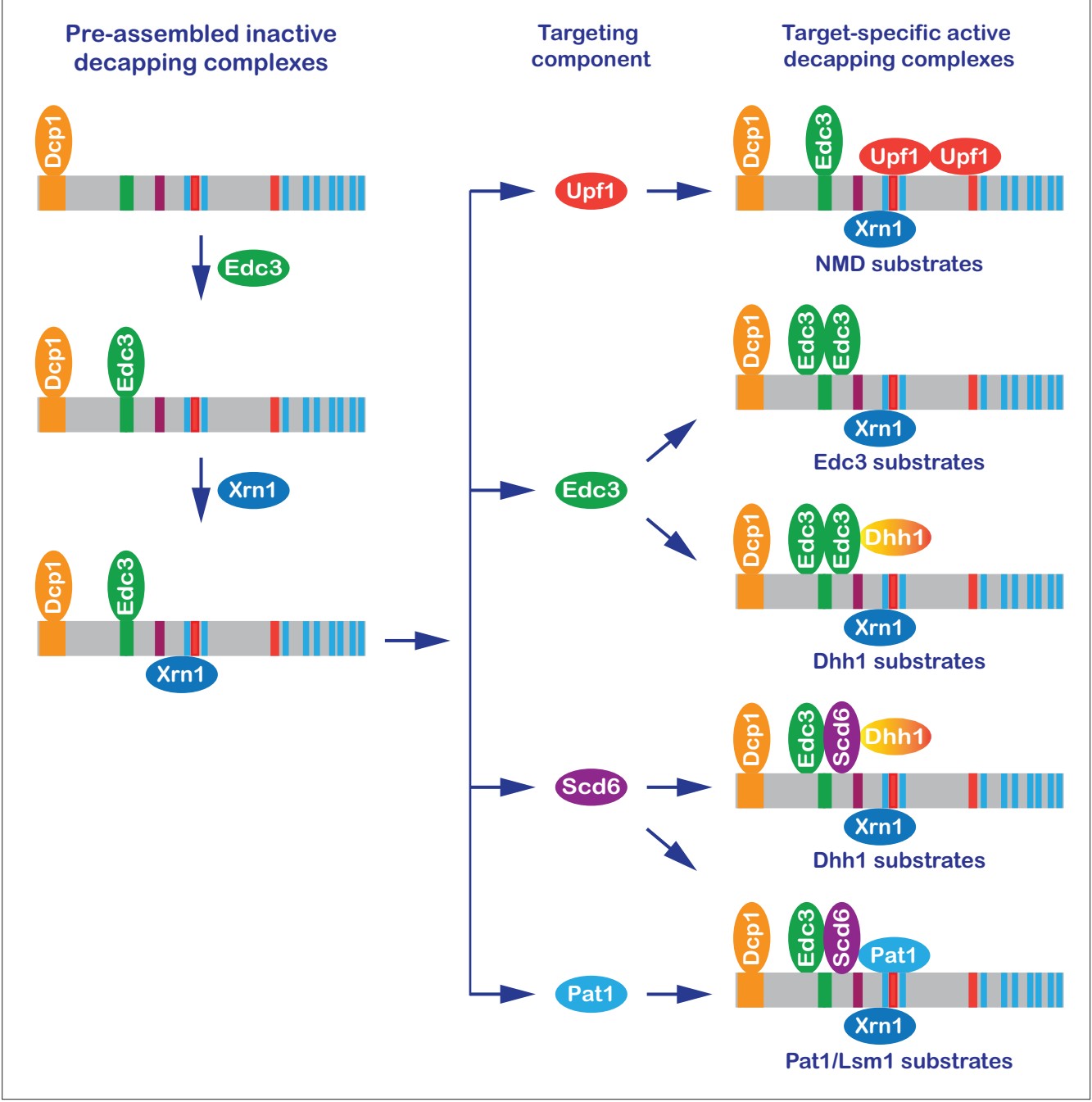

**Figure 8.** A new model: formation of target-specific decapping complexes controls mRNA decapping. In brief, the Dcp1–Dcp2 decapping enzyme interacts with Edc3 and then Xrn1, forming a preassembled inactive Dcp1–Dcp2–Edc3–Xrn1 complex. The resulting complex binds to individual targeting components, forming target-specific decapping complexes. The final assembly and enzymatic activation of these complexes are postulated to occur on mRNPs destined for degradation.

decay of Dhh1 substrates (*Figure 4D*), and it also promotes Sdc6 joining to the decapping complex (*Figure 7*), we speculate that the Dcp1–Dcp2–Edc3–Dhh1 complex likely contains Edc3 either as an Edc3:Edc3 homodimer or as an Edc3:Scd6 heterodimer (*Figure 8*). Because the Scd6-binding motifs in Dcp2 exhibit functional redundancies, we suspect that the Dcp1–Dcp2–Scd6–Pat1 complex formed in the absence of Edc3 (*Figure 7B*) likely contain Scd6 as an Scd6:Scd6 homodimer (*Figure 7G*). Since Scd6 also binds to the Edc3-binding motif E3-1, the dimeric Scd6:Scd6-containing decapping complexes may form even in an *EDC3* background. The dimeric forms of Edc3:Edc3, Edc3:Scd6, and

Scd6:Scd6 can all occupy the E3-1 and E3-2 motifs and compete with each other for binding to Dcp2. Collectively, our results indicate that the *cis*-binding elements located in the Dcp2 C-terminal domain promote assembly of multiple decapping complexes with distinct composition and substrate specificities. Edc3 appears to be both a common core component of multiple decapping complexes and a unique targeting component of specific decapping complexes.

Our observation that Edc3 is a common component of multiple decapping complexes raises the intriguing issue of Edc3 function in these complexes. Numerous biochemical and structural studies, all using C-terminally truncated Dcp2 fragments and largely the Lsm domain of Edc3, suggested that Edc3 promotes catalysis by the decapping enzyme by alleviating the autoinhibition imposed by Dcp2 inhibitory elements, thus enhancing substrate binding (*Charenton et al., 2016*; *Fromm et al., 2012*; *Mugridge et al., 2018*; *Nissan et al., 2010*; *Paquette et al., 2018*). Our genetic experiments reveal that the core Edc3 component must carry out two additional important functions in mRNA decapping. One function is to provide the decapping enzyme with a set of unique Edc3-binding surfaces or modules and thus endow the decapping enzyme with Edc3 targeting specificity. Consistent with this proposition, eliminating Edc3 binding to Dcp2 caused selective stabilization of both Edc3 and Dhh1 substrates (*Figure 1C, D*). Another Edc3 function is to promote the assembly of additional target-specific decapping complexes, such as the Dcp1–Dcp2–Edc3–Upf1 and Dcp1–Dcp2–Edc3–Scd6 complexes, by remodeling Dcp2 or providing weak but specific binding surfaces for specific targeting or coupling factors. In support of this notion deletion of *EDC3* diminishes the Dcp2-bridged Dcp1:Upf1, Dcp1:Xrn1, and Dcp1:Scd6 interactions (*Figures 5A–7B*), and triggers a switch of Upf1 binding from the $U1_1$ to the $U1_2$ motif in Dcp2 (*Figure 5B*). Our proposition that Edc3 exists both as a common core component of multiple decapping complexes and as a unique targeting component of specific decapping complexes provides a unified theory for explaining the apparently contradictory proposed functions for Edc3, that is, that Edc3 functions as a general decapping activator (*Kshirsagar and Parker, 2004*) or as a transcript-specific decapping activator (*Badis et al., 2004*; *Dong et al., 2007*).

## Edc3-, Upf1-, and Pat1-mediated active recruitment of the decapping enzyme makes distinct contributions to the overall decay of the respective targeted mRNAs

Our genetic analyses of Dcp2 *cis* element mutants revealed the relative mRNA decay contributions of specific factor-mediated decapping enzyme recruitment events and provided significant insights into the decay mechanisms for different decapping substrates. For the Edc3 substrate *RPS28B* mRNA, loss of the Edc3-binding motif by E3D or E3D1 in Dcp2 caused complete mRNA stabilization relative to deletion of *EDC3* itself (*Figure 1C, D*). For the Edc3 substrate *YRA1* pre-mRNA, and all the tested Dhh1 substrates, loss of the Edc3-binding site caused only partial stabilization relative to the respective *EDC3* or *DHH1* deletions (*Figures 1C and 4B*). Additional loss of the Pat1-binding motifs caused further stabilization of the latter Edc3 and Dhh1 substrates (*Figure 4B*). These results indicate that both Edc3 and Dhh1 substrates can be targeted by Pat1-mediated recruitment of the decapping enzyme. However, since loss of the Pat1-binding motifs alone had no effect on these Edc3 and Dhh1 substrates (*Figure 2B*), it appears that Pat1-mediated recruitment of the decapping enzyme merely functions as a backup system in the decay of Edc3- and Dhh1-regulated mRNAs. Because the combined deletions of the Edc3- and Pat1-binding motifs caused substantial stabilization of the Edc3 and Dhh1 substrates (*Figure 4B*), our results indicate that decapping is rate limiting for both Edc3 and Dhh1 substrates, suggesting that the major functions of Edc3 or Pat1 in decay of these mRNAs is recruiting the decapping enzyme.

For NMD substrates, loss of both Upf1-binding motifs caused only marginal stabilization (two- to threefold), but deletion of *UPF1* caused substantial stabilization (>tenfold) (*Figure 1C*). These results indicate that Upf1-mediated recruitment of the decapping enzyme only makes a minor contribution to the overall decay of NMD substrates, suggesting that decapping is not a major rate-limiting step in NMD, active recruitment of the decapping enzyme to NMD substrates may be largely dispensable, and Upf1 must carry out one additional major function upstream of the recruitment of the decapping enzyme. Consistent with these ideas, in the absence of active recruitment of the decapping enzyme by Upf1, NMD substrates are still degraded by decapping-dependent pathway (*Figure 3B*). In addition, further deletions of the Edc3- and Pat1-binding motifs do not have significant effects on

NMD substrates (*Figure 4C*), and in fact, even eliminating the entire C-terminal domain only caused marginal stabilization of NMD substrates (*He and Jacobson, 2015*), as observed for loss of both Dcp2 Upf1-binding motifs.

For Pat1/Lsm1 substrates, loss of the Pat1-binding motifs had no effect, but deletion of *PAT1* caused substantial stabilization (*Figure 2B*). These results indicated that the Pat1-mediated recruitment of the decapping enzyme does not make a significant contribution to the overall decay of Pat1/Lsm1 substrates, suggesting that decapping is not rate limiting in decay of the Pat1/Lsm1 substrates, active recruitment of the decapping enzyme to the Pat1/Lsm1 substrates can be dispensable, and Pat1 must carry out one additional major function upstream of the recruitment of the decapping enzyme. In support of this conclusion, in the absence of active recruitment of the decapping enzyme by Pat1, Pat1/Lsm1 substrates are still degraded by the decapping-dependent pathway (*Figure 3C*). Importantly, eliminating all the known-binding motifs in the Dcp2 C-terminal domain also did not have any effect on the decay Pat1/Lsm1 substrates (*Figure 4C*).

## Dcp2 directly recruits the 5′ to 3′ exoribonuclease Xrn1 to the decapping complexes

Our two-hybrid experiments revealed that the 5′ to 3′ exoribonuclease Xrn1 binds to Dcp2 and is directly recruited to decapping complexes by Dcp2. This conclusion is supported by several new observations. First, Xrn1 binds to a specific internal Dcp2 fragment (*Figure 6D*, *Figure 6—figure supplement 1B*). Second, Xrn1 also interacts with both Dcp1 and Edc3, two other core components of the decapping enzyme, and each of the interactions requires the Dcp2 C-terminal domain as well as an intact Xrn1-binding region in this domain (*Figure 6B, E*). Third, Xrn1 is a common component of both the Dcp1–Dcp2–Edc3–Edc3–Xrn1 and Dcp1–Dcp2–Edc3–Upf1–Xrn1 complexes (*Figure 6F*). Finally, Xrn1's recruitment to the decapping complexes requires prior Edc3 binding to Dcp2, as deletion of *EDC3* eliminated both Dcp1:Xrn1 and Upf1:Xrn1 interactions (*Figure 6B*).

Our observation that Xrn1 is directly recruited by Dcp2 to decapping complexes suggests that decapping and 5′ to 3′ exoribonucleolytic decay are physically and mechanistically linked. This coupling appears to be conserved over eukaryotic evolution, but the mechanism of this coupling may differ for different organisms, for example, *D. melanogaster* Xrn1 binds to Dcp1 (*Braun et al., 2012*) and human Xrn1 binds to Edc4 in its respective decapping complex (*Chang et al., 2019*). Xrn1 binding to decapping complexes likely ensures immediate 5′ to 3′ exoribonucleolytic degradation of decapped transcripts and may also serve to inhibit the catalytic activity or substrate binding of the decapping enzyme until the enzyme is targeted to substrate mRNAs. Consistent with this idea, Xrn1 binding to Dcp2 requires the inhibitory element (*Figure 6D*) and overexpression of Xrn1 in *D. melanogaster* cells can inhibit the decapping of different reporter mRNAs (*Braun et al., 2012*).

Based on the observation that the same Pat1 C-terminal extension binds to multiple helical leucine-rich motifs in Dcp2 and a single such motif in Xrn1, Pat1 was proposed to coordinate the decapping and exonucleolytic decay events in general 5′ to 3′ mRNA decay by first recruiting the decapping enzyme and then Xrn1 to targeted mRNAs (*Charenton et al., 2017*). This Pat1-mediated sequential coupling model of decapping and 5′ to 3′ exonucleolytic decay is at odds with our result that Xrn1 is directly recruited by Dcp2 to the decapping complexes. Since Pat1 targets only a subset of yeast transcripts (*He et al., 2018*), and loss of the Pat1-binding motifs had no effect on all tested decapping substrates (*Figure 2B*) yet loss of the Edc3- and Upf1-binding motifs each yielded specific effects (*Figure 1C*), we suggest that Pat1-mediated coupling may be limited to a small number of mRNAs and may not make a significant general contribution to the overall decay of most mRNAs.

## A new model for control of mRNA decapping

Our genetic data suggest a new model for control of mRNA decapping in yeast (*Figure 8*) whose main features include: *cis*-regulatory elements located in the Dcp2 C-terminal domain control selective targeting of the decapping enzyme by forming distinct multicomponent decapping complexes; the Dcp1 and Dcp2 subunits are shared common components of all decapping complexes; Edc3 functions as a shared common component of multiple decapping complexes, but is also a unique targeting component of dimeric Edc3-containing complexes; Xrn1 associates with Dcp2 after the binding of Edc3 to Dcp2 and is a component of multiple or perhaps all targeting complexes; Upf1 and Pat1 each function as unique targeting components in their respective Upf1- or Pat1-containing

complexes; Scd6 may function as a second targeting component of the Dhh1 complexes and likely also collaborates with Pat1 to target the Pat1/Lsm substrates; the targeting components Edc3, Scd6, Upf1, and Pat1 each have at least two separate binding modules, one for the Dcp1–Dcp2–Edc3 core complex and another for their targeted mRNPs; and the final assembly and decapping activation of the target-specific decapping complexes occur on a to-be-degraded mRNP and the decapping event is coupled with immediate 5′ to 3′ degradation by Xrn1.

We propose that monomeric Edc3 is present in the NMD and Pat1/Lsm1 substrate-specific decapping complexes because: (1) Dcp2 contains only one Edc3-binding motif mapped to a 17-amino acid fragment (E3-1) (*Figure 1D*) and this fragment can only engage one Edc3 monomer for binding in the *K. lactis* Dcp1-Dcp2-Edc3-m$^7$-GDP structure (*Charenton et al., 2016*), and (2) Edc3 and Upf1 or Pat1 co-occupy the Dcp2 C-terminus (*Figure 5A*) and the joining of Upf1 and Pat1 to the decapping complexes is sensitive to Edc3 copy number, as endogenous Edc3 inhibits both Edc3–Upf1 and Edc3–Pat1 interactions in the two-hybrid assay (*Figure 5C*). We propose that an Edc3 dimer is present in the Edc3 and Dhh1 substrate-specific decapping complexes because our results suggest that one monomer functions as a core component and another monomer functions as a targeting component. In this scheme targeting of the Edc3- and Dhh1-specific decapping complexes to their respective substrates is mostly controlled by homodimerization of the targeting and core Edc3 components.

Target-specific decapping complexes other than those specific for NMD, Edc3, Dhh1, and Pat1 substrates must exist in yeast cells. Our observation that Scd6 binds to the sole Edc3-binding motif on Dcp2 also suggests the existence of non-Edc3-containing decapping complexes in yeast cells. In addition, the leucine-rich motifs $L_6$ to $L_9$ originally proposed for Pat1 binding in fact may not bind Pat1 in the context of full-length Dcp2 and the motifs $L_6$ and $L_7$ each still lack an assigned binding activity, suggesting that these motifs may bind other factors. Further, loss of all the known factor-binding motifs in Dcp2 does not have any discernible effect on decapping of Pat1/Lsm1 substrates, raising the possibility that these mRNAs may be decapped by the Dcp1/Dcp2 complex alone. Collectively, our results indicate that yeast cells likely contain many distinct decapping complexes. The notion of a single yeast decapping enzyme is too simplistic and should be replaced by the concept of a decapping enzyme family that shares the core components Dcp1 and Dcp2, similar to the PP1 and PP2A phosphatase families involved in protein dephosphorylation (*Shi, 2009*; *Virshup and Shenolikar, 2009*).

## Materials and methods

**Key resources table**

| Reagent type (species) or resource | Designation | Source or reference | Identifiers | Additional information |
|---|---|---|---|---|
| Chemical compound, drug | [α-$^{32}$P]-dCTP | Perkin Elmer | Blu513Z | |
| Chemical compound, drug | 5-Bromo-4-chloro-3-indolyl-beta-D-galactopyranoside (X-GAL) | USB | 7240-90-6 | 40 µg/ml |
| Chemical compound, drug | Phenol | Fisher | A92-500 | |
| Chemical compound, drug | Phenol:choroform:IAA (25:24:1) | Ambion | AM9732 | |
| Chemical compound, drug | Herring Sperm DNA | Promega | D1815 | |
| Chemical compound, drug | Geneticin (G418 Sulfate) | Gibco | 11811-023 | 400 µg/ml |
| Chemical compound, drug | Hygromycin B | Roche | 10843555001 | 200 µg/ml |
| Peptide, recombinant protein | Taq DNA polymerase | Roche | 04-728-874-001 | |
| Commercial assay or kit | QuikChange XL Site-Directed Mutagenesis Kit | Agilent Technologies | 200,519 | |
| Commercial assay or kit | Random Primed DNA labeling Kit | Roche | 11-004-760-001 | |
| Strain, strain background (*Escherichia coli*) | DH5α | Invitrogen | 11319-019 | Electrocompetent cells |
| Strain, strain background (*Saccharomyces cerevisiae*) | *Supplementary file 1* | This paper | Yeast strains used in this study | See Materials and methods |

*Continued on next page*

*Continued*

| Reagent type (species) or resource | Designation | Source or reference | Identifiers | Additional information |
|---|---|---|---|---|
| Genetic reagent (plasmid) | *Supplementary file 2* | This paper | Plasmids used in this study | See Materials and methods |
| Genetic reagent (plasmid) | *Supplementary file 4* | This paper | DNA probes used in this study | See Materials and methods |
| Sequence-based reagent | *Supplementary file 3* | This paper | Oligonucleotides used in this study | See Materials and methods |
| Software, algorithm | Multi-Gauge software | Fujifilm | Science lab 2005 | |
| Software, algorithm | GraphPad Prism 9 for Windows | GraphPad Software, LLC | https://www.graphpad.com | |
| Other | Zeta-Probe Blotting Membranes | BioRad | 1620159 | *He et al., 2008* |
| Other | Mini Quick Spin RNA Columns | Roche | 11-814-427-001 | *He et al., 2008* |

## Experimental model and cell growth conditions

The experiments described in this study used the yeast *S. cerevisiae* as a model system. Yeast strains used for phenotypic analyses of mRNA decay were all derived from the W303 background (*MATa ade2-1 his3-11,15 leu2-3,112 trp1-1 ura3-1 can1-100*). Yeast strains used for two-hybrid analyses were all derived from the GGY1::171 background (*his3 leu2 URA3::GAL1-lacZ gal4Δ gal80Δ*) (*Fields and Song, 1989*). All yeast strains used in this study are listed in *Supplementary file 1*.

Yeast cells used for RNA isolation and transformation were all grown at 30°C in YEPD media (1% yeast extract, 2% peptone, 2% D-glucose). For integrative yeast transformation, transformants were selected at 30°C on different media depending on the selection markers of the transforming DNA fragments. For the drug resistance genes *KanMX* and *HygroR*, G418 (Gibco #11811-023), and hygromycin (Roche #10843555001) were included in YEPD media (1% yeast extract, 2% peptone, 2% D-glucose, 30 g agar/l, 400 µg/ml G418, or 200 µg/ml hygromycin). For the auxotrophic markers, *URA3* and *ADE2*, synthetic -*ura* or -*ade* drop-out media (6.7 g/l of yeast nitrogen base without amino acids, 2 g/l -*ura* or -*ade* drop-out mix, 30 g/l of agar, 100 ml/l of 20% D-glucose) were used. For yeast two-hybrid analyses, transformants were first selected on synthetic -*leu-his* drop-out media and then replica-plated on SSX media [6.7 g/l of yeast nitrogen base without amino acids, 2 g/l -*leu-his* drop-out mix, 30 g/l of agar, 100 ml/l of 20% sucrose, 100 ml/l of 1 M potassium phosphate buffer pH 7.0, 2 ml/l of 20 mg/ml of 5-bromo4-chloro-3-indolyl β-D-galactoside (X-gal) (USB #7240-90-6) in formamide] for color development.

## Method details

### Methodological overview: generation and functional analysis of Dcp2 element deletion mutants

To dissect the roles of different Dcp2 regulatory elements (*Figure 1A*) in mRNA decapping, we generated specific deletions in the C-terminal domain of *DCP2* that eliminated either a single element or combinations of different elements and analyzed the consequences of these deletions on Dcp2 interactions with specific decapping activators and decay of different decapping substrates. In total, we constructed three sets of element deletions and generated 50 *dcp2* mutant alleles. The first set of *dcp2* alleles contains single deletions as well as double, triple, and quadruple deletions in all possible combinations of the inhibitory element and the Edc3- and Upf1-binding motifs. The second set of *dcp2* alleles contains consecutive deletions of the nine leucine-rich Pat1-binding motifs, proceeding either from the N-terminal end of this set of motifs or from its C-terminus. The third set of *dcp2* alleles contains deletions of different combinations of Edc3-, Upf1-, and Pat1-binding motifs.

To link the potential defect in mRNA decapping to a specific Dcp2 interaction, we analyzed each of the *dcp2* element mutant alleles in two parallel assays. In the first assay, each *dcp2* allele was fused to the *GAL4* DNA-binding domain and the encoded fusion proteins were tested for interactions with those encoded by *EDC3*, *UPF1*, and *PAT1* fused to the *GAL4* activation domain in the yeast two-hybrid system. In the second assay, each *dcp2* allele was N-terminally tagged with a triple HA epitope

and integrated at the genomic locus of *DCP2* for functional analysis. Using quantitative northern blotting analyses, we measured the steady-state levels of different decapping substrates in each *dcp2* element mutant. Decapping substrates analyzed in this study include the NMD substrates *CYH2* pre-mRNA and *ade2-1*, *can1-100*, and *trp1-1* mRNAs; the Edc3 substrates *YRA1* pre-mRNA and *RPS28B* mRNA; the Dhh1 substrates *EDC1* and *SDS23* mRNAs; the Pat1/Lsm1 substrates *AGA1*, *BUR6*, *DIF1*, and *LSM3* mRNAs; and the Pat1/Lsm1/Dhh1 substrates *CHA1*, *HSP12*, and *HXT6* mRNAs. It should be noted that the presence of the HA-tag at the N-terminus of Dcp2 had no effect on its function in mRNA decay (*Figure 1C*, compare *HA-DCP2* to *WT*).

## Plasmid construction

Plasmids used in this study are listed in *Supplementary file 2*. Plasmids containing the original WT *DCP2* allele, *HA-DCP2* allele, and *dcp2* alleles with different C-terminal truncations were previously described (*He and Jacobson, 2015*). A plasmid containing the *edc3::URA3* allele was described in *Dong et al., 2007*. Plasmids containing the *xrn1::ADE2*, *ski2::URA3*, *ski7::URA3*, and *scd6::KanMX6* alleles were described in *He et al., 2018*. Plasmids containing full-length *DCP1*, *UPF1*, *UPF2*, *UPF3*, *EDC3*, *PAT1*, *LSM1*, and *DHH1* alleles, or different *DCP2* fragments fused to either *GAL4(AD)* or *GAL4(DB)* were described in *He and Jacobson, 2015*. Plasmids constructed in this study are described below.

## Construction of the *dcp2* element deletion alleles

*dcp2* alleles harboring regulatory element deletions were all constructed in Bluescript by using the plasmid HFSE1645 either as an initial template or as a cloning vector (see *Supplementary file 2*). This plasmid contains the WT *HA-DCP2* allele as a 3.7 kb *SalI–BglII/NotI* fragment, including 320 bp from the promoter/5'-UTR region and 401 bp from the 3'-UTR region. Specific element deletions were all generated by using the QuikChange Site-Directed Mutagenesis Kit from Agilent Technologies (cat# 200519) according to the manufacturer's instructions. PCR primers used to generate specific Dcp2 element deletions are listed in *Supplementary file 3*. *dcp2* alleles containing single element deletions were generated by using plasmid HFSE1645 as a template. *dcp2* alleles containing two or more element deletions were generated by two different strategies. One strategy used sequential rounds of site-direct mutagenesis and the other used molecular cloning to combine pre-existing element deletions from different *DCP2* regions. Details on construction of each *dcp2* element deletion allele are provided in *Supplementary file 2*. Specific element deletions in individual *dcp2* mutant alleles were all confirmed by DNA sequencing. Each of these *dcp2* alleles can be isolated from the corresponding plasmids as a *SalI–BglII* DNA fragment. For clarity, we use the following matched pairs of abbreviations for each specific element and its deletion: the inhibitory element: IE/ID; the Edc3-binding motif: E3/E3D, E3-1/E3D1, and E3-2/E3D2; the Upf1-binding motifs $U1_1$/U1D1 and $U1_2$/U1D2; and the nine leucine-rich motifs $L_1$ to $L_9$/LD1 to LD9. Analysis of *dcp2* alleles harboring different deletions of the inhibitory element and the Edc3- and Upf1-binding motifs revealed that loss of the inhibitory element affected neither the binding of Edc3, Upf1, and Pat1 to Dcp2, nor the steady-state levels of different decapping substrates (*Figure 1B, C*). Thus, *dcp2* alleles harboring a deletion of the inhibitory element are not discussed further and we focused our analysis on deletions of the Edc3-, Upf1-, and Pat1-binding motifs.

## Construction of C-terminally truncated *dcp2* alleles

To facilitate the construction of the *dcp2-KanMX6* knock-in alleles harboring different *DCP2* C-terminal truncations, we generated modified versions of the *dcp2-N925*, *N770*, *N635*, *N475*, *N400*, and *N300* alleles in pRS315. These modified *dcp2* alleles are identical to those that were described in *He and Jacobson, 2015* except that each has a shorter 3'-UTR fragment and two additional restriction sites *BamHI* and *NotI* added to the 3'-end. The modified *dcp2* alleles were constructed in two steps. In the first step, a 5' *XbaI–NcoI* fragment isolated from the original WT *DCP2* allele and a 3' *NcoI–SalI* fragment amplified from the *DCP2* 3'-UTR region were ligated to pRS315 digested by *XbaI* and *XhoI*, generating pRS315-DCP2-WT-M1(HFSE1632). In the second step, the individual *SalI–NcoI dcp2* fragments were isolated from each of the original *dcp2* truncation alleles and these DNA fragments were then ligated to HFSE1632 digested by *SalI* and *NcoI*. Each of the modified *dcp2* truncation alleles can be isolated as either a *XbaI–BamHI* or a *SalI–BamHI* fragment.

## Construction of the *dcp2-KanMX6* knock-in alleles

All *dcp2-KanMX6* knock-in alleles were constructed in Bluescript by using the plasmid HFSE1636 as the cloning vector (see **Supplementary file 2**). This plasmid contains the previously described *dcp2-N245-KanMX6* allele as a 3.5 kb *NotI–BamHI/NotI/SalI* fragment, including 925 bp from the *DCP2* promoter/5′-UTR region and 401 bp from the 3′-UTR region (**He and Jacobson, 2015**). In this *dcp2* allele, the 1452 bp *BglII–EcoRI* KanMX6 selection cassette was inserted into the promoter region 588 bp upstream from the first base of the *DCP2* initiation codon. Two classes of *dcp2* knock-in alleles were constructed. One class contains different *dcp2* element deletions and the other contains different *dcp2* C-terminal truncations. For construction of the *dcp2* element deletion knock-in alleles, individual *SalI–BglII* DNA fragments were isolated from each of the original *dcp2* element deletion alleles in Bluescript and ligated to HFSE1636 previously digested by *SalI* and *BamHI*. For construction of the *dcp2* truncation knock-in alleles, individual *SalI–BamHI* DNA fragments were isolated from each of the modified *dcp2* truncation alleles in pRS315 and ligated to HFSE1636 previously digested by *SalI* and *BamHI*. Each of the *dcp2* knock-in alleles can be isolated as a *NotI–NotI* fragment for integrative yeast transformation.

## Construction of the *Gal4(DB)-dcp2* fusion alleles

Coding sequences from each of the *dcp2* element deletion alleles were fused to the *GAL4* DNA-binding domain by using plasmid pMA424 as the cloning vector (see **Supplementary file 2**). To facilitate the construction of these fusion alleles, we first used the plasmid HFSE1718 as a cloning vector to generate an intermediate allele from each of the original *dcp2* element deletion alleles. Plasmid HFSE1718 contains the entire WT *DCP2* coding region and 222 bp from the 3′-UTR region as a *BamHI–NcoI/SalI* DNA fragment. To generate the intermediate *dcp2* alleles, depending on specific cases, either a 5′ PCR-amplified *BamHI–XhoI* fragment, or a 3′ restriction *XhoI–NcoI* fragment, or both of these fragments were obtained from the *dcp2* element deletion alleles in Bluescript and then ligated to plasmid HFSE1718 previously digested by *BamH–XhoI*, *XhoI–NcoI*, or *BamH–NcoI*, respectively. To generate the final *Gal4(DB)-dcp2* fusion alleles, each of the *dcp2* intermediate alleles was isolated from the corresponding plasmid as a *BamHI–SalI* DNA fragment and then ligated to pMA424 previously digested by *BamHI–SalI*. All *Gal4(DB)-dcp2* fusion alleles were confirmed by DNA sequencing.

## Construction of the *XRN1* or *SCD6* and *Gal4(DB)* or *Gal4(AD)* fusion alleles

The entire *XRN1* coding sequence was fused to the *GAL4* DNA-binding domain and activation domain by using plasmids pMA424 and pGAD-C2 as cloning vectors, respectively. Both fusion alleles were constructed through a three-piece ligation reaction by making use of the unique S*acI* restriction site located at nt 293–298 of the *XRN1* coding region. The *XRN1* coding sequences from nt 1–309 were amplified by PCR using oligonucleotide pair XRN1-5′-BamHI-F and XRN1-5′-SacI-R (**Supplementary file 3**). The resulting PCR product was digested by *BamHI* and *SacI*, yielding a 5′ *BamHI–SacI* *XRN1* fragment encompassing the first 297 nts of its coding sequences. A 3′ *SacI–SalI* *XRN1* fragment encompassing the coding sequences from nt 298 to 4584 was isolated from plasmid HFSE1532. The 5′ *BamHI–SacI* and 3′ *SacI–SalI* *XRN1* fragments were then ligated to pMA4242 and pGAD-C2 digested by *BamHI* and *SalI* to generate the final *XRN1* fusion alleles. Similarly, the entire 1050 bp coding region of was amplified by PCR using oligonucleotides SCD6-TH-F1 and SCD6-TH-C1 as primers. The resulting PCR product was cut by *EcoRI* and *SalI* and then ligated to pMA424 or pGAD-C2 previously digested by *EcoRI* and *SalI*. The *XRN1* and *SCD6* fusion alleles were confirmed by DNA sequencing.

## Construction of the *upf1::KanMX6* and *edc3::Hygro* alleles

The *upf1::KanMX6* and *edc3::Hygro* alleles were constructed in Bluescript. To construct Bs-ks-*upf1::KanMX6*, a PCR-amplified 417 bp *NotI–BglII/NcoI* fragment from the *UPF1* 5′-UTR region and a PCR-amplified 416 bp *NcoI/EcoRI–SalI* fragment from the *UPF1* 3′-UTR region were ligated to Bluescript digested by *NotI* and *SalI* in a three-piece ligation reaction (*NotI–NcoI/NcoI–SalI*) to generate an intermediate plasmid containing a *upf1* allele lacking its entire CDS. The resulting intermediate plasmid was digested by *BglII* and *EcoRI* and then ligated to a 1452 bp *BglII–EcoRI* KanMX6 selection cassette. To construct Bs-ks-*edc3::Hygro*, a 455 bp PCR-amplified *NotI–XbaI/BamHI* fragment from the *EDC3* 5′-UTR region and a 512 bp PCR-amplified *BamHI/EcoRI–SalI* fragment from the *EDC3* 3′-UTR region were ligated to Bluescript digested by *NotI* and *SalI* in a three-piece ligation reaction

(*NotI–BamHI/BamHI–SalI*) to generate an intermediate plasmid containing an *edc3* allele lacking its entire CDS. The resulting plasmid was digested by *BamHI* and *SalI* and then ligated to a 1666 bp *BglII–SacI* Hygromycin resistant gene and a 508 bp PCR-amplified *SacI–SalI* fragment from the *EDC3* 3'-UTR region. The *upf1::KanMX6* and *edc3::Hygro* alleles can each be isolated as a *NotI–SalI* fragment for integrative yeast transformation.

## Strain construction

Yeast strains used in this study are listed in *Supplementary file 1*. The WT strain (HFY114) and its isogenic derivative harboring a deletion *of UPF1* were described previously (*He et al., 1997*), as were isogenic strains harboring deletions of *EDC3* (CFY25), *PAT1* (SYY2674), *LSM1* (SYY2680), or *DHH1* (SYY2686) (*He and Jacobson, 2015*). The yeast two-hybrid tester strain (GGY1::171) and its isogenic derivatives harboring deletions of *EDC3* (SYY1774), *PAT1* (SYY2451), and *DHH1* (SYY2467), or the *dcp2-N245* truncation of the Dcp2 C-terminal domain (SYY2390) were described previously (*He and Jacobson, 2015*). Yeast strains constructed in this study are described below.

## Construction of yeast strains harboring *dcp2* element deletions

To assess the consequences of deleting specific Dcp2 regulatory elements on mRNA decay, we constructed a set of yeast strains with the W303 background that harbor different *dcp2* element deletion alleles. Each of these alleles was tagged by a triple-HA epitope at the 5'-end of its coding sequence and was integrated at the genomic locus of *DCP2* by gene replacement (*Guthrie and Fink, 1991*). As a control, we also constructed a yeast strain harboring the HA-tagged WT *DCP2* allele. Plasmids harboring different *dcp2-KanMX6* alleles were digested by *NotI* to release the DNA fragments harboring specific *dcp2* element deletion knock-in alleles. About 2 μg of each digested plasmid was transformed into the WT yeast strain HFY114 by the high-efficiency LiOAC method (*Schiestl and Gietz, 1989*). After transformation, cells were cultured in 1 ml YEPD media at room temperature for 90 min and then plated on G418-containing YEPD plates to select for integration events. Plates were incubated at 30°C for 3–4 days. Individual stable transformants were isolated from the plates and patched again on G418-containing YEPD plates. The patched transformants served as master cells for both genotyping and long-term storage. The correct integration and deletion of specific elements for each *dcp2-kanMX* knock-in allele were confirmed by genomic DNA PCR and sequencing. The primers used for genomic DNA PCR and sequencing are listed in *Supplementary file 3*. Approximately 10 transformants were screened for each integrative yeast transformation.

To assess the decay mechanisms of different decapping substrates, we also introduced an *EDC3* deletion into yeast cells harboring *dcp2-E3D* or *E3D1* alleles, as well as deletions *of XRN1, SKI2,* or *SKI7* into yeast cells harboring *dcp2-E3D, E3D1, U1D1-U1D2,* and *LD1-9* alleles. Plasmids harboring the *edc3::URA3, xrn1::ADE2, ski2::URA3,* and *ski7::URA3* null alleles were used for yeast transformation. In each case, *a NotI–SalI* DNA fragment harboring the respective null allele was used for gene replacement. Each of the knock-out alleles was confirmed by genomic DNA PCR. In standard YEPD media at 30°C, yeast strains harboring individual *dcp2* element mutant alleles grow at rates comparable to those of WT cells, with doubling times of about 90 min.

## Construction of yeast strains expressing different Dcp2 C-terminal truncations

To assess the roles of the Dcp2 C-terminal domain in the formation of different decapping complexes, we constructed a set of two-hybrid tester strains harboring different C-terminal truncations of Dcp2 in both *EDC3* and *edc3Δ* backgrounds. These strains were also constructed by gene replacement. The transformation and selection procedures were identical to those described for the construction of yeast strains harboring different *dcp2* element deletion alleles except that: (1) plasmids harboring different *dcp2* C-terminal truncation knock-in alleles were used, and (2) each digested plasmid was transformed into the GGY1::171 and GGY1::171 *edc3::HygroR* (SYY3064) stains. The correct integration and C-terminal truncation for each *dcp2-kanMX* knock-in allele were confirmed by genomic DNA PCR and sequencing.

To assess the roles of Edc3, Upf1, Pat1, and Scd6 in the formation of different decapping complexes, we constructed additional yeast two-hybrid tester strains harboring single gene deletions of *EDC3, UPF1,* or *SCD6*; double gene deletions of *UPF1, PAT1,* or *SCD6* and *EDC3*; and the

*dcp2-N245* C-terminal truncation of Dcp2 and *EDC3* deletion. Strains harboring the single *EDC3*, *UPF1*, or *SCD6* deletions were constructed by transforming a DNA fragment containing either the *edc3::HygroR* or the *upf1::KanMX6* or the *scd6::KanMX6* null allele into the GGY1::171 strain. Strains harboring the double deletions of *UPF1*, *PAT1*, or *SCD6* and *EDC3,* and strains harboring the *dcp2-N245* C-terminal truncation and *EDC3* deletion were constructed by transforming a DNA fragment containing the *edc3::HygroR* null allele into the single *UPF1*, *PAT1*, or *SCD6* deletion strains *upf1::KanMX6* (SYY2973), *pat11::KanMX6* (SYY2451), *scd6::KanMX6* (SYY2976) strains or the *dcp2-N245* truncation strain *dcp2-N245::KanMX6* (SYY2390), respectively. The *edc3::HygroR* and *scd6::KanMX6* knock-out alleles in the respective strains were confirmed by genomic DNA PCR.

## Yeast two-hybrid interaction assay

Two-hybrid assays employed previously described procedures (*Fields and Song, 1989*; *He et al., 1997*; *He and Jacobson, 1995*; *He and Jacobson, 2015*; *He et al., 2014*) and all tester strains used were in the GGY1::171 background. In each case, a *GAL4(DB)* fusion construct (1.5 µg) was cotransformed with a *GAL4(AD)* construct (1.5 µg) into a tester strain using the high-efficiency LiOAc method. Transformants were plated on standard synthetic *-leu-his* drop-out media, incubated for 3–5 days at 30°C, and then replica-plated on X-Gal-containing SSX plates to observe the color development of the transformant population. The color phenotypes of two independent transformants from each interaction assay are presented in the figures. To assess the potential for self-activation, each of the *GAL4(DB)* fusions was also cotransformed with a GAL4(AD) empty vector into the GGY1::171 strain. In this study, only *GAL4(DB)-DHH1* exhibited weak self-activation (*He and Jacobson, 2015*).

## Cell samples for RNA isolation

Cells used for RNA isolation were all grown in YEPD media at 30°C. In each case, 15 ml YEPD in a 50 ml tube was inoculated with 1.5 $OD_{600}$ overnight culture and the resulting culture was grown in a shaking incubator (200 rpm) to an $OD_{600}$ of 0.7. Yeast cells in the 50 ml tube were pelleted by centrifugation in a benchtop centrifuge at 5000 rpm for 5 min. Cell pellets were resuspended in 0.5 ml fresh YEPD liquid medium and the cell suspension was then transferred to a 2 ml microcentrifuge tube. Cells were pelleted by centrifugation in a microcentrifuge at 12,000 rpm for 1 min and the liquid medium was removed from the tube. Cell pellets were frozen on dry ice and then stored at −80°C until RNA isolation.

## RNA isolation

Total RNA was isolated from yeast cells by using the hot phenol method described previously (*Herrick et al., 1990*). Briefly, each cell pellet from a 15-ml culture was resuspended in 500 µl buffer A (50 mM NaOAc pH5.2, 10 mM Ethylenediaminetetraacetic acid (EDTA), 1% Sodium dodecyl sulfate (SDS), 1% Diethyl pyrocarbonate (DEPC)) and mixed with 500 µl phenol presaturated with 50 mM NaOAc pH5.2, 10 mM EDTA (Fisher, Cat# A92-500) prewarmed to 65°C. RNA was extracted by six cycles of 10 s of vortexing followed by a 50 s water bath incubation at 65°C. Samples were centrifuged in a microcentrifuge at 12,000 rpm for 3 min. After centrifugation, the phenol layer from each sample was removed with a Pasteur pipette and 500 µl prewarmed buffer-saturated phenol was then added to each sample, followed by another six cycles of RNA extraction. After the final extraction cycle, samples were centrifuged in a microcentrifuge at 12,000 rpm for 10 min and the aqueous layer from each sample was recovered and transferred to a new microcentrifuge tube. Additional (400 µl) phenol/chloroform/isoamyl alcohol (25:24:1) (Ambion, Cat# AM9732) was added to the sample and the mixture was vortexed for 2 min followed by a 10-min centrifugation at 12,000 rpm. The aqueous layer was recovered and subjected to another round of phenol/chloroform extraction. The aqueous layer was recovered and 40 µl NaOAc (3 M, pH 5.2) and 1 ml ethanol were added to each sample. RNA was precipitated at −70°C for 1 hr. Samples were centrifuged at 12,000 rpm for 15 min and pellets were washed two times with 70% ethanol. RNA pellets were air-dried for 15 min and dissolved in 80–100 µl RNase free distilled water. The RNA concentration of each sample was determined by measuring the $A_{260}$ value of a diluted sample.

## Northern blotting analysis

Procedures for northern blotting were described previously (*He and Jacobson, 1995*). In brief, 15 µg total RNA from each sample was loaded onto a formaldehyde-containing 1% agarose gel that was electrophoresed in 1× 3-(N-morpholino)propanesulfonic acid (MOPS) buffer (40 mM MOPS, 10 mM NaoAc, 1 mM EDTA, pH 7.0) overnight (1 hr at 70 V and 16 hr at 23 V). RNA separated on the gel was transferred a cellulose membrane (BioRad, Zeta-probe #1620159) by vacuum blotting with a sequence of 5 min in 50 mM NaOH/100 mM NaCl, 5 min in 100 mM Tris–HCl pH 7.0, and 1 hr in 20× SSC buffer. After the transfer, the membrane was crosslinked with a UV Stratalinker 2400 and washed with a RPDW buffer (0.1× SSC, 0.1% SDS) at 58°C for 30 min. For random primed DNA probes, prehybridization of the membrane was carried out in Pre-Hyb buffer (50% formamide, 5× SSPE, 10× Denhardt's solution, 1% SDS, 0.5 mg/ml sheared salmon sperm DNA) at 42°C for 2 hr, and hybridization was carried out in Hyb buffer (50% formamide, 5× SSPE, 2× Denhardt's solution, 5% dextran sulfate, 1% SDS, 0.25 mg/ml sheared salmon sperm DNA) at 42°C overnight. The membrane was washed with RPDW buffer 2× at room temperature for 10 min each, and then 2× at 58°C for 30 min each. Transcript-specific hybridization signals on the membrane were detected and imaged with a FUJI BAS-2500 analyzer. The images of northern blots were analyzed with MultiGauge software.

Random primed DNA probes were generated by using the Random Primed DNA Labeling Kit from Roche (cat# 11-004-760-001). Generally, 25–50 ng of purified DNA fragment was used to make the probe. The reaction was carried out in a total volume of 20 µl, containing 10 µl denatured DNA fragment, 2 µl 10× concentrated reaction mixture, 1 µl 10 mM dATP, 1 µl 10 mM dGTP, 1 µl 10 mM dTTP, 3 µl α-$^{32}$P-dCTP (6000 Ci/mmol, Perkin Elmer, Blu513Z), and 2 µl Klenow enzyme. The reaction was incubated at 37°C for 1–4 hr and stopped by heating at 65°C for 5 min. The reactions were then diluted to 80 µl with sterile distilled water and purified with a mini-Quick-Spin Column (Roche #11814427001) according to the manufacturer's instructions.

DNA fragments used as template for making α-$^{32}$P-dCTP-labeled probes were isolated from plasmids. Depending on the specific gene, the DNA fragment covered part of the coding region, the entire coding region, both exon and intron, the entire 3′-UTR, or the entire gene. DNA sequences for each of these template DNA fragments are listed in *Supplementary file 4*. DNA probes used in this study included those that are specific for NMD substrates (*CYH2*, *CAN1*, *ADE2*, *TRP1*, *CPA1*, and *EST1*), Edc3 substrates (*YRA1* and *RPS28B*), Dhh1 substrates (*EDC1* and *SDS23*), Pat1/Lsm1 substrates (*LSM3*, *DIF1*, *BUR6*, and *AGA1*), and Pat1/Lsm1/Dhh1 substrates (*HXT6*, *HSP12*, and *CHA1*). Each northern blot was also hybridized with an *SCR1* probe to serve as a loading control. With a few exceptions, each experiment was repeated independently two or three times.

## Quantification and statistical analysis

The transcript-specific image signals from each northern blot were quantified with MultiGauge software. The band intensity data were saved as txt files and then exported to and analyzed in Microsoft Excel. To determine the relative expression levels of a specific transcript in different strains, transcript-specific signals in these strains were first normalized to the corresponding *SCR1* signals and the resulting normalized signals in different strains were then divided by those from the WT strain. With a few exceptions, each northern blotting experiment was repeated independently at least two times. Bar graphs in relevant figures were generated by GraphPad Prism 9, mostly using the average ± SEM data.

There are a large number of northern blots in our paper (157 blots in total), and 16 out of 157 (about 10%) of the blots do not have error bars. The 16 graphs that do not have error bars were all presented in our initial phenotypic analyses of *dcp2* element mutants (*Figures 1C and 2B*). Eight blots (*RPS28B*, *EDC1*, *SDS23*, *HXT6*, *HSP12*, *LSM3*, *BUR6*, and *DIF1* mRNAs) in *Figure 1C* did not have replicates (*Figure 1—figure supplement 2*). However, the phenotypic analyses for each of these substrates in the relevant *dcp2* element mutants were independently repeated in our subsequent experiments shown in *Figures 1D, 3B, and 4B, E*. Likewise, eight blots (*HSP12*, *can1-100*, *ade2-1*, *RPS28B*, *EDC1*, and *SDS23* mRNAs, and *CYH2* and *YRA1* pre-mRNAs) in *Figure 2B* did not have replicates (*Figure 2—figure supplement 1*). However, the phenotypic analyses for six out eight of these substrates in relevant *dcp2* element mutants were independently repeated in our subsequent experiments (*Figure 4B*). In short, only two mRNAs in *Figure 2B*, *HSP12* and *ade2-1*, did not have repeat experiments. Further, in the experiments of *Figure 2B*, deletion of the leucine-rich elements

has no effect. From a genetics perspective each of the individual mRNA substrates comprises an independent experiment which thus has thirteen independent repeats. Our data for this experiment are thus extremely strong even without error bars. In sum, our analyses followed rigorous standards.

## Acknowledgements

This work was supported by a grant to AJ (1R35GM122468) from the U.S. National Institutes of Health. We thank Robin Ganesan and Kotchaphorn Mangkalaphiban for comments on the manuscript.

## Additional information

### Competing interests

Allan Jacobson: is co-founder, director, and Scientific Advisory Board chair for PTC Therapeutics Inc. The other authors declare that no competing interests exist.

### Funding

| Funder | Grant reference number | Author |
| --- | --- | --- |
| National Institute of General Medical Sciences | MIRA 1R35GM122468 | Allan Jacobson |

The funders had no role in study design, data collection, and interpretation, or the decision to submit the work for publication.

### Author contributions

Feng He, Conceptualization, Data curation, Formal analysis, Investigation, Methodology, Validation, Writing – original draft, Writing – review and editing; Chan Wu, Data curation, Formal analysis, Visualization, Writing – review and editing; Allan Jacobson, Conceptualization, Formal analysis, Funding acquisition, Project administration, Resources, Supervision, Writing – original draft, Writing – review and editing

### Author ORCIDs

Feng He http://orcid.org/0000-0002-7523-2602
Allan Jacobson http://orcid.org/0000-0002-5661-3821

### Decision letter and Author response

Decision letter https://doi.org/10.7554/eLife.74410.sa1
Author response https://doi.org/10.7554/eLife.74410.sa2

## Additional files

### Supplementary files

- Supplementary file 1. Yeast strains used in this study.
- Supplementary file 2. Plasmids used in this study.
- Supplementary file 3. Oligonucleotides used in this study.
- Supplementary file 4. DNA fragments used as probes in this study.
- Transparent reporting form

### Data availability

Source data associated with figures of northern blotting analyses have been deposited in the Dryad repository (https://datadryad.org/stash) and within that site can be found at https://doi.org/10.5061/dryad.pc866t1px. The availability of source data files is indicated in the text.

The following dataset was generated:

| Author(s) | Year | Dataset title | Dataset URL | Database and Identifier |
|---|---|---|---|---|
| He F, Wu C, Jacobson A | 2021 | Dcp2 C-terminal cis-binding elements control selective targeting of the decapping enzyme by forming distinct decapping complexes | https://dx.doi.org/10.5061/dryad.pc866t1px | Dryad Digital Repository, 10.5061/dryad.pc866t1px |

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
