## [Editor Report]

This paper is an important contribution to the fields of mRNA decapping and mRNA decay. Using a series of elegant genetic approaches and assays, this study tackles the difficult challenge of mapping binding interactions and subsequently assigning functions to binding partners involved in mRNA decapping. As a result, the findings reported in this work represent an important milestone towards deepening our mechanistic understanding of mRNA decapping and mRNA decay.

---

## [Decision Letter]

**Decision letter after peer review:**

[Editors’ note: the authors submitted for reconsideration following the decision after peer review. What follows is the decision letter after the first round of review.]

Thank you for submitting the paper "Dcp2 C–terminal *Cis*–Binding Elements Control Selective Targeting of the Decapping Enzyme by Forming Distinct Decapping Complexes" for consideration by *eLife*. Your article has been reviewed by 2 peer reviewers, and the evaluation has been overseen by a Reviewing Editor and a Senior Editor. The reviewers have opted to remain anonymous.

Comments to the Authors:

We are sorry to say that, based on the reviews (below) and after consultation between the reviewers and Reviewing Editor, we have decided that your manuscript, as it stands, cannot be considered further for publication by *eLife*.

Specifically, there were significant concerns regarding the statistical significance and strength of the data and whether the proposed model is well–enough supported by the data and takes other possible interpretations of the data into account. The reviewers thought that it would not be unlikely that the concerns could be addressed within the two months of revision that *eLife* thrives for, but if you are able to satisfactorily address all the issues raised, a revised manuscript could be considered as a new submission at a future date. We apologize for the unusually long time it took to reach this decision, but hope you find the reviewers' comments useful.

*Reviewer #2:*

The authors address how specific mRNA decapping components arrange into larger complexes and how this influences mRNA decapping/ degradation. To that end, the authors have used a tour–de–force experimental approach and have performed a very large number of Y2H experiments and northern blots.

I feel that the data is not always very strong (lack of error–bars). I fear that the hypothesis that is made (one element interacts with one binding partner; one mRNA is degraded in one manner) is not always correct and additional interactions that can take have not been considered.

The final model that the authors propose does not contain many new aspects and neglects known aspects (e.g. higher order complexes as found in P–bodies).

Unfortunately, I feel that the data is not always very strong. I fear that the hypothesis that is made (one element interacts with one binding partner; one mRNA is degraded in one manner) is not correct and additional interactions that can take have not been considered. In that light I have a large number of remarks that show I often disagree with the drawn conclusions or that I don't see much novelty. In that light I think that the paper is not suitable for publication in *eLife*, but would be a better fit for a more specialized journal (e.g. RNA).

Methods: "With a few exceptions, each northern blotting experiment was repeated independently at least two times. Bar graphs in relevant figures were generated by generated by GraphPad Prism 9, mostly using the average {plus minus} SEM data.". This description is not satisfactory at all. What are "few exceptions" (in a large number of the bar–graphs error–bars are lacking, which is not acceptable; a single blot does not seem to be an exception) and why are there exceptions at all (all experiments should be in triplicates)? And what was repeated, where those technical replicates or biological replicates? In summary, I would think that one needs to have at least 3 real replicates (independent growth of the cells, independent gel, blot and analysis). Two technical replicates (where the same blot was run twice with the same sample) is not very useful. Also, all gels should be available in full size in some supplement. The differences are small in many cases and this requires data and analysis that is much more rigorous as what is currently done.

The argumentation in the manuscript assumes that specific substrates are degraded by specific mechanism. I am not sure if that is really the case. A substrate is likely degraded by a combination of multiple mechanisms. For some substrates the decay rate might be slightly enhanced by one factor, for other substrates another factor might slightly shift the use of the different pathways. Often the authors just assume that substrates are only degraded in a specific manner, which is not true.

The authors also assume that the motifs in Dcp2 have a single function, either Edc3 binding, Upf1 binding or Pat1 binding. Based on previous studies, it is however clear that Edc3 can also bind to HLMs (that the authors refer to as Pat binding motifs only). Furthermore, the region around the motif that the authors call E3 has also been shown to have an inhibitory effect on the Dcp2 enzyme (e.g. 10.1093/nar/gky233). This effect has not been discussed in detail. Deletion of such elements can of course change mRNA levels, independent of any binding to activators. Deletion of a motif can thus have multiple effects (lower recruitment of a factor and increase/ decrease of activity through direct Dcp2 inhibition).

The authors should discuss any (growth) phenotypes of the strains they prepared. In case any of the deletions or truncations are functionally important I would expect that those strains have serious defects.

Pull–down experiments between the different components would be able to complement the Y2H experiments. This is important as I wonder to what degree do the endogenous proteins in the Y2H experiments interfere with the results? E.g. is the WT Dcp2 present in the cells that address the behavior of specific Dcp2 version.

Finally, I would like to mention that it is often hard to link the text to the figures. The figures are very large and finding the exact gels/ bars quickly is not always easy.

Page 3: "Dcp2 is a 970–amino acid protein" That depends strongly on the organism, this statement should be refined.

Page 3: "Edc3 was also originally thought …". Edc3 has also been reported to be important for the formation of processing bodies. This should be mentioned.

Page 4: "Edc1 and Edc2 were isolated as high–copy suppressors …. but they do not appear to be required for mRNA decapping in vivo. ". Deletion of Edc1 and Edc2 impairs mRNA decay (e.g. 10.1093/genetics/157.1.27). Based on that they are required for in vivo mRNA decapping. Please clarify.

Figure 5: Why is there an interaction between UPF1 and Dcp1 in the DelteEdc3 background in Figure 5A (row3, column2), but this interaction is not there in Figure 5B (row1, column4).

Page 5: "Wild–type (WT) Dcp2 exhibited a strong two–hybrid interaction with Upf1". This has been shown before: doi.org/10.1371/journal.pone.0026547. And that manuscript should thus be referenced.

Page 5: "Deletion of either the first (U1D1) or the second (U1D2) Upf1–binding motif had no discernible effect on Upf1's binding to Dcp2, but loss of both Upf1–binding motifs (U1D1–U1D2) eliminated Upf1 binding to Dcp2 (Figure 1B).". It appears that E3D–ID–U1D2 has an effect on the Dcp2–UPF interaction. Is that a real effect or not?

Page 6: "However, loss of both Upf1–binding motifs caused approximately two– to three–fold increases in the levels of these transcripts". Based on the bar–graphs the effect appears less than two–fold in some cases and rarely more than two–fold.

Page 6 "These increases were much smaller in magnitude than those caused by deletion of UPF1, which usually led to >10–fold increases for these transcripts.". This finding strongly suggests that additional UPF1 binding motifs must be present in Dcp2, or that Dcp2 does not play an important role in NMD. The authors should be more specific here, and not only mention that "decapping is maybe not rate–limiting" (what would then be rate limiting?) or by mentioning that "other pathways could be important" (which ones?). In the end, it raises the question if the Dcp2:Upf1 interaction is biologically important.

Page 7: The discussion of the Edc3 effects is very handwaving. For instance: "RPS28B mRNA is not likely degraded by an alternative pathway." Why not, there can be an alternative pathway that is completely independent of Edc3. Or: "…YRA1 pre–mRNA and Edc3 may play an additional role in the decay of this transcript." What roles of Edc3 would that be if it is not substrate recruitment, and why would that other role not be important for other mRNAs? Edc3 is known to enhance Dcp2 decapping efficiency in general.

Page 7: "…E3 also controls targeting of the decapping enzyme to Dhh1–regulated mRNAs". Is there any evidence that the E3 motif can directly interact with the Dhh1 helicase? Does E3 motif e.g. contain an FDF–like motif? Furthermore, it is known that Edc3 and Dhh1 directly interact. as the authors note and is it thus not most likely that Dhh1 is recruited to Dcp2 with the E3 motif and Edc3? This could be tested by changing the FDF motif in Edc3.

Page 8:"The partial stabilization of Dhh1 substrates caused by loss of E3 can be explained similarly as described above for YRA1 pre–mRNA.". I am not sure what the authors mean. They should be more explicit.

Page 8: "Edc3 binds to the conserved 17–amino acid segment." (E3–1). It is known that the Edc3 Lsm domain interacts with leucine rich motifs. Both E3–1 and E3–2 potentially contain such motifs (Figure S1). How do the authors explain then that only E3–1 is important? They could do some simple modeling to address this on a structural level.

Page 9: "These results indicate that in the context of full–length Dcp2, leucine–rich motifs L1 to L5 control the selective binding of Pat1 to Dcp2, most likely with a contribution from each motif, in contrast to the proposed mode for Pat1 binding to Dcp2 based on structural data (Charenton et al., 2017).". I disagree. First: based on the data the interaction between Dcp2 and Pat1 appears to require around 4 HLMs. Those can be either 1–4 (as in LD9–5) or 5–9 (as in LD1–4). The L1 to L5 are not more important than the others. Second, the Pat1:Dcp2 structure clearly shows that one HLM motif interacts with the C–terminal domain from Pat1, forming a 1:1 complex. Avidity effects can than result a more efficient recruitment of Pat1 to Dcp2 when multiple HLMs (any random ones) are present in Dcp2. The off–rates are just slower when more motifs are present. The data is thus fully consistent with the structure.

Figure 2—figure supplement 1: Why does deletion of Pat or Lsm1 stabilize the YRA1 and RPS28B mRNAs? These effects are on the same level as the effects that the authors discuss in the other figures and based on that I assume that they are real. However, the bar–graphs again have no error–bars.

Page 9: "…one possible explanation for this surprising observation is that decapping is not rate–limiting". Do these mRNAs contain strong secondary structure elements in the 5' end that would slow down Xrn1? Xrn1 is normally very processive and rapid and thus not rate limiting. What could be rate limiting if not decapping?

Page 9: "…Pat1 performs an unidentified major function upstream…". Please be more specific in what that could be. This is unsatisfactory.

Page 11: "This indicates that these decapping substrates can all be decapped by an alternative route when the normal Dcp2 cis–element–mediated active recruitment of the decapping enzyme is blocked.". This plainly shows that Dcp2 can be recruited to the mRNA by alternative manners, which is not surprising. In my opinion the network of interactions is very redundant and deleting one interaction (e.g. through the removal of a Dcp2 motif) is backed up by the other interactions. Pat1 and Dhh1 interacts with Dcp1 directly for instance, maybe that is the default route and the routes via the Dcp2 C–terminal IDR might be less important. In that light, it has also been shown that the complete Dcp2 C–terminal region can be deleted without causing large in vivo effects.

Page 12: "combining the leucine–rich element deletions LD1–8, LD1–9, LD9–3, and LD9–2 with the Edc3–binding element deletions E3D or E3D1 caused additional substantial stabilization of the Edc3 substrate YRA1 pre–mRNA". This merely reflects that fact that Edc3 can interact with the HLMs and the Pat can interact with the HLM that the authors call E3 motif here.

Page 12: "...cells harboring E3D1 consistently had lower transcript levels than those harboring E3D for Edc3 and Dhh1 substrates ". I really don't see that. HA–dcp2–E3D and HA–dcp2–E3D1 are the same for instance (Figure 4—figure supplement 2, YRA1 and RPS28 mRNAs). Maybe I am looking at the wrong thing, it is sometimes hard to find the corresponding graph.

Page 12: "Similarly, cells harboring LD9–3 also had consistently lower transcript levels than those harboring LD9–2 for both Edc3 and Dhh1 substrates, suggesting that LD9–3 deletion maintains more function of Dcp2 than that of LD9–2 in decay of these mRNAs." Again, I don't see that, the bars are the same to me.

Page 13: " raising the possibility that Pat1/Lsm1 substrates can be decapped without the function of any decapping activators.". Or there are just additional interactions that recruit Dcp2 to the mRNA and that are independent of the Dcp2 C–terminal region. The Dcp2 C–terminal region is not required and only adds some additional functionality.

Page 14: "Deletion of EDC3 caused additional 7–8–fold stabilization of the Edc3 substrate YRA1 pre–mRNA in E3D or E3D1 cells". Really, maybe I am again looking at the wrong graphs, but in my interpretation the levels change from around 3 (E3D or E3D1) to around 10 (when DeltaEdc3 is there too). That is a 3.5 fold stabilization. Seem that the authors exaggerate the effect.

Page 14: " Deletion of EDC3 also caused additional stabilization of the Dhh1 substrates EDC1 and SDS23 mRNAs and the Pat1/Lsm1/Dhh1 substrates HSP12 and HXT6 mRNAs". Those effects appear very small to me. Is that statistically really relevant?

Page 17 "assembly of the Dcp1–Dcp2–Edc3–Upf1 decapping complex". Is this a new finding? It was known that Edc3 and Upf1 interact (10.1371/journal.pone.0026547). Why is this (direct) Edc3–Upf1 interaction not shown in 5D?

Page 17: "Interestingly, deletion of EDC3 enhanced Pat1:Dcp1 interaction". The most logical explanation is that Edc3 also binds to the HLMs. Deletion of Edc3 thus makes these HLMs available for the interaction with Pat1. The complete discussion in the paper seems too complex to explain this.

page 19:"suggesting that Xrn1 binding to Dcp2 may be dependent on a specific Dcp2 conformation.". The only logical explanation is that a sequence in the C–terminal region is inaccessible in a specific Dcp2 conformation. As many structures of Dcp2 are known this should be addressed in more detail. Currently the explanation is too handwaving.

Page 20:"We suspect that the observed Xrn1:Edc3 interaction may involve dimerization of exogenous Edc3 with endogenous Edc3 bound to Dcp2 in a decapping complex. ". Edc3 is always a stable dimer. Why would that only play a role here and not in the discussions above?

Page 22:"a new direct role of Edc3 in selective targeting of the decapping enzyme to Dhh1–regulated mRNAs. ". This appears trivial to me, as Dcp2:Edc3 and Edc3:Dhh1 interactions have been structurally described in detail (as the authors also write), so nothing new in my eyes.

Page 23: "Edc3–binding motif can promote assembly of at least one additional decapping complex, a Dcp1–Dcp2–Edc3–Edc3 complex….". That is trivial and can be extended to the known observation that Edc3 can bridge two Dcp2 proteins. In that manner Edc3 thus supports the formation very large complexes that contain many Dcp2 proteins that can all independently recruit factors.

Page 24:"both as a common core component of multiple decapping complexes and as a unique targeting component of specific decapping complexes provides a unified theory for explaining the apparently contradictory proposed functions for Edc3,". This, in my opinion, is generally accepted in the literature and not novel.

Page 28:" distinct multi–component decapping complexes". I am not sure if these really exist. The interactions are all weak and in a cellular environment constant rearrangements of the interactions will take place. Also, the decapping complexes are not isolated but form higher order complexes that are, when large enough. visible as P–bodies and that contain many Dcp2 proteins. The provided model is thus not relevant in my eyes as it oversimplifies things too much.

*Reviewer #3:*

Yeast Dcp2 is a large protein, containing a structured N terminus (residues 1–245) and a disordered C–terminus comprised of 725 amino acids. Prior studies by Jacobson and colleagues indicate that deletion of the C–terminus of Dcp2 results in dysregulation of 1/3 of all protein–coding transcripts in yeast (He et al., *eLife* 2018). They also previously showed that the C–terminus of Dcp2 harbors binding sites for cofactors such as Edc3, Pat1 and Upf1 which act on different classes of transcripts during normal mRNA decay and quality control pathways such as NMD. (He and Jacobson RNA, 2015).

In the present study, the authors perform an analysis of Dcp2 protein interactions with Edc3 , Pat1, and Upf1 and find the binding sites are separable with respect to the effects of steady–state levels of specific transcripts in yeast: mutation of a single Edc3 binding site in the C–terminus of Dcp2 increases steady–state levels of YRA1 and RPS28B mRNA compared to the Edc3 gene deletion to varying extents; mutation of both UPf1 binding sites in Dcp2 C–terminus partially stabilizes NMD transcripts compared to the Upf1 gene deletion; mutation of the Pat1 binding sites in the C–terminus of Dcp2 does not affect steady–state levels Pat1 sensitive transcripts. The results suggest decapping is rate–limiting for decay of Edc3 and Upf1 transcripts but not for those sensitive to Pat1.

The most significant finding in the manuscript is that Edc3 may play a broader role in mRNA decay than was previously appreciated.

EDC3 was previously shown to regulate two transcripts in budding yeast, RPS28B and YRA1 ( Badis et al., Mol Cell, 2004 ;Dong et al., Mol Cell, 2007). Surprisingly, Dhh1 substrates are upregulated when the Edc3 binding site on Dcp2 is mutated (E3D) but not when the EDC3 is deleted. This observation suggests one additional factor may bind the Edc3 binding site of Dcp2. Moreover, deletion of the EDC3 gene together with mutation in its binding site in Dcp2 causes additional stabilization of Edc3 sensitive YRA1 mRNA and, to a lesser degree, Dhh1 sensitive transcripts. These observations suggest an extra function for Edc3 outside of recruitment of Dcp2 to mRNA and provide additional evidence Edc3 is involved in decay of Dhh1 sensitive mRNAs.

Additionally, the authors uncover a failsafe mechanism for control of decapping and 5'–3' decay. Partial increases in steady–state levels of Edc3 and Dhh1 sensitive transcripts caused by mutation of Edc3 binding site (E3D) can be restored to levels observed in Edc3 and Dhh1 deletion strains when Pat1 binding sites are eliminated.

To understand the interplay between cofactors that bind the C–terminal domain of Dcp2, the authors used yeast two–hybrid analyses. The data suggest Edc3 is a core component of multiple decapping complexes that can promote interactions of the Dcp2 C–terminal domain with Upf1 and Xrn1 or antagonize its interactions with Pat1.

Some open–ended questions for future studies include: 1–to identify the functional consequences of Dcp2–Xrn1 interactions for RNA degradation. It would be interesting to determine if there is an overlap in residues of Xrn1 required for its binding to Dcp2 with those shown to bind the 80S ribosome (Tesina et al., NSMB, 20219). 2–determine the identity of cofactors that work together with Edc3 to promote degradation of Dhh1 sensitive transcripts. One wonders if Scd6 and Edc3 work together in this regard. 3–to understand the impact on how binding site mutations in the C–terminal domain Dcp2 impact subcellular localization. Prior studies showed the Edc3 binding element (E3) promotes localization of Dcp2 to P–bodies (Xing et al., *eLife*, 2020); deletion of SCD6 and EDC3 promotes nuclear localization of an inactive form of Dcp2 (Tishinov, J. Sci. Sci. 2021).

The manuscript would be suitable for publication in a venue such as *eLife* if the following concerns are addressed:

1. The authors suggest one additional factor besides Edc3 is required for the decay of Dhh1 sensitive transcripts. Might this factor be Scd6? (re Jacobson and Hinnebusch, PLoS Genet 2018).

2. Related, if both Edc3 and Scd6 bind E3, the deletion of this element might result in relocalization of Dcp2 in the nucleus as described for the Edc3 and Scd6 double deletion strain (Tishinov and Spang, J Cell Sci 2021). To test, does mutation of Dcp2 NLS (K450T, described by Tshinov et al) reduce steady–state levels of RPS28b mRNA in the Dcp2 E3D background compared to Dcp2 E3D in isolation?

3. The yeast two–hybrid data presented in Figures 5 and 6 are dense and confusing. EDC3 deletion in tester strains has differential effects on interactions of the c–terminal domain of Dcp2 with Pat1, Upf1 and Xrn1. Deletion of Edc3 promotes binding of the C–terminal domain to Pat1 but inhibits binding to Upf1 and Xrn1. Is the former result from a competition between Edc3 and Pat1 binding? Is the latter from the ability of Edc3 to remodel the C–terminal domain of Dcp2 for recruitment of Upf1 and Xrn1?

4. In the discussion, manuscript p24, the authors state "Our genetic experiments challenge this proposed function for Edc3 and suggest that the core Edc3 component of each decapping complex may inhibit the enzymatic activity or substrate binding of the decapping enzyme, consistent with the existence of an inhibitory element in the Dcp2 C–terminus." Is there any evidence to suggest Edc3 inhibits decapping, as stated in the above sentence? If there are published data showing lesions in Edc3 decrease transcript levels, it should be cited otherwise the authors should include evidence of this claim.

5. The report of Xrn1 binding to Dcp2 is preliminary, as there are no functional data to suggest binding of Xrn1 to Dcp2 is important for 5'–3' decay (Figure 6). What are the consequences of disrupting the Dcp2–Xrn1 interaction identified in this study on 5'–3' mRNA decay? As the authors note in discussion, Pat1 can also recruit Xrn1 to the decapping complex but in my opinion, it is difficult to conclude whether or not the newly identified Dcp2:Xrn1 interaction 'challenges the Pat1 coupling model' as stated on manuscript p28.

6. The model depicted in Figure 7 is one of several that are consistent with the data. Can they rule out the existence of multiple decapping complexes with Upf1, Pat1 and Edc3 linking Dcp2 to a core of an mRNP and substrate targeting is dictated by another RNP component (or components) that bind transfactors or RNA directly? It seems unlikely Dcp2 exist on its own in a cell as depicted in their model.

7. The model posits that Edc3 acts as a dimer on Edc3 and Dhh1 substrates but not on Pat1/Lsm1 or NMD substrates. I could not find data to support this. Do the authors have evidence that Edc3 acts as a monomer on the indicated transcripts? Biochemical structural and genetic data from Parker and Song labs indicate Edc3 is a dimer in solution and in yeast (Ling, MCB , 2008).

8. In addition, Figure 7 has inconsistencies with other figures. The authors show Pat1 and Edc3 co–occupy the C–terminal domain of Dcp2 in Figure 5D but in the model depicted in Figure 7 this co–occupancy is omitted? If the authors do not think Edc3 and Pat1 co–occpy Dcp2 C–terminus to promote decay of Edc3 sensitive transcripts, then how does one explain the failsafe mechanism?

[Editors’ note: further revisions were suggested prior to acceptance, as described below.]

Thank you for resubmitting your work entitled "Dcp2 C–terminal *Cis*–Binding Elements Control Selective Targeting of the Decapping Enzyme by Forming Distinct Decapping Complexes" for further consideration by *eLife*. Your revised article has been evaluated by James Manley (Senior Editor), Ruben Gonzalez (Reviewing Editor), and only one of the previous two reviewers (Reviewer 3).

Please note that rather than asking Reviewer 2 to review your Response Letter and Revised Manuscript, we instead asked Reviewer 3 to comment on your assessment of and response to Reviewer 2's criticisms. As you will see from Reviewer 3's comments, outlined below for your attention, the manuscript has been improved but there are some remaining issues, including a few issues raised by Reviewer 2, that need to be addressed.

In addition to their formal comments below, Reviewer 3 also asked me to informally pass on an additional piece of constructive criticism. Specifically, Reviewer 3 found the tone of your response to be unnecessarily antagonistic, which they felt was not fully warranted. In addition, Reviewer 3 thought that, as written, the original manuscript was unfairly dismissive of much of the previous biochemical work and suspects that this dismissiveness might have impacted Reviewer 2's response to the original manuscript as well as the overall review process. Below are Reviewer 3's formal comments, which I would ask you to address in a final Response Letter and Revised Manuscript.

*Reviewer #3:*

The manuscript by Feng He et al. is an important contribution to the field–not only for mRNA decay but also for the area of intrinsically disordered proteins and proteins that have intrinsically disordered regions like Dcp2. It is quite challenging to reconstitute and resolve such proteins, and Feng He and colleagues have done a textbook job of mapping binding to short linear motifs (using Y2H) then assigning function to the motifs/binding partners in decay by a combination of genetic mutations and analyses of steady mRNA levels by Northern Blot. This type of analyses is 'gold standard' for moving from descriptive to truly mechanistic studies.

The authors have addressed most of my concerns, adding additional data to the manuscript, including an interaction analysis of Dcp2 with Scd6 and a discussion of the impact of E3 motif deletions on Dcp2 localization.

After considering the original submission, the revised manuscript and rebuttal I recommend publication in *eLife*. Some of the points in the revised manuscript require clarification, and some of the comments of Rev 2 should be addressed before the manuscript is accepted.

1. On p30 line 693 the authors speculate the Edc3 core component of each decapping complex may enhance the autoinhibitory activity of Dcp2. This implies Edc3 promotes ––instead of alleviates–– autoinhibition, contrary to what is implied in Feng et al. 2015 and Paquette et al., 2018. My question remains, what evidence is there that Edc3 is an inhibitor of decapping?

2. Related on page 25 lines 563–565, the authors suggest "Scd6 can bind full–length Dcp2 as a monomer, or dimer with the dimer having higher affinity and each monomer binding three different motifs, one from each of three pairs. " Is there literature indicating Scd6 can form a dimer? Why do the authors rule out that a single Scd6 can bind multiple motifs through its Lsm domain using allovalency, as elaborated by Klein, Pawson and Tyers (PMID: 14521832) and reviewed in PMID: 28597296. In my mind, and that of Rev 2, there are multiple models for how Pat1, Scd6 and Upf1 bind multiple sites in the Dcp2 C–terminal domain which require validation with biophysical methods beyond the scope of the authors manuscript. I suggest the authors present the simplest model consistent with their data, and mention that they cannot rule out other models. Nailing down the binding stoichiometry of cofactors for Dcp2 for a given class of mRNA would require biophysical methods beyond the scope of the present manuscript.

3. I appreciate the authors looking into whether or not deletion of the E3 motif affects decapping by altering the localization of Dcp2 but am not convinced that the data in their manuscript convincingly address this question. For example, they show steady–state levels of RPS28B mRNA increase by E3D but are not affected by the double mutation containing E3D and U1D1 (which "eliminates 12 of 18 residues of the NLS, including the four critical lysine residues changed in the K450T mutation" described by Tishnov and Spang). On ms p26 line 594 the authors tentatively conclude loss of E3 may not affect the nuclear localization of Dcp2 or the NLS identified in Dcp2 may not function as an NLS. Is it possible that there is an increase in cytoplasmic Dcp2 containing U1D1 and E3D mutations compared to Dcp2–E3D but no effect on decapping (due to lack of binding to Edc3 sensitive transcripts)? Because Tishinov and Spang directly monitor nuclear localization by fluorescence, whereas the authors of the present manuscript did not, I would suggest toning down the conclusion that the "NLS in Dcp2 identified (by Tishnov and Spang) is not an NLS." Likewise, if they think K450T abrogates nuclear localization of Dcp2 by abrogating binding of Pat1, then they ought to show that with a binding experiment, such as Y2H. Either the section should be re–written with more explicit mention of the caveats associated with inferring nuclear localization from steady–state mRNA levels, compared to direct methods to quantify localization by fluorescence microscopy as reported by Tishinov and Spang or the discussion of nuclear localization (p 25 lines 567 to p27 line 613) should be removed from this otherwise dense manuscript.

4. The authors have responded reasonably to Rev 2 's comments they deem 'appropriate'. In particular, I agree with the authors that there are an extraordinary number of northern blots (157) and 90% of them have error bars. The concern of this reviewer about rigor is unfair.

5–The authors' responses to Rev 2's comments that they deem are 'wrong' or confused are well–founded with the exception of two points that warrant further discussion or modification. (i)––the authors show in full–length Dcp2, leucine–rich motifs L1 to L5 control selective binding to Pat1 to Dcp2, most likely with a contribution from each motif, in contrast to the proposed mode of Pat1 binding to Dcp2 based on structural data (Charenton et al., 2017)." I agree with Reviewer 2 that this effect could be due to avidity, or if only one binding site on Pat1 engages a single motif (L1,L2, L3, L4 or L5) at a time , through allovalency (see point 2 above). The authors may discuss how multiple sites L1–L5 could collaborate to bind a single pocket on Pat1, as described with the crystal structure, consistent with allovalency. (ii)––There is a disagreement about whether full–length Edc3 always forms a dimer as asserted by Rev 2, or if in cells it may be regulated and form a monomer, as asserted by the authors. The authors' argument that hEdc3 is not a dimer is specious: sedimentation velocity–AUC by Ling et al., 2008 reveals a molecular mass of 94500 which is 15% smaller than the dimer (predicted to be 112,154 Da) but by no means is this magnitude of discrepancy between predicted and measured molar mass in SV–AUC unreasonable given the assumptions that go into fitting (such as partial specific volume, which will depend on protein conformation ––e.g. if the Lsm and YjefN domains are beads on a string or pack together to form a globular unit.) While Ling et al. noted that hEdc3 was somewhat aggregated in equilibrium AUC, there was no evidence of aggregation in the SV–AUC data, so I do not think Ling misinterpreted their AUC data. Note well, equilibrium AUC makes greater demands on sample stability than SV–AUC, because the former takes several days whereas the latter can be performed overnight: the aggregation observed in equilibrium AUC is simply a manifestation of a protein that is unstable over timescales of days. To be careful, Ling et all addressed whether human Edc3 is a monomer or dimer using size–exclusion chromatography, noting "full–length hEdc3 is a dimer at all concentrations tested. Again, these experiments are relatively quick (hours) compared to equilibrium AUC (days). Last, yeast two–hybrid experiments by Ling et al. on the budding yeast Edc3 are also consistent with a dimer. There are no data in the author's manuscript to suggest Edc3 is a monomer as part of the decapping mRNPs depicted in Figure 8, so like Reviewer 2, I suggest the authors take more care with their interpretation of dimeric and monomeric states of Edc3 (and Scd6 for that matter).

---

## [Author Response]

[Editors’ note: The authors appealed the original decision. What follows is the authors’ response to the first round of review.]

Reviewer #2:The authors should discuss any (growth) phenotypes of the strains they prepared. In case any of the deletions or truncations are functionally important I would expect that those strains have serious defects.

In contrast to the reviewer’s expectations, all our Dcp2 element mutants grow well, with doubling times similar to those of wild–type cells. We added the growth phenotypes of Dcp2 element mutants at lines 1017–1019 on pages 46–47.

Pull–down experiments between the different components would be able to complement the Y2H experiments. This is important as I wonder to what degree do the endogenous proteins in the Y2H experiments interfere with the results? E.g. is the WT Dcp2 present in the cells that address the behavior of specific Dcp2 version.

If the reviewer is recommending experiments to further confirm the existence of different decapping complexes, we totally agree. In fact, such experiments will be our major research undertaking for the next few years, but we believe that such experiments are far beyond the scope of this data–heavy manuscript. If the reviewer is suggesting that pull–down experiments be employed to confirm the binary interactions between different *dcp2* mutants and Upf1, Edc3, and Pat1 shown in Figures 1B, 2A, and 4A, we disagree because these experiments were all complemented by our genetic analyses of mRNA decay phenotypes, and our genetic data are consistent with our Y2H results.

With regard to the reviewer’s concerns about Y2H experiments, two points should be noted: (i) We have substantial expertise with Y2H analysis and have used this technique successfully to identify factors in mRNA decay and map their domain–domain interactions (He et al., 1996, 1997; He et al., 2013; He and Jacobson, 1995, 2015; He et al., 2014). Our results from Y2H analyses of mRNA decay factors were consistent with the results generated by co–IP from the Izaurralde group for both fly and mammalian systems, but showed significant discrepancies with the results of pull–down experiments from the Parker group. We previously discussed these discrepancies including the Dcp2–Dhh1(Decker et al., 2007) and Dcp1 or Dcp2–Pat1(Nissan et al., 2010) interactions (He and Jacobson, 2015). We believe that Y2H and co–IP are less prone to artifacts than pull–down experiments, at least with decay factor interactions. Thus, the suggested pull–down experiments could generate additional data, but would probably create unnecessary confusion, and (ii) Our Y2H experiments in Figures 1B, 2A, and 4A assessed direct interactions between different *dcp2* alleles and Upf1, Edc3, and Pat1, and the results from these experiments were very unlikely to be affected by the presence of wild–type Dcp2 in the tester strains, as Dcp2 does not form dimers and *dcp2* mutant alleles were overexpressed relative to endogenous Dcp2. Only when we were assessing indirect interactions that are bridged by Dcp2 (as in the experiments of Figures 5 and 6) would the presence of Dcp2 or its specific elements have significant effects on the final outcomes. For our Y2H experiments shown in Figures 5 and 6, all the tester strains contain just the endogenous *DCP2* and no extra copy of *DCP2* is involved at all*.* For all these reasons we did not change the text in response to this comment.

Finally, I would like to mention that it is often hard to link the text to the figures. The figures are very large and finding the exact gels/ bars quickly is not always easy.

We agree with the reviewer on this point. To ensure an extremely thorough study, we analyzed 50 different *dcp2* alleles, resulting in several panels that were large and busy. To address this problem, we have now added subtitles for the panels of Figures 5, 6, and 7.

Page 3: "Dcp2 is a 970–amino acid protein" That depends strongly on the organism, this statement should be refined.

We have replaced “yeast” with “*Saccharomyces cerevisiae*” at line 23 on page 3.

Page 3: "Edc3 was also originally thought …". Edc3 has also been reported to be important for the formation of processing bodies. This should be mentioned.

Our manuscript focuses on mRNA decay and processing bodies do not play a role in mRNA decay (Decker et al., 2007; Eulalio et al., 2007). Hence, no revisions were made to address this point.

Page 5: "Deletion of either the first (U1D1) or the second (U1D2) Upf1–binding motif had no discernible effect on Upf1's binding to Dcp2, but loss of both Upf1–binding motifs (U1D1–U1D2) eliminated Upf1 binding to Dcp2 (Figure 1B).". It appears that E3D–ID–U1D2 has an effect on the Dcp2–UPF interaction. Is that a real effect or not?

We believe that E3D–ID–U1D2 does have an effect on Dcp2–Upf1 interaction. However, this subtle effect was not caused by loss of U1_2_, but most likely by loss of E3. This explanation is based on the results presented in Figures 5A and 5B, showing that Edc3 enhances Upf1 binding to the U1_1_ element. Hence, no revisions were made to address this point.

Page 6: "However, loss of both Upf1–binding motifs caused approximately two– to three–fold increases in the levels of these transcripts". Based on the bar–graphs the effect appears less than two–fold in some cases and rarely more than two–fold.

The manuscript includes all the relevant quantitative data (see Figure 1—figure supplement 1, where the fold increases in *dcp2–U1D1–U1D2* cells are: 2.6±0.4 for *CYH2* pre–mRNA, 2.0±0.0 for *can1–100* mRNA, 1.6±0.4 for *ade2–1* mRNA, and 2.1±0.3 for *trp1*–*1* mRNA). These fold changes indicate that our description was accurate, and no changes were made to the text.

Page 7: The discussion of the Edc3 effects is very handwaving. For instance: "RPS28B mRNA is not likely degraded by an alternative pathway." Why not, there can be an alternative pathway that is completely independent of Edc3. Or: "…YRA1 pre–mRNA and Edc3 may play an additional role in the decay of this transcript." What roles of Edc3 would that be if it is not substrate recruitment, and why would that other role not be important for other mRNAs? Edc3 is known to enhance Dcp2 decapping efficiency in general.

We adjusted the wording at line 117 on page 7 to make our statement more precise.

Page 8:"The partial stabilization of Dhh1 substrates caused by loss of E3 can be explained similarly as described above for YRA1 pre–mRNA.". I am not sure what the authors mean. They should be more explicit.

We adjusted the wording at lines 135–137 on page 8 to make our description more explicit.

Page 8: "Edc3 binds to the conserved 17–amino acid segment." (E3–1). It is known that the Edc3 Lsm domain interacts with leucine rich motifs. Both E3–1 and E3–2 potentially contain such motifs (Figure S1). How do the authors explain then that only E3–1 is important? They could do some simple modeling to address this on a structural level.

We did consider how the 17–amino acid segment (E3–1) might engage the Edc3 Lsm domain and determined that our E3–1 element matches exactly the Edc3–bingding helix of *K. lactis* Dcp2 in the Dcp1–Dcp2–Edc3–m^7^–GDP structure solved by the Graille group (Charenton et al., 2016). We also identified the residues of E3–1 that are important for Edc3 binding. A similar motif is found in *S. pombe* Dcp2, but is interrupted by an insertion of eight amino acids. Because our manuscript focuses on Dcp2 element functions, we did not discuss structural aspects of E3–1, E3–2, and all other HLMs. That said, our new data in Figure 7 further confirmed that Edc3 binds only to E3–1 and that Scd6 can bind to both E3–1 and E3–2.

Page 14: " Deletion of EDC3 also caused additional stabilization of the Dhh1 substrates EDC1 and SDS23 mRNAs and the Pat1/Lsm1/Dhh1 substrates HSP12 and HXT6 mRNAs". Those effects appear very small to me. Is that statistically really relevant?

Those effects are indeed very small, but they are reproducible and statistically relevant (Figure 4E, Figure 4—figure supplement 5), leading us to believe they are correct and biologically relevant. These small effects can be easily explained by our multiple–step kinetic model. No changes were made to the text in response to this comment.

Reviewer #3:Yeast Dcp2 is a large protein, containing a structured N terminus (residues 1–245) and a disordered C–terminus comprised of 725 amino acids. Prior studies by Jacobson and colleagues indicate that deletion of the C–terminus of Dcp2 results in dysregulation of 1/3 of all protein–coding transcripts in yeast (He et al., eLife 2018). They also previously showed that the C–terminus of Dcp2 harbors binding sites for cofactors such as Edc3, Pat1 and Upf1 which act on different classes of transcripts during normal mRNA decay and quality control pathways such as NMD. (He and Jacobson RNA, 2015).In the present study, the authors perform an analysis of Dcp2 protein interactions with Edc3, Pat1, and Upf1 and find the binding sites are separable with respect to the effects of steady–state levels of specific transcripts in yeast: mutation of a single Edc3 binding site in the C–terminus of Dcp2 increases steady–state levels of YRA1 and RPS28B mRNA compared to the Edc3 gene deletion to varying extents; mutation of both UPf1 binding sites in Dcp2 C–terminus partially stabilizes NMD transcripts compared to the Upf1 gene deletion; mutation of the Pat1 binding sites in the C–terminus of Dcp2 does not affect steady–state levels Pat1 sensitive transcripts. The results suggest decapping is rate–limiting for decay of Edc3 and Upf1 transcripts but not for those sensitive to Pat1.The most significant finding in the manuscript is that Edc3 may play a broader role in mRNA decay than was previously appreciated.EDC3 was previously shown to regulate two transcripts in budding yeast, RPS28B and YRA1 (Badis et al., Mol Cell, 2004 ;Dong et al., Mol Cell, 2007). Surprisingly, Dhh1 substrates are upregulated when the Edc3 binding site on Dcp2 is mutated (E3D) but not when the EDC3 is deleted. This observation suggests one additional factor may bind the Edc3 binding site of Dcp2. Moreover, deletion of the EDC3 gene together with mutation in its binding site in Dcp2 causes additional stabilization of Edc3 sensitive YRA1 mRNA and, to a lesser degree, Dhh1 sensitive transcripts. These observations suggest an extra function for Edc3 outside of recruitment of Dcp2 to mRNA and provide additional evidence Edc3 is involved in decay of Dhh1 sensitive mRNAs.Additionally, the authors uncover a failsafe mechanism for control of decapping and 5'–3' decay. Partial increases in steady–state levels of Edc3 and Dhh1 sensitive transcripts caused by mutation of Edc3 binding site (E3D) can be restored to levels observed in Edc3 and Dhh1 deletion strains when Pat1 binding sites are eliminated.To understand the interplay between cofactors that bind the C–terminal domain of Dcp2, the authors used yeast two–hybrid analyses. The data suggest Edc3 is a core component of multiple decapping complexes that can promote interactions of the Dcp2 C–terminal domain with Upf1 and Xrn1 or antagonize its interactions with Pat1.Some open–ended questions for future studies include: (1) to identify the functional consequences of Dcp2–Xrn1 interactions for RNA degradation. It would be interesting to determine if there is an overlap in residues of Xrn1 required for its binding to Dcp2 with those shown to bind the 80S ribosome (Tesina et al., NSMB, 20219). (2) determine the identity of cofactors that work together with Edc3 to promote degradation of Dhh1 sensitive transcripts. One wonders if Scd6 and Edc3 work together in this regard. 3–to understand the impact on how binding site mutations in the C–terminal domain Dcp2 impact subcellular localization. Prior studies showed the Edc3 binding element (E3) promotes localization of Dcp2 to P–bodies (Xing et al., eLife, 2020); deletion of SCD6 and EDC3 promotes nuclear localization of an inactive form of Dcp2 (Tishinov, J. Sci. Sci. 2021).

The reviewer’s brief history of our research and his/her highlights of our manuscript’s most significant findings are concise and accurate. Further, the open–ended questions that ought to be addressed for future research are thought provoking, mechanistically relevant, and biologically important. We certainly intend to pursue them.

The manuscript would be suitable for publication in a venue such as eLife if the following concerns are addressed:1. The authors suggest one additional factor besides Edc3 is required for the decay of Dhh1 sensitive transcripts. Might this factor be Scd6? (re Jacobson and Hinnebusch, PLoS Genet 2018).

This reviewer is correct about Scd6. We have strong evidence supporting the involvement of Scd6 in the decay of Dhh1–sensitive transcripts: (i) Scd6 interacts with multiple decapping factors including forming direct interactions with Dcp2 and Pat1, and forming Dcp2 bridged interactions with Dcp1 and Edc3; (ii) Scd6 binds multiple elements on Dcp2 including the Edc3–binding element and the leucine–rich motifs; (iii) Scd6’s binding to Dcp2 requires three distinct pairs of binding motifs E3–1 and E3–2, L_3_ and L_5_, and L_8_ and L_9_, and the two motifs in each of these three binding motif pairs all have redundant Scd6 binding activities, suggesting Scd6 can bind to full–length Dcp2 either as a monomer or as a dimer with the dimer having a higher affinity, and each monomer engaging three different binding motifs, one from each of the three pairs; (iv) Edc3 promotes the joining of Scd6 into the decapping complex to form a Dcp1–Dcp2–Edc3–Scd6 complex; and (v) in the absence of Edc3, Pat1 promotes the joining of Scd6 to the decapping complex to form a Dcp1–Dcp2–Scd6–Pat1 complex.

These new results indicate that Scd6 can collaborate with either Edc3 or Pat1 in promoting mRNA decapping and that Scd6 likely exists in both Edc3–containing and non–Edc3–containing decapping complexes. Our current hypothesis is that the E3–1 and E3–2 motifs on Dcp2 promote the assembly of three distinct decapping complexes containing different dimeric forms of Edc3 and Scd6, i.e., Edc3:Edc3, Edc3:Scd6, and Scd6:Scd6. Decapping complexes containing Edc3–Edc3–Dhh1 or Edc3–Scd6–Dhh1 target Dhh1–sensitive transcripts. Decapping complexes containing Edc3–Scd6–Pat1 or Scd6–Scd6–Pat1 target the Pat1/Lsm1–sensitive transcripts. Partitioning Edc3, Scd6, and Pat1 into different decapping complexes explains the genetic redundancy observed for the Edc3–binding site and the inhibitory activity of Edc3 observed for Dcp1–Pat1 interaction (Figure 5A).

We added the Scd6 data as Figure 7, described the new Scd6 results under two separate subtitles in the Results section (lines 461 to 541, pages 21 to 25), and incorporated the Scd6 results into our model in Figure 8.

2. Related, if both Edc3 and Scd6 bind E3, the deletion of this element might result in relocalization of Dcp2 in the nucleus as described for the Edc3 and Scd6 double deletion strain (Tishinov and Spang, J Cell Sci 2021). To test, does mutation of Dcp2 NLS (K450T, described by Tshinov et al) reduce steady–state levels of RPS28b mRNA in the Dcp2 E3D background compared to Dcp2 E3D in isolation?

This reviewer is suggesting a very insightful experiment whose purpose is to address whether loss of the E3 element might cause Dcp2 localization to the nucleus. To address this question, this reviewer is asking us to construct a *dcp2* E3D–K450 double mutant and analyze its decapping activity by comparing it to a *dcp2* E3D mutant using the steady–state levels of *RPS28B* mRNA as an indirect readout, as decay of this mRNA is mostly decapping limited and may be sensitive to subtle changes in Dcp2 subcellular localization.

Although we have not done the precise experiment suggested by Reviewer #3, the answer to this question is likely to already be in our data. The NLS identified by Tishinov and Spang maps to Dcp2 residues 450–467. This NLS is two residues downstream of the Pat1–binding motif L_1_ (residues 443–447) and has a twelve–residue overlap with the first Upf1–binding U1_1_ motif (residues 456–475). The Dcp2 K450T mutation contains the NLS–inactivating changes in four key lysine residues (K460T, K461T, K463Tand K465T) that were eliminated by our Upf1–binding element deletion U1D1. To answer the reviewer’s question, it is necessary to compare the *RPS28B* mRNA levels in *dcp2–E3D* and *dcp2–E3D–U1D1* cells. As shown in Figure 1C and Figure 1—figure supplement 2, the *RPS28B* mRNA levels in *dcp2–E3D* and *dcp2–E3D–U1D1* cells are comparable. Compared to wild–type cells, fold increases for the mRNA are 1.9 in *edc3*∆ cells, 1.7 in *dcp2–E3D* cells, and 1.9 in *dcp2–E3D–U1D1* cells. Thus, in the E3D background, loss of the four key lysine residues of the NLS had no detectable effect on the level of *RPS28B* mRNA. From this negative result, we can draw two tentative conclusions: loss of E3 may not affect the nuclear localization of Dcp2, or the NLS identified in Dcp2 may not function as an NLS.

Interestingly, we noticed that, in the E3D background, loss of the Pat1–binding motifs L_1_ to L_8_ (LD1–8) or L_1_ to L_9_ (LD1–9) consistently yielded lower *RPS28B* mRNA levels. As shown in Figure 4B and Figure 4–Supplement 2, the *E3D–LD1–8* and *E3D–LD1–9* alleles yielded lower *RPS28B* mRNAs levels than the *E3D* allele. Compared to wild–type cells, the *RPS28B* mRNA had about 1.60–fold increases in *edc3*∆, 1.40 in *dcp2–E3D*, and 1.2 in *dcp2–E3D–LD1–8*, and 1.1 in *dcp2–E3D–LD1–9* cells. The effects accompanying loss of the Pat1–binding motifs on *RPS28B* mRNA levels were also evident in the experiments shown in Figure 4C and Figure 4–Supplement 2. Although this experiment did not include the *dcp2–E3* allele as a control, *edc3∆* cells serve as a proxy. *E3D–U1D2–LD1–9*, *E3D–U1D1–LD1–9*, or *E3D–U1D1–U1D2–LD1–9* alleles all yielded lower *RPS28B* mRNA levels than the *edc3∆* allele (respective fold increases of 1.1, 1.3, 1.2 vs 1.7). These results indicate that, in the E3D background, loss of the Pat1–binding leucine–rich motifs appear to result in increased cytoplasmic decapping activity, suggesting that the Pat1–binding leucine–rich motifs, but not the NLS, may actually control the nuclear import of Dcp2. This implies that Pat1 may promote Dcp2 nuclear localization under certain conditions, an idea consistent with a report that Pat1 is a shuttling protein (Teixeira and Parker, 2007). To reconcile our results with Tishinov and Spang’s observations, we propose that, in the absence of Edc3 and Scd6, the K450T mutation may abolish Pat1 binding to Dcp2 and thus block Pat1–mediated Dcp2 nuclear import.

We added the above results of the combined deletions of E3D and U1D1 or LD1–8 and L1–9 into the Results section under a subtitle “The Pat1–binding Leucine–rich Motifs May also Control the Nuclear Import of Dcp2” (lines 543 to 589, pages 25 to 27), and discussed the implication of these results in the Discussion section (lines 808 to 811 on page 36).

3. The yeast two–hybrid data presented in Figures 5 and 6 are dense and confusing. EDC3 deletion in tester strains has differential effects on interactions of the c–terminal domain of Dcp2 with Pat1, Upf1 and Xrn1. Deletion of Edc3 promotes binding of the C–terminal domain to Pat1 but inhibits binding to Upf1 and Xrn1. Is the former result from a competition between Edc3 and Pat1 binding? Is the latter from the ability of Edc3 to remodel the C–terminal domain of Dcp2 for recruitment of Upf1 and Xrn1?

We agree with the reviewer that our two–hybrid data in Figures 5 and 6 are dense and consequently confusing and we are open to any suggestion that might make our data presentation more effective. To make the specific question being tested in each experiment clear, we now added a title for each panel in Figures 5 and 6 and added similar titles to the components of our new Figure 7.

The reviewer’s interpretation that there are distinct effects of *EDC3* deletion on different molecular interactions is correct. Based on the data presented in our manuscript, we postulate that Edc3 is partitioned into two different pools. One pool exists in the Dcp1–Dcp2–Edc3 core decapping complex and another pool functions as a targeting component for the Edc3 and Dhh1 substrates. Edc3 in the targeting pool competes with Pat1 for binding to the Dcp1–Dcp2–Edc3 core complex. As discussed in our responses to comment #1, the E3–1 and E3–2 motifs likely promote the assembly of three distinct decapping complexes containing different dimeric forms of Edc3 and Scd6. Among these three dimeric Edc3 and Scd6–containing complexes, only the Edc3:Scd6 and Scd6:Scd6, but not Edc3:Edc3–containing decapping complexes, can recruit Pat1. The common requirement of the E3–1 and E3–2 motifs for the binding of these different Edc3 and Scd6 dimers and the distinct binding specificities of Pat1 for these dimer–containing decapping complexes explains our observed competition between Edc3 and Pat1. Our genetic data are consistent with the interpretation that Edc3 joining to the Dcp1–Dcp2 complex either remodels the C–terminal domain of Dcp2, or provides additional weak but specific binding surfaces for factors such Upf1, Scd6, and Xrn1 to promotes more efficient assembly of distinct decapping complexes.

We incorporated the new Scd6 results into our model in Figure 8 that now can easily explain the observed competition between Edc3 and Pat1. We also discussed the role of the core Edc3 component in the assembly of different decapping complexes (see lines 676–682 on pages 30–31).

4. In the discussion, manuscript p24, the authors state "Our genetic experiments challenge this proposed function for Edc3 and suggest that the core Edc3 component of each decapping complex may inhibit the enzymatic activity or substrate binding of the decapping enzyme, consistent with the existence of an inhibitory element in the Dcp2 C–terminus." Is there any evidence to suggest Edc3 inhibits decapping, as stated in the above sentence? If there are published data showing lesions in Edc3 decrease transcript levels, it should be cited otherwise the authors should include evidence of this claim.

This is good advice and “challenge” is probably not the appropriate word to use because it implies some certainty. We have changed the problematic sentence to “Since the Dcp2 C–terminal domain also contains an autoinhibitory element (He and Jacobson, 2015; Paquette et al., 2018), we speculate that the core Edc3 component of each decapping complex may function to enhance the autoinhibitory activity of Dcp2 before the decapping enzyme is finally targeted to specific mRNPs.” This revision is located at lines 668–671 on page 30.

We did not have direct evidence to suggest Edc3 inhibits decapping. Given the dual roles of Edc3 (i.e., functioning as a core component in different decapping complexes and as a targeting component of Edc3 and Dhh1substrate–specifc decapping complexes) that we proposed in our manuscript, it is difficult to assess the potential inhibitory function of Edc3 on mRNA decapping. However, in our published work (He and Jacobson, 2015), we did have indirect evidence indicating that Edc3, and possibly also Scd6, have the postulated inhibitory activities, as loss of the Dcp2 E3 element consistently leads to a more active decapping enzyme.

5. The report of Xrn1 binding to Dcp2 is preliminary, as there are no functional data to suggest binding of Xrn1 to Dcp2 is important for 5'–3' decay (Figure 6). What are the consequences of disrupting the Dcp2–Xrn1 interaction identified in this study on 5'–3' mRNA decay? As the authors note in discussion, Pat1 can also recruit Xrn1 to the decapping complex but in my opinion, it is difficult to conclude whether or not the newly identified Dcp2:Xrn1 interaction 'challenges the Pat1 coupling model' as stated on manuscript p28.

This, too, is good advice in that we still have much to learn about Xrn1 binding to Dcp2 and the role of that interaction in decapping regulation. Accordingly, we have rewritten the last two sentences of the paragraph, changing “challenge” to “suggest” and “speculate” as in the following: “Since Pat1 targets only a subset of yeast transcripts (He et al., 2018), and loss of the Pat1–binding motifs had no effect on all tested decapping substrates (Figure 2B) yet loss of the Edc3– and Upf1–binding motifs each yielded specific effects (Figure 1C), we suggest that Pat1–mediated coupling may be limited to a small number of mRNAs and may not make a significant contribution to the overall decay of these mRNAs. We speculate that the observed Pat1:Xrn1 interaction may not recruit Xrn1 to decapped transcripts, but instead may serve to dissociate Xrn1 from decapping complexes after Pat1–mediated decapping.” These revisions are located on lines 760–766 on page 34.

6. The model depicted in Figure 7 is one of several that are consistent with the data. Can they rule out the existence of multiple decapping complexes with Upf1, Pat1 and Edc3 linking Dcp2 to a core of an mRNP and substrate targeting is dictated by another RNP component (or components) that bind transfactors or RNA directly? It seems unlikely Dcp2 exist on its own in a cell as depicted in their model.

We think it is very unlikely that multiple decapping complexes with Upf1, Pat1, and Edc3 link Dcp2 to a core of an mRNP. The following points argue against the “Dcp2–linked multiple decapping complexes” model and strongly favor our “target–specific distinct decapping complexes” model:

First, the available evidence in the literature indicates that mRNA decapping mostly occurs on polyribosomes, a conclusion supported by clear experimental results: (i) decapped mRNAs are associated with translating ribosomes (Hu et al., 2010; Hu et al., 2009; Pelechano et al., 2015); (ii) the decapping regulators Upf1–3 (Atkin et al., 1997), Dhh1 (Sweet et al., 2012), Pat1 (Wyers et al., 2000), Lsm1 (Bonnerot et al., 2000), and Scd6 (Weidner et al., 2014) are all associated with polyribosomes; and (iii) the decay enzymes Dcp2 (our unpublished data) and Xrn1 are also mostly polyribosome–associated (Tesina et al., 2019).

Second, Upf1 and Pat1 both use structured domains (Upf1–CH and Pat1–C) to bind Dcp2. The major binding elements that we identified for these two factors, U1_1_ and L_1_ (Figure 5B), are just five residues apart and this would make the simultaneous binding of both factors to the same Dcp2 molecule very unlikely. We also tested for possible interactions between Upf1 and Pat1 or Scd6 in the two–hybrid system and found that they do not interact. In addition, the observed competition between Edc3 and Pat1 for Dcp2 binding (Figures 5A and 5B) argues against the “Dcp2–linked multiple decapping complexes” model, but strongly favors our “target–specific distinct decapping complexes” model.

Third, most decapping factors accumulate to some extent in P–bodies (Parker and Sheth, 2007). Localization of these decapping factors in P–bodies could potentially have some regulatory functions in mRNA decapping, but P–bodies are not the sites where mRNA decapping occurs (Decker et al., 2007; Eulalio et al., 2007). P–bodies are also not visible under normal growth conditions, but are only formed when cells are subjected to specific stress conditions or when the cellular decapping or 5’ to 3’ exonucleolytic decay activities are severely compromised (Parker and Sheth, 2007). The Parker and Rosen groups recently analyzed the composition and dynamics of yeast P–bodies (Xing et al., 2020) and all their results were generated from *dcp1∆* cells that completely lack decapping activity. Their observed associations of Dcp2, Edc3, Pat1, and Upf1 with P–bodies in *dcp1∆* cells are likely the consequence of the loss of decapping activity and are mostly irrelevant to the present study that was conducted under normal cellular and environmental conditions.

We also think that it is very unlikely that substrate targeting is dictated by another RNP component. In our 2015 RNA paper (He and Jacobson, 2015), we analyzed the interactions between Dcp2 and the decapping activators Edc3, Pat1, Lsm1, Dhh1, Upf1, Upf2, and Upf3. We found that Edc3, Pat1 and Upf1 bind directly to Dcp2 and that, in contrast, Lsm1, Dhh1, Upf2, and Upf3 do not bind to Dcp2 and thus are likely components of specific mRNPs. The targeting components in our model (Edc3, Upf1, and Pat1; Figure 8) are all RNA–binding proteins, and each of these factors also interacts directly with at least one specific mRNP component. Edc3 binds directly to specific RNA elements in *RPS28B* mRNA and *YRA1* pre–mRNA (Badis et al., 2004; Dong et al., 2010; He et al., 2014) and thus is itself a component of specific mRNPs; Edc3 self–associates and binds to Dhh1 (He and Jacobson, 2015); Pat1 binds to the Lsm1–7 complex (Bouveret et al., 2000; Sharif and Conti, 2013; Wu et al., 2014), and Dhh1 (Sharif et al., 2013); and Upf1 binds to Upf2 (He et al., 1997). Thus, the targeting components Edc3, Pat1, and Upf1 directly connect the decapping enzyme to specific mRNPs.

As noted above, Dcp2 contains an autoinhibitory element (residues 344–379). We recently investigated Dcp2’s 3D structure predicted by AlphaFold and found that the inhibitory element binds at the catalytic center of Dcp2 and likely can block its substrate binding. In addition, loss of the entire Dcp2 C–terminal domain yields a disregulated but still active decapping enzyme in vivo (He et al., 2018; He and Jacobson, 2015). Collectively, these observations and results lead us to believe that Dcp2 may be able to exist on its own and that assembly of the final decapping complexes may involve multiple sequential steps. Hence, we have not changed the text to address the reviewer’s question.

7. The model posits that Edc3 acts as a dimer on Edc3 and Dhh1 substrates but not on Pat1/Lsm1 or NMD substrates. I could not find data to support this. Do the authors have evidence that Edc3 acts as a monomer on the indicated transcripts? Biochemical structural and genetic data from Parker and Song labs indicate Edc3 is a dimer in solution and in yeast (Ling, MCB, 2008).

It is correct that biochemical, structural, and genetic data from the Parker and Song labs indicate that Edc3 is a dimer in both solution and in crystal structures (Ling et al., 2008). However, this conclusion was drawn from a human Edc3 C–terminal fragment (Residues 250–507), i.e., the YjeN domain, but not the full–length protein. This Edc3 fragment lacks the Lsm domain (residues 1–82) and the disordered FDF domain (residues 83–249). Ling et al. noticed that full–length Edc3 protein formed aggregates in solution and had a sedimentation coefficient smaller than that expected for a dimer. Our published genetic experiments also indicate that yeast Edc3 forms a dimer in vivo (He and Jacobson, 2015; He et al., 2014). However, in these experiments, Edc3 was overexpressed from a high copy number plasmid.

We also have strong genetic data indicating that Edc3 can function as a monomer and that Edc3 dimerization may subject to regulation. In our 2014 MCB paper (He et al., 2014), we analyzed the Edc3 requirement for decay of *YRA1* pre–mRNA and *RPS28B* mRNA. We found that the Edc3 Lsm domain alone is sufficient to promote efficient decay of *YRA1* pre–mRNA. In contrast, decay of RPS28B mRNA requires both the Lsm and YjeN domains of Edc3. Autoregulated *RPS28B* mRNA decay also requires the Rps28b protein. Rps28b binds to the RB motif (residues 201–231) located in the Edc3 FDF domain (residues 70–276) (He et al., 2014; Kolesnikova et al., 2013) and loss of the RB motif stabilizes the *RPS28B* mRNA (Kolesnikova et al., 2013). Interestingly, loss of the entire FDF domain including the RB motif promotes constitutive destabilization of *RPS28B* mRNA (He et al., 2014). These data indicate that the Edc3 FDF domain encodes additional regulatory activities. As *RPS28B* mRNA decay requires the binding of an Edc3 dimer to its decay–inducing element in the 3’–UTR (He et al., 2014), these genetic data are consistent with the idea that Rps28b regulates the dimerization of Edc3. In our unpublished data, we mapped the Dhh1–binding region of Edc3 to residues 71–196 of the FDF domain. This fragment of Edc3 binds weakly to Dhh1. However, the binding of the fragment to Dhh1 is greatly enhanced by the YjeN domain of Edc3, indicating that Dhh1likely binds to an Edc3 dimer in vivo.

We do not have experimental evidence indicating that Edc3 acts as a monomer on NMD and Pat1/Lsm1 substrates. We proposed that monomeric Edc3 is a component of the NMD and Pat1/Lsm1 substrate–specific decapping complexes because this Edc3 subunit functions as a shared core component of all decapping complexes. The targeting of the NMD and Pat1/Lsm1–specific decapping complexes to their respective substrates is controlled by the binding of Upf1 and Pat1 to their specific Dcp2 *cis* elements. We proposed that an Edc3 dimer is present in the Edc3 and Dhh1 substrate–specific decapping complexes because our results suggest that one monomer functions as a core component and another monomer functions as a targeting component. In this scheme targeting of the Edc3 and Dhh1–specific decapping complexes to their respective substrates is controlled by homodimerization of the targeting and core Edc3 components. Our new Scd6 data in Figure 7 indicate that Scd6 can collaborate with either Edc3 or Pat1 to form distinct decapping complexes, suggesting that the Edc3:Scd6 dimer–containing complex may also target the Dhh1 substrates, and that the Edc3:Scd6 or Scd6:Scd6 dimer–containing decapping complexes may target the Pat1/Lsm1 substrates (Figure 8).

We proposed that monomeric Edc3 is present in the NMD and Pat1/Lsm1 substrate–specific decapping complexes also because: (i) Dcp2 contains only one Edc3–binding motif mapped to a 17–amino acid fragment (E3–1) (Figure 1D) and this fragment can only engage one Edc3 monomer for binding in the *K. lactis* Dcp1–Dcp2–Edc3–m^7^–GDP structure (Charenton et al., 2016) and (ii) Edc3 and Upf1 or Pat1 co–occupy Dcp2 C–terminus (Figure 5A) and the joining of Upf1 and Pat1 to the decapping complexes is sensitive to Edc3 copy number, as endogenous Edc3 inhibits both Edc3–Upf1 and Edc3–Pat1 interactions in the two–hybrid assay (Figure 5C).

To clarify this issue, we included some of the information discussed above in the paper (see lines 784–795 on page 35).

8. In addition, Figure 7 has inconsistencies with other figures. The authors show Pat1 and Edc3 co–occupy the C–terminal domain of Dcp2 in Figure 5D but in the model depicted in Figure 7 this co–occupancy is omitted? If the authors do not think Edc3 and Pat1 co–occpy Dcp2 C–terminus to promote decay of Edc3 sensitive transcripts, then how does one explain the failsafe mechanism?

The original Figures 7 and 5D were correct. We omitted Pat1 in the Edc3 substrate–specific decapping complex because Edc3:Edc3, Edc3:Scd6,and Scd6:Scd6 appear to compete for binding to the E3–1 and E3–2 motifs for decapping complex formation as we described in our responses to comments 1 and 3 above. Scd6 binds directly to Pat1 (Figure 7C) and thus likely provides at least one additional binding site besides the L1–L5 motifs for Pat1 in the decapping complexes. Our hypothesis is that Pat1 can joining the Edc3:Scd6 or Scd6:Scd6 dimer–containing decapping complexes, but not the Edc3:Edc3 dimer–containing decapping complexes that Edc3–sensitive transcripts.

To explain the failsafe mechanism for Edc3–sensitive substrates, we proposed that these transcripts are normally decapped rapidly through a deadenylation–independent mechanism by the Dcp1–Dcp2–Edc3–Edc3 decapping complex (Badis et al., 2004; Dong et al., 2007). When this pathway is blocked by either *cis* or *trans* mutations, the transcripts proceed through the slower deadenylation–dependent pathway and are most likely decapped by the Dcp1–Dcp2–Edc3:Scd6–Pat1 or Dcp1–Dcp2–Scd6–Scd6–Pat1 complexes. The combined element deletions of E3D or E3D1 and LD1–8 or LD1–9 block the assembly of all three possible decapping complexes and thus cause significant stabilization of Edc3–sensitive substrates.

The new Scd6 results in Figure 7 and the adjusted model in Figure 8 support our explanations.

– Methods: "With a few exceptions, each northern blotting experiment was repeated independently at least two times. Bar graphs in relevant figures were generated by generated by GraphPad Prism 9, mostly using the average {plus minus} SEM data.". This description is not satisfactory at all. What are "few exceptions" (in a large number of the bar–graphs error–bars are lacking, which is not acceptable; a single blot does not seem to be an exception) and why are there exceptions at all (all experiments should be in triplicates)? And what was repeated, where those technical replicates or biological replicates? In summary, I would think that one needs to have at least 3 real replicates (independent growth of the cells, independent gel, blot and analysis). Two technical replicates (where the same blot was run twice with the same sample) is not very useful. Also, all gels should be available in full size in some supplement. The differences are small in many cases and this requires data and analysis that is much more rigorous as what is currently done.

The reviewer jumped to the last paragraph of the methods section and exaggerated the meaning of our words “a few exceptions.” There are an extraordinary number of northern blots in our paper (157 blots in total), and only 16 out of 157 (about 10%) of the blots did not have error bars. The 16 graphs that do not have error bars were all presented in our initial phenotypic analyses of *dcp2* element mutants (Figures 1C and 2B). Eight blots (*RPS28B*, *EDC1*, *SDS23*, *HXT6*, *HSP12*, *LSM3*, *BUR6*, and *DIF1* mRNAs) in Figure 1C did not have replicates (Figure 1—figure supplement 2). However, the phenotypic analyses for each of these substrates in the relevant *dcp2* element mutants were independently repeated in our subsequent experiments shown in Figures 1D, 3B, 4B, and 4E. Likewise, eight blots (*HSP12*, *can1–100*, *ade2–1*, *RPS28B*, *EDC1*, and *SDS23* mRNAs, and *CYH2* and *YRA1* pre–mRNAs) in Figure 2B did not have replicates (Figure 2—figure supplement 1C). However, the phenotypic analyses for six out eight of these substrates in relevant *dcp2* element mutants were independently repeated in our subsequent experiments (Figure 4B). In short, only two mRNAs in Figure 2B, *HSP12* and *ade2–1,* did not have repeat experiments. These two mRNAs were added to Figure 2B as “bonuses” and could certainly be eliminated from the figure without affecting any of our conclusions. Further, in the experiments of Figure 2B, deletion of the leucine–rich elements has no effect. From a genetics perspective each of the individual mRNA substrates comprises an independent experiment which thus has thirteen independent repeats. Our data for this experiment are thus extremely strong even without error bars.

In sum, our analyses did follow rigorous standards, with three or more independent biological replicates. This is now noted specifically in the methods section of the paper (lines 1136 to 1151 on page 52). Clearly, we believed that experimental repeats were critically important as most changes in mRNA levels that we observed in the *dcp2* element mutants were small, but biologically significant. In fact, it was because of these small changes that we were able to infer the rate–limiting steps for different decapping substrates and the likely existence of upstream functions for different decapping activators implemented before decapping enzyme recruitment. Finally, we did note that source data associated with our northern blotting figures were deposited in the Dryad repository (https://datadryad.org/stash), and within that site can be found at doi:10.5061/dryad.pc866t1px.

– The argumentation in the manuscript assumes that specific substrates are degraded by specific mechanism. I am not sure if that is really the case. A substrate is likely degraded by a combination of multiple mechanisms. For some substrates the decay rate might be slightly enhanced by one factor, for other substrates another factor might slightly shift the use of the different pathways. Often the authors just assume that substrates are only degraded in a specific manner, which is not true.

We disagree. We did not make any assumptions in the manuscript. We made the *dcp2* element mutants, analyzed their respective decay phenotypes, interpreted the data, and then proposed a new model for mRNA decapping that was more consistent with our results than with previous models. Our data clearly indicate that the Edc3 binding element promotes Edc3 binding to Dcp2 and enhances the decapping of both Edc3 and Dhh1 substrates. Similarly, the two Upf1–binding motifs promote Upf1 binding to Dcp2 and enhance the decapping of NMD substrates. Our inference of different decay pathways for different decapping substrates was exclusively based on specific results, not assumptions.

– The authors also assume that the motifs in Dcp2 have a single function, either Edc3 binding, Upf1 binding or Pat1 binding. Based on previous studies, it is however clear that Edc3 can also bind to HLMs (that the authors refer to as Pat binding motifs only). Furthermore, the region around the motif that the authors call E3 has also been shown to have an inhibitory effect on the Dcp2 enzyme (e.g. 10.1093/nar/gky233). This effect has not been discussed in detail. Deletion of such elements can of course change mRNA levels, independent of any binding to activators. Deletion of a motif can thus have multiple effects (lower recruitment of a factor and increase/ decrease of activity through direct Dcp2 inhibition).

We disagree. We never assumed that “the motifs in Dcp2 have a single function” and in fact, we were eager and excited to identify new functions for different Dcp2 elements because these new functions may give us the hints about potential genetic redundancy or unusual regulatory activities. Our analyses of the Edc3–binding element provide proof that we did not have any prior assumptions. We observed that *cis* deletion of the Edc3–binding element, but not *trans* deletion of the *EDC3* gene causes selective stabilization of Dhh1 substrates. Based on this observation, we inferred that the Edc3–binding motif likely has a second function and promotes the binding of another non–Edc3 factor (indeed Reviewer #3 thought that this non–Edc3 factor might be Scd6, and our new data in Figure 7 indicate that Scd6 indeed binds to the E3–1 motif). Our results indicate that the two Upf1–binding motifs promote Upf1 binding to Dcp2 and enhance the specific decapping of NMD substrates and that leucine–rich motifs L_1_–L_5_ mostly promote Pat1 binding and can enhance the specific decapping of Dhh1 substrates under some conditions. These are experimental results, not assumptions. In fact, we are still actively searching for additional factors that may bind to each of the Dcp2 elements.

With respect to Edc3–binding to HLMs, this reviewer was undoubtedly referring to early published work from the Sprangers and Izaurralde groups for *S. pombe* Dcp2 (Fromm et al., 2012). Using in vitro pull–down assays and NMR titration techniques, the authors of that paper identified four HLMs that bind to the Lsm domains of both Edc3 and Scd6. In their experiments, the Edc3 Lsm domain appears to have stronger binding to these HLMs than that of Scd6. While these in vitro observations likely still hold, in vivo evidence for either Edc3 or Scd6 binding and the role of such binding in mRNA decapping for each of these HLMs is still lacking. We refer the nine HLMs from *S. cerevisiae* Dcp2 as Pat1 binding motifs because they bind to Pat1 both in vivo (He and Jacobson, 2015a) and in vitro (Charenton et al., 2017), and each of these HLMs does not bind Edc3 (He and Jacobson, 2015a). Based on these facts, we refer to these HLMs as Pat1–binding motifs, and not Edc3–binding motifs. It would make no sense to name these *S. cerevisiae* Dcp2 HLMs as Edc3–binding motifs because similar *S. pombe* Dcp2 HLMs bind to Edc3 since the available evidence indicates that *S. pombe* decapping is more similar to that of mammalian systems than to that of *S. cerevisiae*. For example, *S. pombe* also contains an Edc4 homolog that is present in mammalian systems, but absent in *S. cerevisiae* (Wang et al., 2013). Simply put, decapping regulation in *S. pombe* and *S. cerevisiae* has sufficient significant differences that render expectations of similarities between the two systems to be unwarranted. Our new data in Figure 7 confirmed that Edc3 binds solely to E3–1, but Scd6 can bind to multiple motifs in Dcp2 including E3–1, E3–2, L_3_, L_5_, L_8_, and L_9_.

With respect to the Edc3–binding motif and its nearby inhibitory activity, this reviewer was referring to published work from the Gross group, again for *S. pombe* Dcp2 (Paquette et al., 2018). Based mostly on in vitro kinetic analysis of different Dcp1–Dcp2 complexes, Paquette et al. identified two Dcp2 segments that manifest inhibitory activities and named these segments as IM1 and IM2. In *S. pombe* Dcp2, IM1 is located immediately downstream of the second HLM, one of the Edc3–binding motifs identified in vitro by the Sprangers and Izaurralde groups as described above. IM1 is similar to the inhibitory element (IE) that we identified in *S. cerevisiae* Dcp2 (He and Jacobson, 2015a). We did not discuss the inhibitory activities of *S. pombe* Dcp2 in our paper because we felt it was not relevant to our results. In *S. cerevisiae* Dcp2, the Edc3–binding element of Dcp2 is 81 amino acids away from the inhibitory element. In our experiments, two deletions generated around the Edc3 binding region (E3Dand E3D1) did not touch any of the residues in the inhibitory element, but both caused selective stabilization of the Edc3 and Dhh1 substrates. We did not observe additional effects for these two deletions.

– Page 4: "Edc1 and Edc2 were isolated as high–copy suppressors …. but they do not appear to be required for mRNA decapping in vivo. ". Deletion of Edc1 and Edc2 impairs mRNA decay (e.g. 10.1093/genetics/157.1.27). Based on that they are required for in vivo mRNA decapping. Please clarify.

The reviewer was referring to published work from the Parker lab (Dunckley et al., 2001). In that paper, it was shown that deletion of *EDC1* or *EDC2* had no effect on decapping of the reporter *MFA2* mRNA in otherwise wild–type cells, but can have an effect on mRNA decapping in strains compromised for decapping activity. “Edc1p and Edc2p are not rate limiting for mRNA decay” and “The *edc1∆* and *edc2∆* slow mRNA decapping in strains compromised for decapping activity” are two paragraph titles from the Results section of that paper. Reviewer #2’s statement that “Deletion of Edc1 and Edc2 impairs mRNA decay” is incorrect, i.e., it does not reflect the findings of the original paper.

– Figure 5: Why is there an interaction between UPF1 and Dcp1 in the DelteEdc3 background in Figure 5A (row3, column2), but this interaction is not there in Figure 5B (row1, column4).

The reviewer failed to understand our data. We clearly stated that “deletion of *EDC3* diminished and loss of the entire C–terminal domain of Dcp2 eliminated Upf1:Dcp1 interaction”. Upf1:Dcp1 interactions were documented in both Figure 5A (row 3, column 2) and Figure 5B (row 1, column 4). The color in Figure 5B (row 1, column 4) is simply less intense because Edc3 has been lost.

– Page 5: "Wild–type (WT) Dcp2 exhibited a strong two–hybrid interaction with Upf1". This has been shown before: doi.org/10.1371/journal.pone.0026547. And that manuscript should thus be referenced.

The reviewer was referring to a paper published by the Parker group (Swisher and Parker, 2011). This paper showed that Upf1 binds to the catalytic domain of Dcp2 (aa 102 to 300) and that this Dcp2:Upf1 interaction is bridged by Edc3. Our results in the present manuscript show that Upf1 binds directly to Dcp2 through two distinct sequence elements located in the C–terminal domain of Dcp2. We have previously discussed these discrepancies (He and Jacobson, 2015a) and we do not see any value to cite this reference. It should be noted Dcp2–Upf1 interaction was originally identified in 1995 in our initial yeast two–hybrid screen using Upf1 as a bait. At the time we named the gene as NMD1 (He and Jacobson, 1995). Our more recent work showed that Dcp2–Upf1 interaction is mediated by two independent Upf1–binding motifs in Dcp2 (He and Jacobson, 2015a). We did not feel we needed a citation here.

– Page 6 "These increases were much smaller in magnitude than those caused by deletion of UPF1, which usually led to >10–fold increases for these transcripts.". This finding strongly suggests that additional UPF1 binding motifs must be present in Dcp2, or that Dcp2 does not play an important role in NMD. The authors should be more specific here, and not only mention that "decapping is maybe not rate–limiting" (what would then be rate limiting?) or by mentioning that "other pathways could be important" (which ones?). In the end, it raises the question if the Dcp2:Upf1 interaction is biologically important.

Here the reviewer is doing his/her best to ignore very important and provocative results in favor of simplistic notions of what should be going on in decapping of NMD substrates. We searched for 30 years by multiple approaches, only found two Upf1–binding motifs in Dcp2, and no additional Upf1–binding motifs. Why, then, does deletion of these binding sites not lead to mRNA stabilization equivalent to the loss of Upf1 itself? The answer most likely lies in a multistep NMD pathway that includes more than one function for Upf1. Our current models for NMD (He and Jacobson, 2015b) suggest that completion of the NMD pathway requires at least three steps, namely recognition and dissociation of a prematurely terminating ribosome, mRNA decapping, and 5’ to 3’ exonucleolytic decay (in that order). If in this pathway ribosome dissociation is the rate–limiting step, and if Upf1 functions at both ribosome dissociation and recruitment of the decapping enzyme, then loss of Upf1 will make the ribosome dissociation step even slower, markedly decrease overall decay rates of NMD substrates, and cause strong accumulation of these mRNAs. However, if loss of the two Upf1–binding motifs in Dcp2 only slows down the decapping step but not ribosome dissociation, then the overall decay rate will only be slightly reduced, resulting in weak accumulation of NMD substrates. In thinking about this explanation, it’s important to remember that many biological pathways have both rate–limiting and non–rate–limiting steps. Clearly, the decapping step could become rate–limiting if the activity of the decapping enzyme is greatly reduced, e.g., under certain cellular or environmental conditions. In short, our observation that loss of the two Upf1–binding motifs causes 2–3–fold selective stabilization of NMD substrates, provides important and novel insights into the functioning of the NMD pathway, and we do not question the biological importance of Dcp2–Upf1 interaction.

– Page 7: The discussion of the Edc3 effects is very handwaving. For instance: "RPS28B mRNA is not likely degraded by an alternative pathway." Why not, there can be an alternative pathway that is completely independent of Edc3. Or: "…YRA1 pre–mRNA and Edc3 may play an additional role in the decay of this transcript." What roles of Edc3 would that be if it is not substrate recruitment, and why would that other role not be important for other mRNAs? Edc3 is known to enhance Dcp2 decapping efficiency in general.

Here the reviewer gets several facts wrong and reveals that he/she is not up to date on the relevant literature. Our discussion of the Edc3 effects is not handwaving, but is based on solid genetic principles of unique functions and functional redundancies that operate in all biological systems. These principles led us to formulate our hypothesis, design follow–up experiments, and uncover the backup decay pathway for both Edc3 and Dhh1 substrates. The reviewer’s question (“why would that other role not be important for other mRNAs?”) clearly indicates that he/she is not familiar with important questions in the field.

With regard to the in vivo function of Edc3, published work from the Jacquier group as well as ours clearly established that Edc3 functions as a transcript–specific decapping activator, as deletion of *EDC3* only caused stabilization of two transcripts in the entire transcriptome, *RPS28B* mRNA and *YRA1* pre–mRNA (Badis et al., 2004; Dong et al., 2007). The two transcripts targeted by Edc3, *RPS28B* mRNA and *YRA1* pre–mRNA, are very unusual and both contain specific Edc3–binding elements. *RPS28B* mRNA contains an Edc3–binding element in its long 3’–UTR (Badis et al., 2004; He et al., 2014) and *YRA1*–pre–mRNA contains two Edc3–binding elements in its large intron (Dong et al., 2010; Dong et al., 2007). Although both transcripts are targeted by Edc3, our experiments indicate that the decay mechanisms for these two transcripts are largely distinct (He et al., 2014). Further, based on genetic analyses of the Edc3–binding element presented in our current manuscript, Edc3 may only have one role in *RPS28* mRNA decay, namely to recruit the decapping enzyme. In contrast, Edc3 likely has two roles in *YRA1* pre–mRNA decay, recruitment of the decapping enzyme and repressing translation or modifying the mRNP.

It has been suggested by the Parker group that Edc3 functions as a general mRNA decapping activator (Kshirsagar and Parker, 2004), but this proposed general function of Edc3 does not fit with the absence of any effect of *EDC3* deletion on any mRNAs other than *RPS28B* mRNA and *YRA1* pre–mRNA. Indeed, the Parker group did not see any effect of *EDC3* deletion on their reporter *MFA2* mRNA. Hence, the big question to emerge from results demonstrating Edc3’s transcript–specific effects has been “is Edc3 really a general decapping activator?” In short, to the best our knowledge, there is no evidence in the literature indicating that *S. cerevisiae* Edc3 enhances Dcp2 decapping efficiency in general. That said, we should note that the experiments in our manuscript did uncover additional functions for Edc3, but not that of a general decapping activator (see below).

– Page 7: "…E3 also controls targeting of the decapping enzyme to Dhh1–regulated mRNAs". Is there any evidence that the E3 motif can directly interact with the Dhh1 helicase? Does E3 motif e.g. contain an FDF–like motif? Furthermore, it is known that Edc3 and Dhh1 directly interact. as the authors note and is it thus not most likely that Dhh1 is recruited to Dcp2 with the E3 motif and Edc3? This could be tested by changing the FDF motif in Edc3.

Here, too, the reviewer appears to lack familiarity with the relevant literature or to have misunderstood the data presented in our manuscript. First, in our earlier published work, we demonstrated that Dhh1 does not bind Dcp2 in vivo (He and Jacobson, 2015a). In the same paper, we also showed that additional factors such Upf2, Upf3, and Lsm1 also do not bind Dcp2. Second, the data we presented in Figures 5A and B clearly show that Dhh1 does not bind Dcp2 directly and that Dhh1 joins the decapping complex through Edc3. The latter is substantiated by our demonstration that loss of Edc3 eliminates Dhh1:Dcp1 interaction.

– Page 9: "These results indicate that in the context of full–length Dcp2, leucine–rich motifs L1 to L5 control the selective binding of Pat1 to Dcp2, most likely with a contribution from each motif, in contrast to the proposed mode for Pat1 binding to Dcp2 based on structural data (Charenton et al., 2017).". I disagree. First: based on the data the interaction between Dcp2 and Pat1 appears to require around 4 HLMs. Those can be either 1–4 (as in LD9–5) or 5–9 (as in LD1–4). The L1 to L5 are not more important than the others. Second, the Pat1:Dcp2 structure clearly shows that one HLM motif interacts with the C–terminal domain from Pat1, forming a 1:1 complex. Avidity effects can than result a more efficient recruitment of Pat1 to Dcp2 when multiple HLMs (any random ones) are present in Dcp2. The off–rates are just slower when more motifs are present. The data is thus fully consistent with the structure.

We disagree with this reviewer on both points. First, our data in Figure 2A clearly show that full–length Dcp2 lacking the first five motifs (L_1–5_), but containing the last four L_6–9_ motifs (*LD1–5* allele) does not bind Pat1. This indicates that the last four L_6–9_ motifs have no Pat1 binding activity. This is a straightforward observation and it is consistent with our C–terminal element deletion analysis showing that loss of last four L_6–9_ had no effect on Pat1–binding (*LD9–6* allele), indicating that the last four L_6–9_ motifs are not required for Pat1 binding activity. We made our conclusion based on these two facts and can’t rationalize an alternative conclusion based on avidity theory. Second, it is true that the Pat1–Dcp2 structural work from the Graille group (Charenton et al., 2017) showed that “one HLM motif interacts with the C–terminal domain from Pat1, forming a 1:1 complex.” However, this reviewer should know how the Graille group generated the Pat1–Dcp2 crystals. Based on our understanding of their paper, the Graille group used a C–terminal fragment of Pat1 (residues 435 to 796) and incubated it individually with short Dcp2 peptides of HLM2, 3, and 10 motifs. They never tested whether the Pat1 C–terminal fragment may bind to two or more different HLM motifs. Whether the full–length Pat1 may have additional HLM–binding regions is also a big question for us. Third, we also have yeast two–hybrid Pat1–Xrn1 interaction data indicating that the Pat1 C–terminal domain is inefficient for Xrn1 binding, suggesting the Pat1–Xrn1 HLM interaction observed in crystals by the Graille group may not occur in vivo or on full–length proteins.

– Figure 2—figure supplement 1: Why does deletion of Pat or Lsm1 stabilize the YRA1 and RPS28B mRNAs? These effects are on the same level as the effects that the authors discuss in the other figures and based on that I assume that they are real. However, the bar–graphs again have no error–bars.

This claim by the reviewer is totally false. Deletion of Pat1 or Lsm1 does not cause stabilization *YRA1pre–mRNA* and *RPS28B* mRNA. Rather, we see a slight destabilization of these two transcripts that is consistent with our model for competition between different decapping complexes (in Figure 2—figure supplement 1: when compared to the corresponding levels in wild–type cells, *YRA1* pre–mRNA showed a level of 0.6 in both *pat1∆* and *lsm1∆* cells, and *RPS28B* mRNA also showed a level of 0.6 in both *pat1∆* and *lsm1∆* cells).

– Page 9: "…one possible explanation for this surprising observation is that decapping is not rate–limiting". Do these mRNAs contain strong secondary structure elements in the 5' end that would slow down Xrn1? Xrn1 is normally very processive and rapid and thus not rate limiting. What could be rate limiting if not decapping?

The reviewer has missed our point, namely that the rate limiting step in decay of these mRNAs is upstream of decapping, not downstream of decapping. And, no, the mRNAs do not contain strong secondary structures.

– Page 9: "…Pat1 performs an unidentified major function upstream…". Please be more specific in what that could be. This is unsatisfactory.

Our manuscript is largely structured on principles of genetics to define genes or their functions. A term like “an unidentified major function upstream or downstream” is standard in genetics.

– Page 11: "This indicates that these decapping substrates can all be decapped by an alternative route when the normal Dcp2 cis–element–mediated active recruitment of the decapping enzyme is blocked.". This plainly shows that Dcp2 can be recruited to the mRNA by alternative manners, which is not surprising. In my opinion the network of interactions is very redundant and deleting one interaction (e.g. through the removal of a Dcp2 motif) is backed up by the other interactions. Pat1 and Dhh1 interacts with Dcp1 directly for instance, maybe that is the default route and the routes via the Dcp2 C–terminal IDR might be less important. In that light, it has also been shown that the complete Dcp2 C–terminal region can be deleted without causing large in vivo effects.

In this comment the reviewer made one assumption and several other points that he/she treated as facts. Both the assumption and the “facts” are false. The reviewer stated that “Pat1 and Dhh1 interacts with Dcp1 directly for instance”, but, to the best our knowledge, there are no data in the literature indicating that *S. cerevisiae* Pat1 and Dhh1 interact with Dcp1 directly. The data in this manuscript (Figure 5A) clearly established that Pat1 and Dhh1 both have no direct interaction with Dcp1. Further, the reviewer stated that “the complete Dcp2 C–terminal region can be deleted without causing large in vivo effects.” This statement, too, is false. As evidence, here’s title of the first paragraph from our 2018 *eLife* paper: “Elimination of the large Dcp2 C–terminal domain causes significant changes in genome–wide mRNA expression” (He et al., 2018). To be precise, that study demonstrated that 1530 transcripts were significantly affected by deletion of the large Dcp2 C–terminal domain. As for the assumption (“This plainly shows that Dcp2 can be recruited to the mRNA by alternative manners”), our published work indicated that decapping of some mRNAs does not require enhanced recruitment of Dcp2 at all (He and Jacobson, 2015a). For these mRNAs, decay is unaffected even by loss of the entire Dcp2 C–terminal domain and Pat1, Lsm1, Dhh1, and Edc3. These data, as well as our observations of the decay behaviors for Pat1/Lsm1 substrates, can be easily explained by a kinetic model without evoking network interactions.

– Page 12: "combining the leucine–rich element deletions LD1–8, LD1–9, LD9–3, and LD9–2 with the Edc3–binding element deletions E3D or E3D1 caused additional substantial stabilization of the Edc3 substrate YRA1 pre–mRNA". This merely reflects that fact that Edc3 can interact with the HLMs and the Pat can interact with the HLM that the authors call E3 motif here.

Once again, the reviewer failed to recognize facts and made false assumptions. Our earlier Dcp2 element mapping experiments clearly demonstrated that Edc3 and Pat1 bind to specific motifs in Dcp2 (He and Jacobson, 2015a). Edc3 binds to the E3–1 element and Pat1 binds to the L_1–9_ elements. Our Dcp2 element deletion analyses in Figures 1A, 2A, and 4A further established the binding specificity of Edc3 and Pat1. As for the assumptions, the results with three *dcp2* alleles proved them wrong, e.g., (i) “Edc3 can interact with the HLMs.” Please examine the *E3D* and *E3D1* alleles in Figure 1B. These two alleles have all nine L_1–9_ motifs intact, yet show no Edc3–binding at all. Thus, Edc3 does not bind HLMs; (ii) “Pat can interact with the HLM that the authors call E3 motif here.” Please examine the *U1D1–UD2–LD1–9* allele in Figure 4B. This *dcp2* allele has an intact E3 element, but lacks all other elements, yet it has no Pat1–binding at all. Thus, Pat1 does not bind the HLM that we named E3.

In contrast to the reviewer’s erroneous assumptions and interpretations, our genetic data clearly suggest that decay of Edc3 and Dhh1 substrates occurs in ordered, multiple–step processes and that Dhh1 and Pat1 function at different steps of the mRNA decay pathway.

– Page 12: "..cells harboring E3D1 consistently had lower transcript levels than those harboring E3D for Edc3 and Dhh1 substrates ". I really don't see that. HA–dcp2–E3D and HA–dcp2–E3D1 are the same for instance (Figure 4—figure supplement 2, YRA1 and RPS28 mRNAs). Maybe I am looking at the wrong thing, it is sometimes hard to find the corresponding graph.

The reviewer was looking at the wrong *dcp2* alleles. Here we are analyzing the genetic relationships between the Edc3–binding element and the Pat1–binding elements L_1–9_ in Figure 4B and Figure 4—figure supplement 2. The relevant *dcp2* alleles are those containing the combined deletions of *E3D* or *E3D1* and *LD1–8*, *LD1–9*, *LD9–2*, or *LD9–3*, but not the *dcp2* alleles containing single element deletions of E3D or E3D1. Our described differences are clear in both the northern blots shown in Figure 4B and the bar graphs shown in Figure 4—figure supplement 2. The mRNA decay phenotypes of the single element deletions of E3D and E3D1 were already established in Figures 1C,1D, and 3B.

– Page 12: "Similarly, cells harboring LD9–3 also had consistently lower transcript levels than those harboring LD9–2 for both Edc3 and Dhh1 substrates, suggesting that LD9–3 deletion maintains more function of Dcp2 than that of LD9–2 in decay of these mRNAs." Again, I don't see that, the bars are the same to me.

Again, the reviewer was looking at the wrong *dcp2* alleles. The described differences are evident in both the northern blots shown in Figure 4B and the bar graphs shown in Figure 4—figure supplement 2.

– Page 13: " raising the possibility that Pat1/Lsm1 substrates can be decapped without the function of any decapping activators.". Or there are just additional interactions that recruit Dcp2 to the mRNA and that are independent of the Dcp2 C–terminal region. The Dcp2 C–terminal region is not required and only adds some additional functionality.

Although the reviewer suggests alternative interpretations for our observation, we think it is most easily explained by a multiple–step kinetic model.

– Page 14: "Deletion of EDC3 caused additional 7–8–fold stabilization of the Edc3 substrate YRA1 pre–mRNA in E3D or E3D1 cells". Really, maybe I am again looking at the wrong graphs, but in my interpretation the levels change from around 3 (E3D or E3D1) to around 10 (when DeltaEdc3 is there too). That is a 3.5 fold stabilization. Seem that the authors exaggerate the effect.

We did not exaggerate the effect. The reviewer ignored our explanation for the quantitative difference. We always compare the deletion mutant cells to wild–type cells and, here, the *cis*–element deletion causes 3–fold increases relative to wild–type whereas the double deletions cause 10–fold increases relative to wild–type. Hence, the additional net effect of deleting *EDC3* is 10–3=7–fold.

– Page 17 "assembly of the Dcp1–Dcp2–Edc3–Upf1 decapping complex". Is this a new finding? It was known that Edc3 and Upf1 interact (10.1371/journal.pone.0026547). Why is this (direct) Edc3–Upf1 interaction not shown in 5D?

The reviewer counters our result by citing an older paper from the Parker group in which it was suggested that Edc3 and Upf1 interact well (Swisher and Parker, 2011). However, we have previously refuted the Parker lab Edc3–Upf1 result (He and Jacobson, 2015a) and, in the same paper, established that Edc3, Pat1, and Upf1 bind to distinct elements of Dcp2. The experiments we presented in Figures 5 and 6 were specifically designed to dissect the basis of several molecular interactions that are bridged by Dcp2, i.e., interactions that are not direct. Bridged interactions between Edc3 and Upf1 were analyzed in Figures 5A and 5C and from those data we clearly stated in the Results section that “These results indicate that Dcp2 bridges an interaction between Upf1 and Edc3.” We are very troubled that this reviewer did not appear to understand our concepts, our experimental design, our results, and our conclusions.

– Page 17: "Interestingly, deletion of EDC3 enhanced Pat1:Dcp1 interaction". The most logical explanation is that Edc3 also binds to the HLMs. Deletion of Edc3 thus makes these HLMs available for the interaction with Pat1. The complete discussion in the paper seems too complex to explain this.

This reviewer is proposing a new explanation for our data and deems it to be the most logical explanation. However, our experimental evidence proved that his/her explanation is wrong. As shown in Figure 1B, we have nine *dcp2* alleles lacking the E3 element but containing all nine intact HLMs (*E3D*, *E3D–ID*, *E3D–U1D1*, *E3D–U1D2*, *E3D–ID–U1D1*, *E3D–ID–U1D2*, *E3D–U1D1–U1D2*, *E3D–ID–U1D1–U1D2*, and *E3D1*). None of these *dcp2* alleles binds to Edc3, a result indicating that Edc3 does not bind to the HLMs. As shown in Figure 5B, in cells harboring the *dcp2–N300* allele with an intact E3 element (and two HLMs), Pat1 does not promote the assembly of any decapping complexes in the absence Edc3. Thus, Pat1 does not bind to the E3 element in the absence of Edc3.

– Page 19:"suggesting that Xrn1 binding to Dcp2 may be dependent on a specific Dcp2 conformation.". The only logical explanation is that a sequence in the C–terminal region is inaccessible in a specific Dcp2 conformation. As many structures of Dcp2 are known this should be addressed in more detail. Currently the explanation is too handwaving.

The reviewer’s proposed explanation is exactly the same as ours, yet when he/she presents it the explanation is the “only logical explanation” and ours is “too handwaving.” Further, the reviewer’s statement “many structures of Dcp2 are known” is inaccurate. To the best our knowledge, the only known Dcp2 structures are from *S. cerevisiae*, *S. pombe*, and *K. lactis* and these structures principally address just the catalytic domains and lack the regulatory domains. For *S. cerevisiae*, the catalytic domain contains 245 amino acids, and the regulatory domain contains 745 amino acids. The entire regulatory domain of Dcp2 appears disordered and no real structures are available. This includes the most recent results from AlphaFold.

– Page 20:"We suspect that the observed Xrn1:Edc3 interaction may involve dimerization of exogenous Edc3 with endogenous Edc3 bound to Dcp2 in a decapping complex. ". Edc3 is always a stable dimer. Why would that only play a role here and not in the discussions above?

The reviewer’s statement “Edc3 is always a stable dimer” has no definitive support in the literature. Edc3 contains three domains (Lsm, FDF, and YjeN), of which the Lsm and YjeN domains form defined structures and the FDF region is mostly disordered. Published work from the Parker and Song groups (Ling et al., 2008) demonstrated that a truncated human Edc3 YjeN domain (residues 250–507) could form dimers in both solution and in crystals. And full–length human Edc3 formed aggregates in solution and had a sedimentation coefficient smaller than that expected for a dimer (Ling et al., 2008). However, whether full–length Edc3 is a monomer, dimer, or both in vivo is unknown. In our functional analysis of Edc3, we showed that monomeric Edc3 can promote *YRA1* pre–mRNA decay in vivo (He et al., 2014). We consider it likely that Edc3 dimer formation is regulated in vivo and have preliminary evidence indicating that Dhh1 and the Rps28b protein may carry out that regulation.

– Page 22:"a new direct role of Edc3 in selective targeting of the decapping enzyme to Dhh1–regulated mRNAs. ". This appears trivial to me, as Dcp2:Edc3 and Edc3:Dhh1 interactions have been structurally described in detail (as the authors also write), so nothing new in my eyes.

This reviewer is completely wrong on this point. Our finding that Edc3 controls the selective targeting of the decapping enzyme to Dhh1 substrates is one of the most significant discoveries in our manuscript. This finding broadens the list of in vivo Edc3–regulated substrates from just two transcripts to more than one thousand (to be precise, 1098 transcripts) (He et al., 2018) and establishes how Dhh1 substrates are decapped in vivo. In our opinion, and that of Reviewer #3, the identification of new substrates and delineation of a new decapping mechanism are very significant mechanistic advances. Unfortunately, Reviewer #2 used fragmented and misleading information to diminish the importance of our finding as “nothing new in my eyes.” Let’s get the facts straight. First, in several model systems, biochemical or structural data show that Dcp2:Edc3 (Charenton et al., 2016; Fromm et al., 2012; Mugridge et al., 2018) and Edc3:Dhh1 (Sharif et al., 2013) interactions occur in vitro. However, in most model systems, the cellular substrates for each of these factors are unknown. Without knowing the in vivo substrates for these factors, the biological significance of these interactions always remains an important unanswered question (but apparently not for the reviewer). Second, in essentially all the in vitro structural studies, either isolated domains or small peptides were used. As noted above, this also leads to the question of whether the observed binding or activities may still hold for the full–length proteins, not even to mention whether the observed Dcp2:Edc3 and Edc3:Dhh1 in vitro interactions be all be applied in vivo.

– Page 23: "Edc3–binding motif can promote assembly of at least one additional decapping complex, a Dcp1–Dcp2–Edc3–Edc3 complex….". That is trivial and can be extended to the known observation that Edc3 can bridge two Dcp2 proteins. In that manner Edc3 thus supports the formation very large complexes that contain many Dcp2 proteins that can all independently recruit factors.

Defining the composition of different decapping complexes is critically important for an understanding both the functions and the mechanisms of actions of mRNA decay factors. The reviewer used a baseless claim (“the known observation that Edc3 can bridge two Dcp2 proteins”) and a biased opinion (“In that manner Edc3 thus supports the formation very large complexes”) to dismiss our finding. We’re pretty sure that the reviewer would be unable to provide a reference showing that yeast “Edc3 can bridge two Dcp2 proteins.”

– Page 24:"both as a common core component of multiple decapping complexes and as a unique targeting component of specific decapping complexes provides a unified theory for explaining the apparently contradictory proposed functions for Edc3,". This, in my opinion, is generally accepted in the literature and not novel.

The reviewer made another baseless claim here. Our proposition of potential dual functions of Edc3 in mRNA decay is indeed novel and we would challenge the reviewer to provide us with a reference in which similar propositions have been put forward before our manuscript.

– Page 28:" distinct multi–component decapping complexes". I am not sure if these really exist. The interactions are all weak and in a cellular environment constant rearrangements of the interactions will take place. Also, the decapping complexes are not isolated but form higher order complexes that are, when large enough. visible as P–bodies and that contain many Dcp2 proteins. The provided model is thus not relevant in my eyes as it oversimplifies things too much.

This reviewer’s uncertainty as to whether distinct multi–component decapping complexes may really exist suggests that he/she did not examine our data carefully or did not understand our data. Using two baseless claims combined with his/her biased assumptions, the reviewer blindly rejected our fact–based model as “not relevant in my eyes” or “oversimplifies things too much.” There’s no evidence in the literature showing that “the interactions are all weak” or that “the decapping complexes are not isolated but form higher order complexes.” Contrary to the reviewer’s baseless claims or assumptions, many of the pertinent interactions are strong with nM affinities (Dutta et al., 2011; Webster et al., 2019), and several distinct decapping complexes or subcomplexes have already been purified from different biological systems (Bouveret et al., 2000; Fenger–Gron et al., 2005). We consider it appalling that a reviewer for *eLife* would fail to use objective criteria and use baseless claims to dismiss our model. At a minimum we would have expected his/her critique to justify his/her dismissal of our work by finding fault in our experimental designs, finding specific data to be unconvincing for reasons other than a few missing error bars, documenting why specific interpretations are wrong or which conclusions are not solid.

The authors address how specific mRNA decapping components arrange into larger complexes and how this influences mRNA decapping/ degradation. To that end, the authors have used a tour–de–force experimental approach and have performed a very large number of Y2H experiments and northern blots.– I feel that the data is not always very strong (lack of error–bars).

We disagree. Our data are strong and follow rigorous standards. We have independent biological replicates and the mRNA decay phenotypes for each of the key *dcp2* mutants were all confirmed with replicates in several independent experiments. There are an extraordinary number of northern blots in our paper (157 blots in total), and only 16 out of 157 (about 10%) of the blots did not have error bars. The 16 graphs that do not have error bars were all presented in our initial phenotypic analyses of *dcp2* element mutants (Figures 1C and 2B). Eight blots (*RPS28B*, *EDC1*, *SDS23*, *HXT6*, *HSP12*, *LSM3*, *BUR6*, and *DIF1* mRNAs) in Figure 1C did not have replicates (Figure 1—figure supplement 2). However, the phenotypic analyses for each of these substrates in the relevant *dcp2* element mutants were independently repeated in our subsequent experiments shown in Figures 1D, 3B, 4B, and 4E. Likewise, eight blots (*HSP12*, *can1–100*, *ade2–1*, *RPS28B*, *EDC1*, and *SDS23* mRNAs, and *CYH2* and *YRA1* pre–mRNAs) in Figure 2B did not have replicates (Figure 2—figure supplement 1C). However, the phenotypic analyses for six out eight of these substrates in relevant *dcp2* element mutants were independently repeated in our subsequent experiments (Figure 4B). In short, only two mRNAs in Figure 2B, *HSP12* and *ade2–1,* did not have repeat experiments. These two mRNAs were added to Figure 2B as “bonuses” and could certainly be eliminated from the figure without affecting any of our conclusions. Further, in the experiments of Figure 2B, deletion of the leucine–rich elements has no effect. From a genetics perspective each of the individual mRNA substrates comprises an independent experiment which thus has thirteen independent repeats. Our data for this experiment are thus extremely strong even without error bars.

In sum, our analyses did follow rigorous standards, with three or more independent biological replicates. This is now noted specifically in the methods section of the paper (lines 1136 to 1151 on page 52). Clearly, we believed that experimental repeats were critically important as most changes in mRNA levels that we observed in the *dcp2* element mutants were small, but biologically significant. In fact, it was because of these small changes that we were able to infer the rate–limiting steps for different decapping substrates and the likely existence of upstream functions for different decapping activators implemented before decapping enzyme recruitment.

– I fear that the hypothesis that is made (one element interacts with one binding partner; one mRNA is degraded in one manner) is not always correct and additional interactions that can take have not been considered.

We disagree. The binding elements that we identified in Dcp2 were not based on a hypothesis, but were based on solid, published experimental evidence (He and Jacobson, 2015a). Edc3 binds to one element, Upf1 binds to two elements, and Pat1 binds multiple elements, and these are all facts, not theoretical suggestions. Our finding that Scd6 also binds to the Edc3–binding motif (E3–1) (Figure 7E) comprises further evidence that the notion is not hypothetical*.* We did not and would never hypothesize that Edc3 binds to multiple elements if there were no data to support the claim. Our observations that decapping of NMD substrates is enhanced by the Upf1 binding elements and that decapping of both Edc3 and Dhh1 substrates is enhanced by the Edc3 binding element are also facts, not hypotheses. Further, we did not hypothesize that “one mRNA is degraded in one manner.” Rather, we actively searched for the alternative decay pathways for individual mRNAs in different *dcp2* element mutants and dedicated two whole Figures (Figures 3 and 4) to this effort. We also found an alternative decay pathway for both Edc3 and Dhh1 substrates when their Edc3–mediated decay is blocked (Figure 4B).

– The final model that the authors propose does not contain many new aspects and neglects known aspects.

We disagree. Three functions have been proposed for decapping activators, including repressing translation, activating the decapping enzyme, and sensing codon optimality. Based on the experimental data presented in our manuscript, we proposed that some decapping activators control the targeting specificity of the decapping enzyme and do so by forming distinct decapping complexes. These are new functions and a new mechanism for decapping activators. Contrary to what was implied by the reviewer, we did not neglect known aspects of higher order complexes as found in P–bodies. We chose not to discuss them because these aspects were irrelevant to our paper. First, P bodies occur under stress conditions and when cellular decapping or 5’ to 3’ exonucleolytic activities are severely compromised while our experiments were all carried out under normal cellular and growth conditions. Second, P–bodies may sequester some biomolecules or inhibit some enzymatic activities, but available evidence indicates that P–bodies accumulate only a small proportion of decay factors from the cytoplasm (Leung et al., 2006; Xing et al., 2020). Third, it is now well established that P bodies are not required for mRNA decay (Decker et al., 2007; Eulalio et al., 2007). Fourth, we favor a model of co–translational decapping for most mRNAs because: (i) decapped normal and nonsense–containing mRNAs are associated with polyribosomes (Hu et al., 2010; Hu et al., 2009; Pelechano et al., 2015); (ii) most decay factors, including Xrn1 (Tesina et al., 2019), Dhh1 (Sweet et al., 2012), and Pat1/Lsm1 (Bonnerot et al., 2000; Wyers et al., 2000) are associated with polyribosomes; and (iii) our recent unpublished data also indicate that the vast majority of Dcp2 is associated with polyribosomes.

–Unfortunately, I feel that the data is not always very strong. I fear that the hypothesis that is made (one element interacts with one binding partner; one mRNA is degraded in one manner) is not correct and additional interactions that can take have not been considered. In that light I have a large number of remarks that show I often disagree with the drawn conclusions or that I don't see much novelty. In that light I think that the paper is not suitable for publication in eLife, but would be a better fit for a more specialized journal (e.g. RNA).

We disagree. This reviewer has ignored the results in our paper that appear to contradict his/her preconceptions about decapping activators and the general mechanism of mRNA decapping.

References

Atkin, A.L., Schenkman, L.R., Eastham, M., Dahlseid, J.N., Lelivelt, M.J., and Culbertson, M.R. (1997). Relationship between yeast polyribosomes and Upf proteins required for nonsense mRNA decay. J Biol Chem *272*, 22163–22172.

Badis, G., Saveanu, C., Fromont–Racine, M., and Jacquier, A. (2004). Targeted mRNA degradation by deadenylation–independent decapping. Mol Cell *15*, 5–15.

Bonnerot, C., Boeck, R., and Lapeyre, B. (2000). The two proteins Pat1p (Mrt1p) and Spb8p interact in vivo, are required for mRNA decay, and are functionally linked to Pab1p. Mol Cell Biol *20*, 5939–5946.

Bouveret, E., Rigaut, G., Shevchenko, A., Wilm, M., and Seraphin, B. (2000). A Sm–like protein complex that participates in mRNA degradation. EMBO J. *19*, 1661–1671.

Charenton, C., Taverniti, V., Gaudon–Plesse, C., Back, R., Seraphin, B., and Graille, M. (2016). Structure of the active form of Dcp1–Dcp2 decapping enzyme bound to m7GDP and its Edc3 activator. Nat Struct Mol Biol *23*, 982–986.

Decker, C.J., Teixeira, D., and Parker, R. (2007). Edc3p and a glutamine/asparagine–rich domain of Lsm4p function in processing body assembly in *Saccharomyces cerevisiae*. J Cell Biol *179*, 437–449.

Dong, S., Jacobson, A., and He, F. (2010). Degradation of YRA1 Pre–mRNA in the cytoplasm requires translational repression, multiple modular intronic elements, Edc3p, and Mex67p. PLoS Biol *8*, e1000360.

Dong, S., Li, C., Zenklusen, D., Singer, R.H., Jacobson, A., and He, F. (2007). YRA1 autoregulation requires nuclear export and cytoplasmic Edc3p–mediated degradation of its pre–mRNA. Mol Cell *25*, 559–573.

Eulalio, A., Behm–Ansmant, I., Schweizer, D., and Izaurralde, E. (2007). P–body formation is a consequence, not the cause, of RNA–mediated gene silencing. Mol Cell Biol *27*, 3970–3981.

He, F., Brown, A.H., and Jacobson, A. (1996). Interaction between Nmd2p and Upf1p is required for activity but not for dominant–negative inhibition of the nonsense–mediated mRNA decay pathway in yeast. Rna *2*, 153–170.

He, F., Brown, A.H., and Jacobson, A. (1997). Upf1p, Nmd2p, and Upf3p are interacting components of the yeast nonsense–mediated mRNA decay pathway. Mol Cell Biol *17*, 1580–1594.

He, F., Celik, A., Wu, C., and Jacobson, A. (2018). General decapping activators target different subsets of inefficiently translated mRNAs. *ELife 7*.

He, F., Ganesan, R., and Jacobson, A. (2013). Intra– and intermolecular regulatory interactions in Upf1, the RNA helicase central to nonsense–mediated mRNA decay in yeast. Mol Cell Biol *33*, 4672–4684.

He, F., and Jacobson, A. (1995). Identification of a novel component of the nonsense–mediated mRNA decay pathway by use of an interacting protein screen. Genes Dev *9*, 437–454.

He, F., and Jacobson, A. (2015). Control of mRNA decapping by positive and negative regulatory elements in the Dcp2 C–terminal domain. RNA *21*, 1633–1647.

He, F., Li, C., Roy, B., and Jacobson, A. (2014). Yeast Edc3 targets RPS28B mRNA for decapping by binding to a 3' untranslated region decay–inducing regulatory element. Mol Cell Biol *34*, 1438–1451.

Hu, W., Petzold, C., Coller, J., and Baker, K.E. (2010). Nonsense–mediated mRNA decapping occurs on polyribosomes in *Saccharomyces cerevisiae*. Nat Struct Mol Biol *17*, 244–247.

Hu, W., Sweet, T.J., Chamnongpol, S., Baker, K.E., and Coller, J. (2009). Co–translational mRNA decay in *Saccharomyces cerevisiae*. Nature *461*, 225–229.

Kolesnikova, O., Back, R., Graille, M., and Seraphin, B. (2013). Identification of the Rps28 binding motif from yeast Edc3 involved in the autoregulatory feedback loop controlling RPS28B mRNA decay. Nucleic Acids Res.

Ling, S.H., Decker, C.J., Walsh, M.A., She, M., Parker, R., and Song, H. (2008). Crystal structure of human Edc3 and its functional implications. Molecular and cellular biology *28*, 5965–5976.

Nissan, T., Rajyaguru, P., She, M., Song, H., and Parker, R. (2010). Decapping activators in *Saccharomyces cerevisiae* act by multiple mechanisms. Mol Cell *39*, 773–783.

Paquette, D.R., Tibble, R.W., Daifuku, T.S., and Gross, J.D. (2018). Control of mRNA decapping by autoinhibition. Nucleic Acids Res *46*, 6318–6329.

Parker, R., and Sheth, U. (2007). P bodies and the control of mRNA translation and degradation. Mol Cell *25*, 635–646.

Pelechano, V., Wei, W., and Steinmetz, L.M. (2015). Widespread Co–translational RNA Decay Reveals Ribosome Dynamics. Cell *161*, 1400–1412.

Sharif, H., and Conti, E. (2013). Architecture of the Lsm1–7–Pat1 complex: a conserved assembly in eukaryotic mRNA turnover. Cell Rep *5*, 283–291.

Sharif, H., Ozgur, S., Sharma, K., Basquin, C., Urlaub, H., and Conti, E. (2013). Structural analysis of the yeast Dhh1–Pat1 complex reveals how Dhh1 engages Pat1, Edc3 and RNA in mutually exclusive interactions. Nucleic Acids Res.

Sweet, T., Kovalak, C., and Coller, J. (2012). The DEAD–box protein Dhh1 promotes decapping by slowing ribosome movement. PLoS Biol *10*, e1001342.

Teixeira, D., and Parker, R. (2007). Analysis of P–body assembly in *Saccharomyces cerevisiae*. Molecular Biology of the Cell *18*, 2274–2287.

Tesina, P., Heckel, E., Cheng, J., Fromont–Racine, M., Buschauer, R., Kater, L., Beatrix, B., Berninghausen, O., Jacquier, A., Becker, T.*, et al.* (2019). Structure of the 80S ribosome–Xrn1 nuclease complex. Nat Struct Mol Biol *26*, 275–280.

Weidner, J., Wang, C., Prescianotto–Baschong, C., Estrada, A.F., and Spang, A. (2014). The polysome–associated proteins Scp160 and Bfr1 prevent P body formation under normal growth conditions. Journal of Cell Science *127*, 1992–2004.

Wu, D., Muhlrad, D., Bowler, M.W., Jiang, S., Liu, Z., Parker, R., and Song, H. (2014). Lsm2 and Lsm3 bridge the interaction of the Lsm1–7 complex with Pat1 for decapping activation. Cell Res *24*, 233–246.

Wyers, F., Minet, M., Dufour, M.E., Vo, L.T., and Lacroute, F. (2000). Deletion of the PAT1 gene affects translation initiation and suppresses a PAB1 gene deletion in yeast. Mol Cell Biol *20*, 3538–3549.

Xing, W., Muhlrad, D., Parker, R., and Rosen, M.K. (2020). A quantitative inventory of yeast P body proteins reveals principles of composition and specificity. *ELife 9*, e56525.

[Editors’ note: what follows is the authors’ response to the second round of review.]

Please note that rather than asking Reviewer 2 to review your Response Letter and Revised Manuscript, we instead asked Reviewer 3 to comment on your assessment of and response to Reviewer 2's criticisms. As you will see from Reviewer 3's comments, outlined below for your attention, the manuscript has been improved but there are some remaining issues, including a few issues raised by Reviewer 2, that need to be addressed.In addition to their formal comments below, Reviewer 3 also asked me to informally pass on an additional piece of constructive criticism. Specifically, Reviewer 3 found the tone of your response to be unnecessarily antagonistic, which they felt was not fully warranted. In addition, Reviewer 3 thought that, as written, the original manuscript was unfairly dismissive of much of the previous biochemical work and suspects that this dismissiveness might have impacted Reviewer 2's response to the original manuscript as well as the overall review process. Below are Reviewer 3's formal comments, which I would ask you to address in a final Response Letter and Revised Manuscript.Reviewer #3 (Recommendations for the authors):The manuscript by Feng He et al. is an important contribution to the field–not only for mRNA decay but also for the area of intrinsically disordered proteins and proteins that have intrinsically disordered regions like Dcp2. It is quite challenging to reconstitute and resolve such proteins, and Feng He and colleagues have done a textbook job of mapping binding to short linear motifs (using Y2H) then assigning function to the motifs/binding partners in decay by a combination of genetic mutations and analyses of steady mRNA levels by Northern Blot. This type of analyses is 'gold standard' for moving from descriptive to truly mechanistic studies.The authors have addressed most of my concerns, adding additional data to the manuscript, including an interaction analysis of Dcp2 with Scd6 and a discussion of the impact of E3 motif deletions on Dcp2 localization.After considering the original submission, the revised manuscript and rebuttal I recommend publication in eLife. Some of the points in the revised manuscript require clarification, and some of the comments of Rev 2 should be addressed before the manuscript is accepted.1. On p30 line 693 the authors speculate the Edc3 core component of each decapping complex may enhance the autoinhibitory activity of Dcp2. This implies Edc3 promotes ––instead of alleviates–– autoinhibition, contrary to what is implied in Feng et al. 2015 and Paquette et al., 2018. My question remains, what evidence is there that Edc3 is an inhibitor of decapping?

We do not have direct evidence indicating that Edc3 is an inhibitor of decapping. However, we do have evidence indicating that the Edc3–binding motif has some inhibitory activity. In our model, we partition Edc3 into two different populations. One fraction functions as a core component of multiple decapping complexes and the other functions as a targeting component of specific decapping complexes. We hypothesized that the core Edc3 component in decapping complexes functions to enforce Dcp2 autoinhibition and that this autoinhibition is released at targeted mRNPs through Edc3 dimerization. Our in vivo experiments from our 2015 RNA paper do not contradict this model, but more experiments are needed to provide definitive proof. Paquette et al.’s in vitro experiments from their 2018 NAR paper showed that Edc3 alleviates Dcp2 autoinhibition. However, they used a four–fold molar excess of Edc3 relative to the decapping enzyme. The enhancement effects observed in these in vitro experiments could still result from Edc3 dimerization. Since we do not have actual experimental evidence for Edc3 as an inhibitor of decapping, we eliminated the problematic sentences from line 683 on page 30 to line 687 on page 31 of the original main text. In addition, to be more cautious, we also eliminated the sentence from line 779 to 781 at the end of Page 34 and the beginning of Page 35, in which we had speculated on the role of Pat1:Xrn1 interaction in mRNA decay.

2. Related on page 25 lines 563–565, the authors suggest "Scd6 can bind full–length Dcp2 as a monomer, or dimer with the dimer having higher affinity and each monomer binding three different motifs, one from each of three pairs. " Is there literature indicating Scd6 can form a dimer? Why do the authors rule out that a single Scd6 can bind multiple motifs through its Lsm domain using allovalency, as elaborated by Klein, Pawson and Tyers (PMID: 14521832) and reviewed in PMID: 28597296. In my mind, and that of Rev 2, there are multiple models for how Pat1, Scd6 and Upf1 bind multiple sites in the Dcp2 C–terminal domain which require validation with biophysical methods beyond the scope of the authors manuscript. I suggest the authors present the simplest model consistent with their data, and mention that they cannot rule out other models. Nailing down the binding stoichiometry of cofactors for Dcp2 for a given class of mRNA would require biophysical methods beyond the scope of the present manuscript.

There is no literature indicating that Scd6 can form a dimer. However, typical Lsm domain proteins do form higher order ring assemblies. We hypothesized that the Scd6 Lsm domain may form a binding–induced dimer in decapping complexes, as its two binding sites in Dcp2 (E3–1 and E3–2) are so close to each other. The binding patterns of Scd6 and Pat1 to full–length Dcp2 in the binary two–hybrid assays are complex and we struggled to generate a consistent model to explain the binding pattern for each factor. For Scd6, we derived a model in which three different regions of Scd6 engage three distinct Dcp2 binding motifs in a cooperative manner. This model explains our data reasonably well. For Pat1, we did not have a good model that can explain our observations. The binary two–hybrid data in Figure 2A suggest that two or more regions of Pat1 may engage two or more Dcp2 binding motifs, and thus can be explained by the avidity model suggested by Reviewer #2. However, Pat1 binding to decapping complexes shown in Figures 5B and 5C suggests that a single region of Pat1 engages a single Dcp2 binding motif. We hadn’t considered an allovalency model for both Scd6 and Pat1 binding to Dcp2 and are certainly open to any model that can better explain our data. We think that allovalency explains the Pat1 binding pattern of Dcp2 reasonably well and appreciate the suggestion. The allovalency model may also provide an alternative explanation for Scd6 binding to Dcp2. We have modified the text description for Pat1 and Scd6 binding to Dcp2, and now discuss the potential modes of action for Pat1 and Scd6 engagement with Dcp2. The pertinent changes to the manuscript are on page 9, lines 170 to 175, and page 25, lines 548 to 555.

3. I appreciate the authors looking into whether or not deletion of the E3 motif affects decapping by altering the localization of Dcp2 but am not convinced that the data in their manuscript convincingly address this question. For example, they show steady–state levels of RPS28B mRNA increase by E3D but are not affected by the double mutation containing E3D and U1D1 (which "eliminates 12 of 18 residues of the NLS, including the four critical lysine residues changed in the K450T mutation" described by Tishnov and Spang). On manuscript p26 line 594 the authors tentatively conclude loss of E3 may not affect the nuclear localization of Dcp2 or the NLS identified in Dcp2 may not function as an NLS. Is it possible that there is an increase in cytoplasmic Dcp2 containing U1D1 and E3D mutations compared to Dcp2–E3D but no effect on decapping (due to lack of binding to Edc3 sensitive transcripts)? Because Tishinov and Spang directly monitor nuclear localization by fluorescence, whereas the authors of the present manuscript did not, I would suggest toning down the conclusion that the "NLS in Dcp2 identified (by Tishnov and Spang) is not an NLS." Likewise, if they think K450T abrogates nuclear localization of Dcp2 by abrogating binding of Pat1, then they ought to show that with a binding experiment, such as Y2H. Either the section should be re–written with more explicit mention of the caveats associated with inferring nuclear localization from steady–state mRNA levels, compared to direct methods to quantify localization by fluorescence microscopy as reported by Tishinov and Spang or the discussion of nuclear localization (p 25 lines 567 to p27 line 613) should be removed from this otherwise dense manuscript.

Our manuscript focuses on the targeting mechanisms of the yeast decapping enzyme. While Dcp2 nuclear localization is interesting and potentially important for decapping regulation, we think that a discussion of Dcp2 nuclear localization in this manuscript is a distraction from the overall message in the paper. In addition, the data we presented on Dcp2 nuclear localization raised new issues that need further experimentation. Hence, we chose to take the advice of this reviewer and have removed the entire discussion of Dcp2 nuclear localization (originally pages 25–27, lines 557 to 603). We also eliminated the last sentence in the Discussion that addressed Dcp2 nuclear localization (page 36, lines 823 to 826).

As a scientific aside, our proposal that Pat1 mediates Dcp2 nuclear location may have some merit. We believe that cis deletions of the Edc3 and Scd6 binding motifs in Dcp2 and trans deletions of Edc3 and Scd6 have different effects on the composition of Pat1–containing decapping complexes and thus may have different effects on Dcp2 nuclear localization. Scd6 interacts directly with Pat1 (Figure 7). In Dcp2 cis E3 deletion mutant cells, Scd6 may still associate with Pat1 in the decapping complex and this association could mask a surface of Pat1, potentially its nuclear localization signals (hypothetical). In contrast, in trans EDC3 and SCD6 double deletion mutants, Pat1 associates with Dcp2 in the decapping complex and the Scd6 binding surface of Pat1 is likely exposed. The exposed surface of Pat1 could be the target of nuclear localization. In addition, the major NLS identified by Tishinov and Spang mapped to the first Upf1–binding site U1_1_ and Upf1 binding to this site could also block nuclear import receptor binding to Dcp2. In yeast cells, Upf1 outnumbers Dcp2 (molecules/per cell: 6905 for Upf1 vs 5761 for Dcp2) and Upf1’s inhibitory activity on the NLS–mediated nuclear import of Dcp2 could be very significant.

4. The authors have responded reasonably to Rev 2 's comments they deem 'appropriate'. In particular, I agree with the authors that there are an extraordinary number of northern blots (157) and 90% of them have error bars. The major concern of this reviewer about rigor is unfair.

We thank this reviewer for taking on the added responsibility of assessing the lengthy arguments in our responses to the comments from reviewer #2.

5. The authors' responses to Rev 2's comments that they deem are 'wrong' or confused are well–founded with the exception of two points that warrant further discussion or modification. (i) The authors show in full–length Dcp2 , leucine–rich motifs L1 to L5 control selective binding to Pat1 to Dcp2, most likely with a contribution from each motif, in contrast to the proposed mode of Pat1 binding to Dcp2 based on structural data (Charenton et al., 2017)." I agree with Reviewer 2 that this effect could be due to avidity, or if only one binding site on Pat1 engages a single motif (L1,L2, L3, L4 or L5) at a time , through allovalency (see point 2 above). The authors may discuss how multiple sites L1–L5 could collaborate to bind a single pocket on Pat1, as described with the crystal structure, consistent with allovalency. (ii) There is a disagreement about whether full–length Edc3 always forms a dimer as asserted by Rev 2, or if in cells it may be regulated and form a monomer, as asserted by the authors. The authors' argument that hEdc3 is not a dimer is specious: sedimentation velocity–AUC by Ling et al., 2008 reveals a molecular mass of 94500 which is 15% smaller than the dimer (predicted to be 112,154 Da) but by no means is this magnitude of discrepancy between predicted and measured molar mass in SV–AUC unreasonable given the assumptions that go into fitting (such as partial specific volume, which will depend on protein conformation –e.g. if the Lsm and YjefN domains are beads on a string or pack together to form a globular unit.) While Ling et al. noted that hEdc3 was somewhat aggregated in equilibrium AUC, there was no evidence of aggregation in the SV–AUC data, so I do not think Ling misinterpreted their AUC data. Note well, equilibrium AUC makes greater demands on sample stability than SV–AUC, because the former takes several days whereas the latter can be performed overnight: the aggregation observed in equilibrium AUC is simply a manifestation of a protein that is unstable over timescales of days. To be careful, Ling et all addressed whether human Edc3 is a monomer or dimer using size–exclusion chromatography, noting "full–length hEdc3 is a dimer at all concentrations tested. Again, these experiments are relatively quick (hours) compared to equilibrium AUC (days). Last, yeast two–hybrid experiments by Ling et al. on the budding yeast Edc3 are also consistent with a dimer. There are no data in the author's manuscript to suggest Edc3 is a monomer as part of the decapping mRNPs depicted in Figure 8, so like Reviewer 2, I suggest the authors take more care with their interpretation of dimeric and monomeric states of Edc3 (and Scd6 for that matter).

Again, we thank reviewer #3 for assessing the lengthy arguments in our responses to reviewer #2’s comments.

As for point (i): As described above in our response to item 2, we originally thought that the avidity model suggested by reviewer #2 could partially explain the binary two–hybrid data (Figure 2A), but appeared to contradict Pat1’s binding pattern to the decapping complex shown in Figures 5B and 5C. We now think that the allovalency model is a better fit for the Pat1 binding pattern of Dcp2. As noted in item 2 (above), we now discuss this binding mode of Pat1 in the revised text of our manuscript.

As for point (ii): We did realize that our arguments based on human Edc3 in vitro analytical data from the Ling et al., 2008 MCB paper were weak and we certainly were not trying to suggest that Ling et al. misinterpreted their data. We used their analytical data to argue that Edc3 has a large, disordered region between the Lsm and YjeF–N domains and that this disordered region likely has regulatory functions, e.g., regulating Edc3 dimerization in Edc3–mediated RPS28B mRNA decay. Consistent with our model that Edc3 may exist as both monomeric and dimeric forms, the Ling et al. graph of molar mass distribution of full–length human Edc3 SV (Figure 1E of Ling et al., 2008) indicates two forms of human Edc3. One minor peak has a molar mass about 55,000 Da, and one major peak has a molar mass slightly under 100,000 Da. The minor peak appears to be an Edc3 monomer and the major peak appears to be an Edc3 dimer. Our response to Reviewer #2 sought to call attention to the possibility that human Edc3 has some unusual biochemical properties that may regulate its ability to dimerize. Reviewer #3 has provided valuable technical insights into the Ling et al. paper, but our data remain consistent with the existence of dimeric and monomeric states of both Edc3 and Scd6. We summarized our reasoning for different Edc3 states in the Discussion section (page 35, lines 799 to 810), and thus did not make any changes to the text of our manuscript that addressed this point further.

Finally, we have eliminated all words that were potentially dismissive of biochemical studies. We apologize for the apparent disrespect and assure you that was not our intent.